# A disease-associated gene desert directs macrophage inflammation through ETS2

C. T. Stankey[1,2,3,35], C. Bourges[1,35], L. M. Haag[4,35], T. Turner-Stokes[1,2], A. P. Piedade[1], C. Palmer-Jones[5,6], I. Papa[1], M. Silva dos Santos[7], Q. Zhang[8], A. J. Cameron[9], A. Legrini[9], T. Zhang[9], C. S. Wood[9], F. N. New[10], L. O. Randzavola[2], L. Speidel[11,12], A. C. Brown[13], A. Hall[14,15], F. Saffioti[6,14], E. C. Parkes[1], W. Edwards[16], H. Direskeneli[17], P. C. Grayson[18], L. Jiang[19], P. A. Merkel[20,21], G. Saruhan-Direskeneli[22], A. H. Sawalha[23,24,25,26], E. Tombetti[27,28], A. Quaglia[15,29], D. Thorburn[6,14], J. C. Knight[13,30,31], A. P. Rochford[5,6], C. D. Murray[5,6], P. Divakar[10], M. Green[32], E. Nye[32], J. I. MacRae[7], N. B. Jamieson[9], P. Skoglund[11], M. Z. Cader[16,33], C. Wallace[16,34], D. C. Thomas[16,33] & J. C. Lee[1,5,6 ✉]

Increasing rates of autoimmune and inflammatory disease present a burgeoning threat to human health[1]. This is compounded by the limited efficacy of available treatments[1] and high failure rates during drug development[2], highlighting an urgent need to better understand disease mechanisms. Here we show how functional genomics could address this challenge. By investigating an intergenic haplotype on chr21q22—which has been independently linked to inflammatory bowel disease, ankylosing spondylitis, primary sclerosing cholangitis and Takayasu's arteritis[3–6]—we identify that the causal gene, *ETS2*, is a central regulator of human inflammatory macrophages and delineate the shared disease mechanism that amplifies *ETS2* expression. Genes regulated by ETS2 were prominently expressed in diseased tissues and more enriched for inflammatory bowel disease GWAS hits than most previously described pathways. Overexpressing *ETS2* in resting macrophages reproduced the inflammatory state observed in chr21q22-associated diseases, with upregulation of multiple drug targets, including TNF and IL-23. Using a database of cellular signatures[7], we identified drugs that might modulate this pathway and validated the potent anti-inflammatory activity of one class of small molecules in vitro and ex vivo. Together, this illustrates the power of functional genomics, applied directly in primary human cells, to identify immune-mediated disease mechanisms and potential therapeutic opportunities.

Nearly 5% of humans live with an autoimmune or inflammatory disease. These heterogeneous conditions, ranging from Crohn's disease and ulcerative colitis (collectively inflammatory bowel disease (IBD)) to psoriasis and lupus, all require better therapies, but only 10% of drugs entering clinical development ever become approved treatments[2]. This high failure rate is mainly due to a lack of efficacy[8] and reflects our poor understanding of disease mechanisms. Genetics provides a unique opportunity to address this, with hundreds of loci now directly linked to the pathogenesis of immune-mediated diseases[9]. Indeed, drugs that target pathways implicated by genetics have a far higher chance of being effective[10].

However, to fully realize the potential of genetics, knowledge of where risk variants lie must be translated into an understanding of how they drive disease[9]. Animal models can help with this, especially for

[1]Genetic Mechanisms of Disease Laboratory, The Francis Crick Institute, London, UK. [2]Department of Immunology and Inflammation, Imperial College London, London, UK. [3]Washington University School of Medicine, St Louis, MO, USA. [4]Division of Gastroenterology, Infectious Diseases and Rheumatology, Charité–Universitätsmedizin Berlin, Berlin, Germany. [5]Department of Gastroenterology, Royal Free Hospital, London, UK. [6]Institute for Liver and Digestive Health, Division of Medicine, University College London, London, UK. [7]Metabolomics STP, The Francis Crick Institute, London, UK. [8]Genomics of Inflammation and Immunity Group, Human Genetics Programme, Wellcome Sanger Institute, Hinxton, UK. [9]Wolfson Wohl Cancer Centre, School of Cancer Sciences, University of Glasgow, Glasgow, UK. [10]NanoString Technologies, Seattle, WA, USA. [11]Ancient Genomics Laboratory, The Francis Crick Institute, London, UK. [12]Genetics Institute, University College London, London, UK. [13]Wellcome Centre for Human Genetics, University of Oxford, Oxford, UK. [14]The Sheila Sherlock Liver Centre, Royal Free Hospital, London, UK. [15]Department of Cellular Pathology, Royal Free Hospital, London, UK. [16]Cambridge Institute of Therapeutic Immunology and Infectious Disease, University of Cambridge, Cambridge, UK. [17]Department of Internal Medicine, Division of Rheumatology, Marmara University, Istanbul, Turkey. [18]Systemic Autoimmunity Branch, NIAMS, National Institutes of Health, Bethesda, MD, USA. [19]Department of Rheumatology, Zhongshan Hospital, Fudan University, Shanghai, China. [20]Division of Rheumatology, Department of Medicine, University of Pennsylvania, Philadelphia, PA, USA. [21]Division of Epidemiology, Department of Biostatistics, Epidemiology and Informatics, University of Pennsylvania, Philadelphia, PA, USA. [22]Department of Physiology, Istanbul University, Istanbul Faculty of Medicine, Istanbul, Turkey. [23]Division of Rheumatology, Department of Pediatrics, University of Pittsburgh, Pittsburgh, PA, USA. [24]Division of Rheumatology and Clinical Immunology, Department of Medicine, University of Pittsburgh, Pittsburgh, PA, USA. [25]Lupus Center of Excellence, University of Pittsburgh, Pittsburgh, PA, USA. [26]Department of Immunology, University of Pittsburgh, Pittsburgh, PA, USA. [27]Department of Biomedical and Clinical Sciences, Milan University, Milan, Italy. [28]Internal Medicine and Rheumatology, ASST FBF-Sacco, Milan, Italy. [29]UCL Cancer Institute, London, UK. [30]Chinese Academy of Medical Sciences Institute, Nuffield Department of Medicine, University of Oxford, Oxford, UK. [31]NIHR Comprehensive Biomedical Research Centre, Oxford, UK. [32]Experimental Histopathology STP, The Francis Crick Institute, London, UK. [33]Department of Medicine, University of Cambridge, Cambridge, UK. [34]MRC Biostatistics Unit, Cambridge Institute of Public Health, Cambridge, UK. [35]These authors contributed equally: C. T. Stankey, C. Bourges, L. M. Haag. ✉e-mail: james.lee@crick.ac.uk

coding variants in conserved genes[11,12]. Unfortunately, most risk variants do not lie in coding DNA, but in less-well-conserved, non-coding genomic regions. Resolving the biology at these loci is a formidable task, as the same DNA sequence can function differently depending on the cell type and/or external stimuli[9]. Most non-coding variants are thought to affect gene regulation[13], but difficulties identifying causal genes, which may lie millions of bases away, and causal cell types, which may only express implicated genes under certain conditions, have hindered efforts to identify disease mechanisms. For example, although genome-wide association studies (GWASs) have identified over 240 IBD risk loci[3], including several possible drug targets, fewer than 10 have been mechanistically resolved.

## Molecular mechanisms at chr21q22

Some genetic variants predispose to multiple diseases, highlighting both their biological importance and an opportunity to study shared disease mechanisms. One notable example is an intergenic region on chromosome 21q22 (chr21q22), where the major allele haplotype predisposes to five inflammatory diseases[3–6]. Such regions, which were originally termed 'gene deserts' owing to their lack of coding genes, often contain GWAS hits but are poorly understood. To test for a shared disease mechanism, we performed co-localization analyses and confirmed that the genetic basis for every disease was the same, meaning that a common causal variant(s) and a shared molecular effect was responsible (Fig. 1a and Extended Data Fig. 1). As these heterogeneous diseases are all immune mediated, we reasoned that this locus must contain a distal enhancer that functioned in immune cells. By examining H3K27ac chromatin immunoprecipitation–sequencing (ChIP–seq) data, which marks active enhancers and promoters, we identified a monocyte/macrophage-specific enhancer within the locus (Fig. 1b). Monocytes and macrophages have a key role in many immune-mediated diseases, producing cytokines that are often targeted therapeutically[14].

We next sought to identify the gene regulated by this enhancer. Although the associated locus lacks coding genes, there are several nearby candidates that have been highlighted in previous studies, including *PSMG1*, *BRWD1* and *ETS2* (refs. 3–6,15) (Fig. 1a). Using promoter-capture Hi-C and expression quantitative locus (eQTL) data from human monocytes (Methods), we found that the disease-associated locus physically interacts with the promoter of *ETS2*–the most distant candidate gene (around 290 kb away)–and that the risk haplotype correlates with higher *ETS2* expression (Fig. 1c). Indeed, increased *ETS2* expression in monocytes and macrophages, either at rest or after early exposure to bacteria, was found to have the same genetic basis as inflammatory disease risk (Extended Data Fig. 1c). To directly confirm that *ETS2* was causal, we used CRISPR–Cas9 to delete the 1.85 kb enhancer region in primary human monocytes before culturing these cells with inflammatory ligands, including TNF (a pro-inflammatory cytokine), prostaglandin E2 (a pro-inflammatory lipid) and Pam3CSK4 (a TLR1/2 agonist) (TPP model; Fig. 1d and Extended Data Fig. 2a–c). This model was designed to mimic chronic inflammation[16], and better resembles disease macrophages than classical IFNγ-driven or IL-4-driven models[17] (Extended Data Fig. 2). As flow cytometry antibodies were not available for the candidate genes, we used PrimeFlow to measure the dynamics of mRNA expression and detected increased levels of all three genes (*ETS2*, *BRWD1* and *PSMG1*) after TPP stimulation of unedited monocytes (Fig. 1e). Deletion of the chr21q22 enhancer did not affect *BRWD1* or *PSMG1* expression, but the upregulation of *ETS2* was profoundly reduced (Fig. 1f), confirming that this pleiotropic locus contains a distal *ETS2* enhancer.

To identify the causal variant, we performed statistical fine-mapping in a large IBD GWAS[3]. Unfortunately, this did not resolve the association owing to high linkage disequilibrium between candidate single-nucleotide polymorphisms (SNPs) (Methods and Fig. 1g). We therefore used a functional approach to first delineate the active enhancers at the locus, and then assess whether any candidate SNPs might alter enhancer activity. This method, massively parallel reporter assay (MPRA), simultaneously tests enhancer activity in thousands of short DNA sequences by coupling each to a uniquely barcoded reporter gene[18]. Sequences that alter gene expression are identified by normalizing the barcode counts in mRNA, extracted from transfected cells, to their matching counts in the input DNA library. After adapting MPRA for primary macrophages (Methods and Extended Data Fig. 3), we synthesized a pool of overlapping oligonucleotides to tile the 2 kb region containing all candidate SNPs, and included oligonucleotides with either risk or non-risk alleles for every variant. The resulting library was transfected into inflammatory macrophages from multiple donors, ensuring that a physiological repertoire of transcription factors could interact with the genomic sequences. Using a sliding-window analysis, we identified a single 442 bp focus of enhancer activity (chromosome 21: 40466236–40466677, hg19; Fig. 1h) that contained three (out of seven) candidate SNPs. Two of these polymorphisms were transcriptionally inert, but the third (rs2836882) had the strongest expression-modulating effect of any candidate SNP, with the risk allele (G) increasing transcription, consistent with the *ETS2* eQTL (Fig. 1h and Extended Data Fig. 1b). This SNP was in the credible set of every co-localizing molecular trait, and lay within a macrophage PU.1 ChIP–seq peak (Fig. 1i). PU.1 is a non-classical pioneer factor in myeloid cells[19] that can bind to DNA, initiate chromatin remodelling (thereby enabling other transcription factors to bind) and activate transcription[20]. To determine whether rs2836882 might affect PU.1 binding, we identified PU.1 ChIP–seq data from heterozygous macrophages and tested for allelic imbalances in binding. Despite not lying within a canonical PU.1 motif, strong allele-specific binding was detected, with over fourfold greater binding to the rs2836882 risk allele (Fig. 1i,j). This was replicated by genotyping PU.1-bound DNA in macrophages from five heterozygous donors (Extended Data Fig. 4a–f). Moreover, assay for transposase-accessible chromatin with sequencing (ATAC–seq) analysis of monocytes and macrophages from rs2836882 heterozygotes revealed allelic differences in chromatin accessibility that were consistent with differential binding of a pioneer factor (Fig. 1k and Extended Data Fig. 4g).

To test for allele-specific enhancer activity at the endogenous locus, we performed H3K27ac ChIP–seq analysis of inflammatory macrophages from rs2836882 major and minor allele homozygotes. While most chr21q22 enhancer peaks were similar between these donors, the enhancer activity overlying rs2836882 was significantly stronger in major (risk) allele homozygotes (Fig. 1l and Extended Data Fig. 4h), contributing to an approximate 2.5-fold increase in activity across the locus (Extended Data Fig. 4i). Collectively, these data reveal a mechanism whereby the putative causal variant at chr21q22–identified by its functional effects in primary macrophages–promotes binding of a pioneer factor, enhances chromatin accessibility and increases activity of a distal *ETS2* enhancer.

## Macrophage inflammation requires ETS2

ETS2 is an ETS-family transcription factor and proto-oncogene[21], but its exact role in human macrophages is unclear, with previous studies using either cell lines or complex mouse models and assessing a limited number of potential targets[22–26]. This has led to contradictory reports, with ETS2 being described as both necessary and redundant for macrophage development[27,28], and both pro- and anti-inflammatory[22–26]. To clarify the role of *ETS2* in human macrophages, and determine how dysregulated *ETS2* expression might contribute to disease, we first used a CRISPR–Cas9-based loss-of-function approach (Fig. 2a). To control for off-target effects, two gRNAs targeting different *ETS2* exons were designed, validated and individually incorporated into Cas9 ribonucleoproteins for transfection into primary monocytes. These produced on-target editing in around 90% and 79% of cells, respectively, and effectively

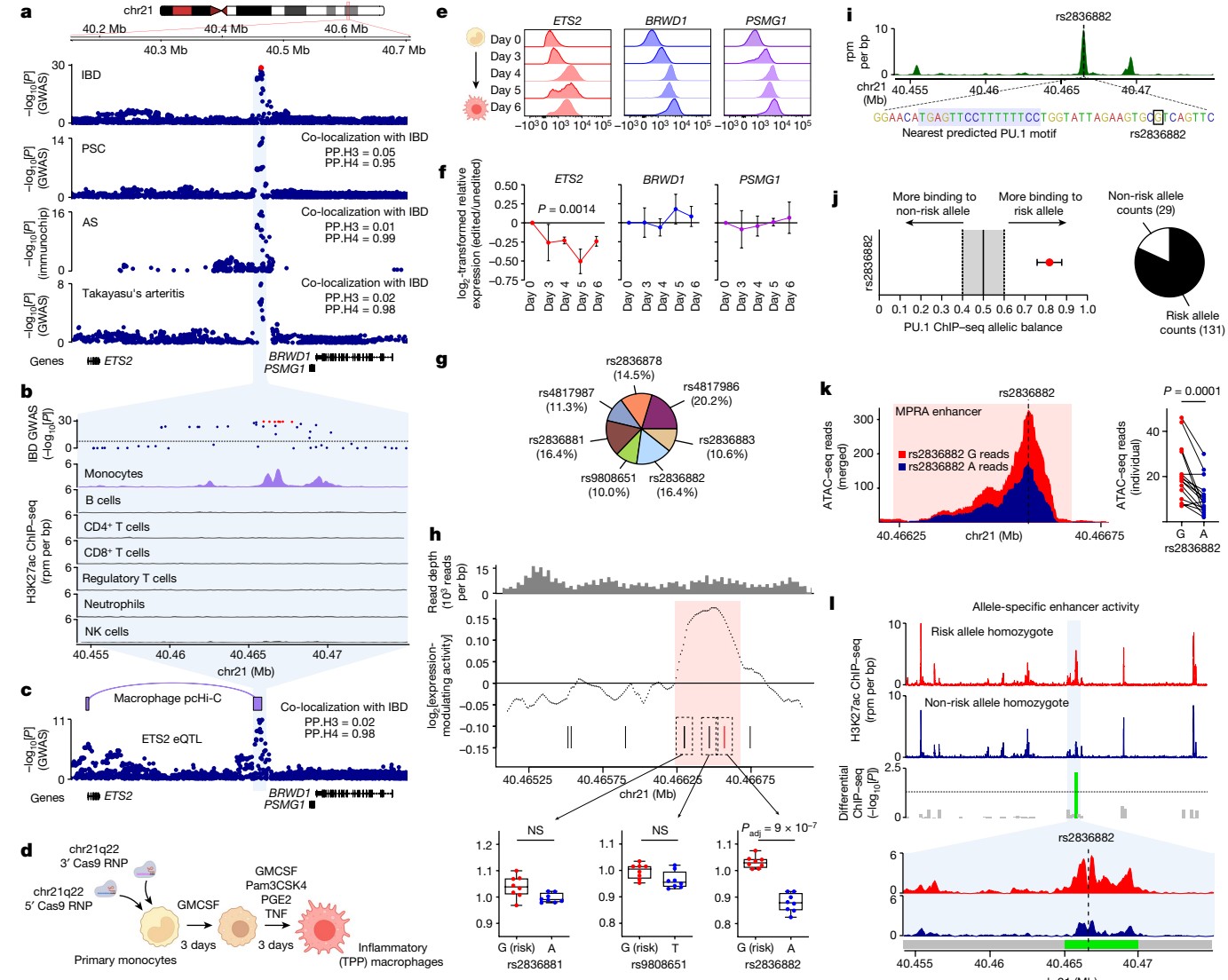

**Fig. 1 | Resolving molecular mechanisms at chr21q22. a**, Disease associations at chr21q22. The red points denote the IBD 99% credible set. Co-localization results for each disease versus IBD. PP.H3, posterior probability of independent causal variants; PP.H4, posterior probability of shared causal variant. **b**, Immune cell H3K27ac ChIP–seq at chr21q22. IBD GWAS results are shown. NK cells, natural killer cells. rpm, reads per million. **c**, The *ETS2* eQTL in resting monocytes, with co-localization versus IBD association. Macrophage promoter-capture Hi-C (pcHi-C) data at the disease-associated locus. **d**, Experimental schematic for studying the chr21q22 locus in inflammatory (TPP) macrophages. **e**, *ETS2*, *BRWD1* and *PSMG1* mRNA expression during TPP stimulation, measured using PrimeFlow RNA assays. Data are from one representative donor of four. **f**, Relative *ETS2*, *BRWD1* and *PSMG1* expression (mean fluorescence intensity (MFI)) in chr21q22-edited macrophages versus unedited cells. *n* = 4. Data are mean ± s.e.m. Statistical analysis was performed using two-way analysis of variance (ANOVA). **g**, SuSiE fine-mapping posterior probabilities for IBD-associated SNPs at chr21q22 (99% credible set). **h**, Macrophage MPRA at chr21q22. Data are oligo coverage (top), enhancer activity (sliding-window analysis with significant enhancer activity highlighted; middle) and expression-modulating effects of SNPs within the enhancer (bottom). For the box plots, the centre line shows the median, the box limits show the interquartile range, and the whiskers represent the minimum and maximum values. *n* = 8. False-discovery rate (FDR)-adjusted *P* values were calculated using QuASAR-MPRA (two-sided). **i**, Inflammatory macrophage PU.1 ChIP–seq peaks at chr21q22. Bottom, magnification of the location of rs2836882 and the nearest predicted PU.1 motif. **j**, BaalChIP analysis of allele-specific PU.1 ChIP–seq binding at rs2836882 in two heterozygous macrophage datasets (data are mean ± 95% posterior distribution of allelic balance). Total counts shown as a pie chart. **k**, Allele-specific ATAC–seq reads at rs2836882 in monocytes from 16 heterozygous donors (including healthy controls and patients with ankylosing spondylitis). Statistical analysis was performed using two-sided Wilcoxon matched-pair tests. **l**, H3K27ac ChIP–seq data from risk (top) or non-risk (bottom) allele homozygotes at rs2836882. Data are shown from two out of four donors. FDR-corrected *P* values were calculated using MEDIPS (two-sided). The diagrams in **d** and **e** were created using BioRender.

reduced *ETS2* expression (Extended Data Fig. 2d–f). Cell viability and macrophage marker expression were unaffected, suggesting that *ETS2* was not required for macrophage survival or differentiation (Extended Data Fig. 2g,h). By contrast, pro-inflammatory cytokine production, including IL-6, IL-8 and IL-1β, was markedly reduced after *ETS2* disruption (Fig. 2b), whereas IL-10—an anti-inflammatory cytokine—was less affected. TNF was not assessed as it had been added exogenously. We

next investigated whether other macrophage functions were affected. Using fluorescently labelled particles that are detectable by flow cytometry, we found that phagocytosis was similarly impaired after *ETS2* disruption (Fig. 2c). We also tested extracellular reactive oxygen species (ROS) production—a major contributor to inflammatory tissue damage[29]. Disrupting *ETS2* profoundly reduced the macrophage oxidative burst—most likely by decreasing expression of key NADPH oxidase

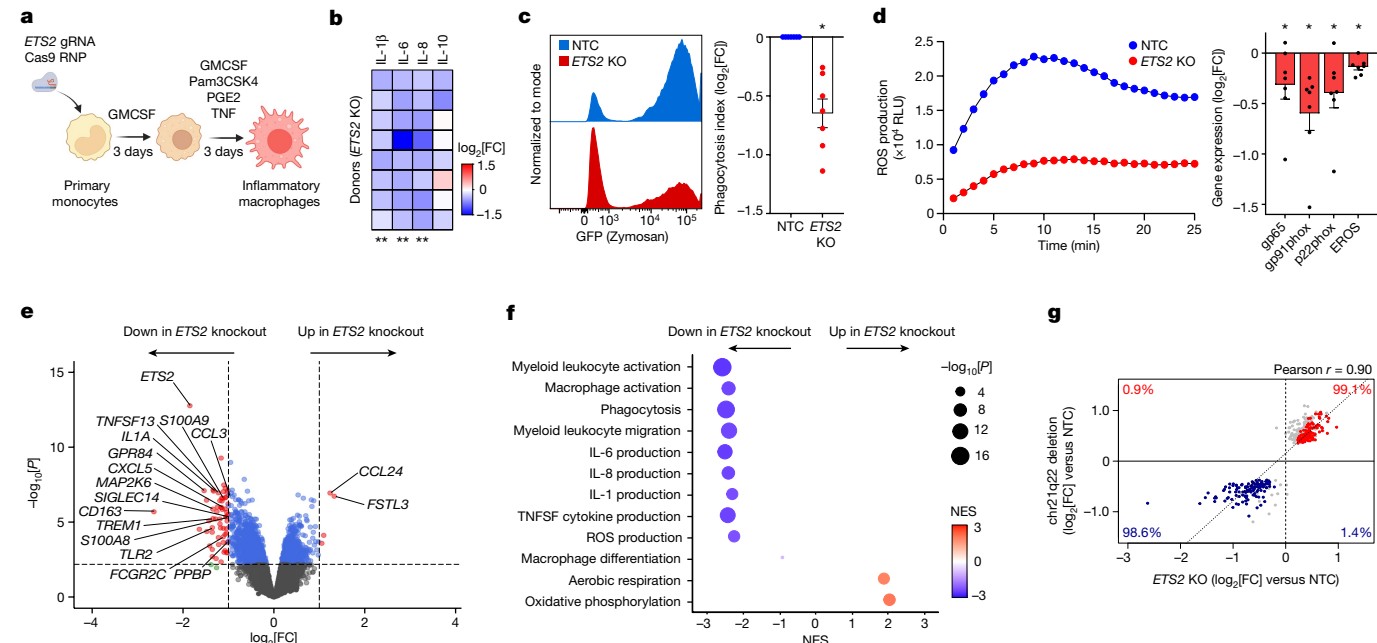

**Fig. 2 | *ETS2* is essential for macrophage inflammatory responses.**
**a**, Experimental schematic for studying *ETS2* in inflammatory (TPP) macrophages. The diagram was created using BioRender. **b**, Cytokine secretion after *ETS2* disruption. Heat map of relative cytokine levels from *ETS2*-edited versus unedited macrophages. *n* = 8. **c**, Phagocytosis of fluorescently labelled zymosan particles by *ETS2*-edited and unedited macrophages (non-targeting control (NTC)) (left). Data are from one representative donor out of seven. Right, the phagocytosis index (the product of the proportion and MFI of phagocytosing cells). *n* = 7. **d**, ROS production by *ETS2*-edited and unedited macrophages. Data from one representative donor out of six (left). Right, NADPH oxidase component expression in *ETS2*-edited and unedited macrophages (western blot densitometry). *n* = 7. Source gels are shown in Supplementary Fig. 1. RLU, relative light units. **e**, RNA-seq analysis of

differentially expressed genes in *ETS2*-edited versus unedited TPP macrophages (limma with voom transformation, two-sided). *n* = 8. The horizontal line denotes the FDR-adjusted significance threshold. **f**, fGSEA of differentially expressed genes between *ETS2*-edited and unedited TPP macrophages. The results of selected GO Biological Pathways are shown. The dot size denotes the unadjusted *P* value (two-sided), and the colour denotes normalized enrichment score (NES). **g**, The log$_2$[fold change (FC)] of genes differentially expressed by chr21q22 enhancer deletion, plotted against their fold change after *ETS2* editing. The percentages denote upregulated (red) and downregulated (blue) genes. The coloured points (blue or red) represent differentially expressed genes after *ETS2* editing (FDR < 0.1, two-sided). For **c** and **d**, data are mean ± s.e.m. Statistical analysis was performed using two-sided Wilcoxon tests (**b**–**d**); *\*P* < 0.05.

components (Fig. 2d and Extended Data Fig. 5a). Together, these data suggest that *ETS2* is essential for multiple inflammatory functions in human macrophages.

To understand the molecular basis for these effects, we performed RNA sequencing (RNA-seq) of *ETS2*-edited and unedited inflammatory macrophages from multiple donors. Disrupting *ETS2* led to widespread transcriptional changes, with reduced expression of many inflammatory genes (Fig. 2e). These included cytokines (such as *TNFSF10/TRAIL*, *TNFSF13*, *IL1A* and *IL1B*), chemokines (such as *CXCL3*, *CXCL5*, *CCL2* and *CCL5*), secreted effector molecules (such as *S100A8*, *S100A9*, *MMP14* and *MMP9*), cell surface receptors (such as *FCGR2A*, *FCGR2C* and *TREM1*), pattern-recognition receptors (such as *TLR2*, *TLR6* and *NOD2*) and signalling molecules (such as *MAP2K*, *GPR84* and *NLRP3*). To better characterize the pathways affected, we performed gene set enrichment analysis (fGSEA) using the Gene Ontology (GO) Biological Pathways dataset. This corroborated the functional deficits, with the most negatively enriched pathways (downregulated by *ETS2* disruption) being related to macrophage activation, inflammatory cytokine production, phagocytosis and ROS production (Fig. 2f). Genes involved in macrophage migration were also downregulated, but those relating to monocyte-to-macrophage differentiation were unaffected—consistent with *ETS2* being required for inflammatory functions but not for monocyte-derived macrophage development. Fewer genes were upregulated after *ETS2* disruption (Fig. 2e), but positive enrichment was noted for aerobic respiration and oxidative phosphorylation (OXPHOS; Fig. 2f)—metabolic processes that are linked to anti-inflammatory phenotypes[30]. Notably, these transcriptional effects were not due to

major changes in chromatin accessibility, although enhancer activity was generally reduced (Extended Data Fig. 2j,k). As expected, deletion of the chr21q22 enhancer phenocopied both the transcriptional and functional effects of disrupting *ETS2* (Fig. 2g and Extended Data Fig. 5a–e). Collectively, these data identify an essential role for *ETS2* in macrophage inflammatory responses, which could explain why dysregulated *ETS2* expression predisposes to disease. Indeed, differential expression of ETS2-regulated genes was observed in resting (M0) macrophages from patients with IBD stratified by rs2836882 genotype (matched for age, sex, therapy and disease activity) (Extended Data Fig. 5f).

## ETS2 coordinates macrophage inflammation

We next studied the effects of increasing *ETS2* expression, as this is what drives disease risk. To do this, we optimized a method for controlled overexpression of target genes in primary macrophages through transfection of in vitro transcribed mRNA that was modified to minimize immunogenicity (Fig. 3a, Methods and Extended Data Fig. 3f). Resting, non-activated macrophages were transfected with *ETS2* mRNA or its reverse complement, thereby controlling for mRNA quantity, length and purine/pyrimidine composition (Fig. 3b). After transfection, cells were exposed to low-dose lipopolysaccharide to initiate a low-grade inflammatory response that could potentially be amplified (Fig. 3a). We found that overexpressing *ETS2* increased pro-inflammatory cytokine secretion, while IL-10 was again less affected (Extended Data Fig. 3g). To better characterize this response, we performed RNA-seq and

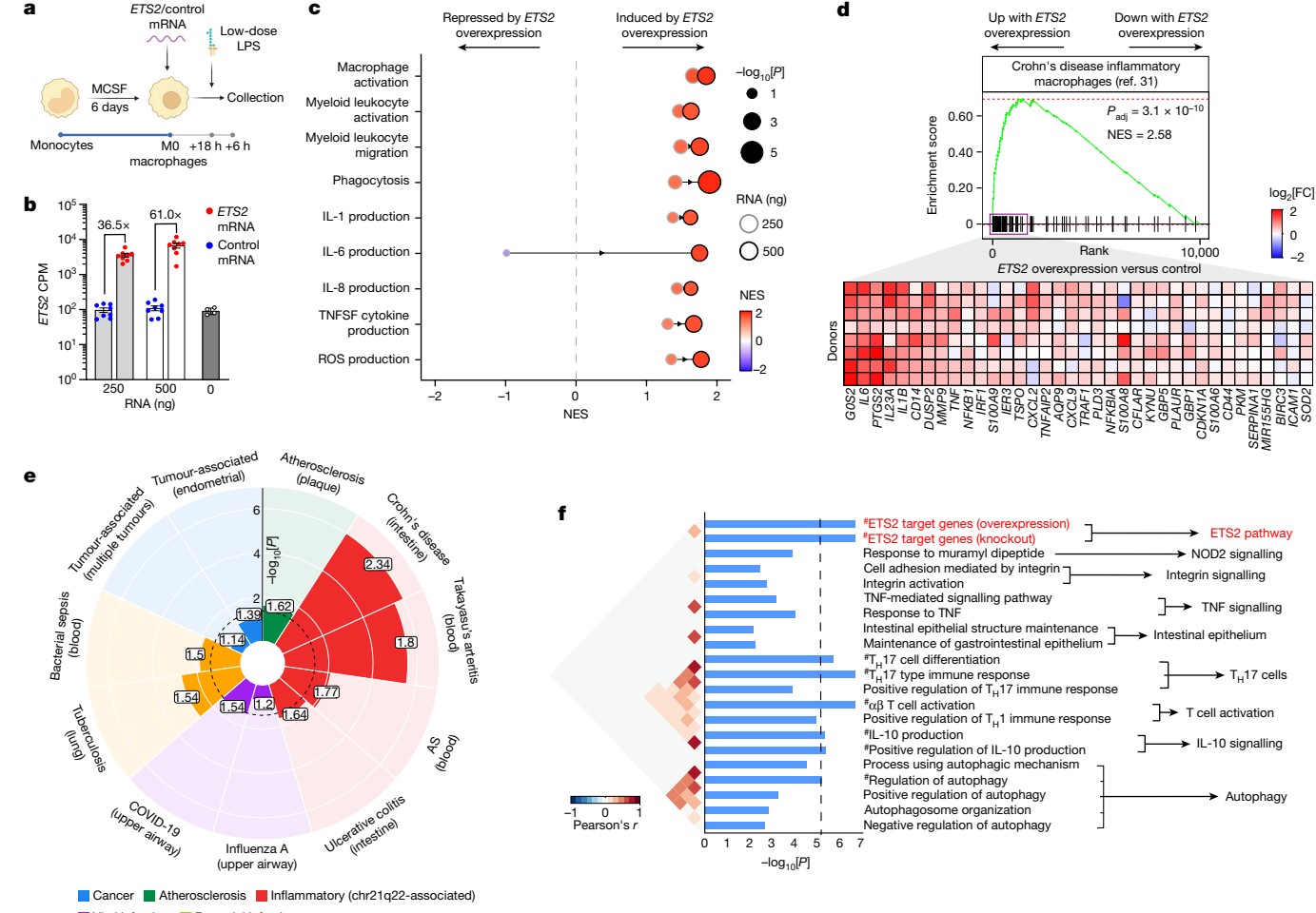

**Fig. 3 | ETS2 orchestrates macrophage inflammatory responses.**
**a**, Experimental schematic for studying the effects of *ETS2* overexpression. The diagram was created using BioRender. **b**, *ETS2* mRNA levels in transfected ($n = 8$) or untransfected (from a separate experiment) macrophages. Data are mean ± s.e.m. CPM, counts per million. **c**, fGSEA analysis of differentially expressed genes between *ETS2*-overexpressing and control macrophages. Results shown for pathways downregulated by ETS2 disruption. The dot size denotes the unadjusted *P* value (two-sided), the colour denotes NES and the border colour denotes the quantity of transfected mRNA. **d**, fGSEA analysis of a Crohn's disease intestinal macrophage signature in *ETS2*-overexpressing

macrophages (versus control). FDR *P*-value, two-sided (top). Heat map of the relative expression of leading-edge genes after *ETS2* overexpression (500 ng mRNA; bottom). **e**, Enrichment of macrophage signatures from patients with the indicated diseases in *ETS2*-overexpressing macrophages (versus control). The colour denotes the disease category, the numbers denote the NES and the dashed line denotes FDR = 0.05. The Crohn's disease signature is from a different study to that shown in **d**. AS, ankylosing spondylitis. **f**, SNPsea analysis of genes tagged by 241 IBD SNPs within *ETS2*-regulated genes (red) and known IBD pathways (black). Significant pathways (Bonferroni-corrected $P < 0.05$) are indicated by hash symbols (#).

re-examined the inflammatory pathways that required *ETS2*. Notably, all of these pathways—including macrophage activation, cytokine production, ROS production, phagocytosis and migration—were induced in a dose-dependent manner by *ETS2* overexpression, with greater enrichment of every pathway when more *ETS2* mRNA was transfected (Fig. 3c). This shows that *ETS2* is both necessary and sufficient for inflammatory responses in human macrophages, consistent with being a central regulator of effector functions, with dysregulation directly linked to disease.

## ETS2 has a key pathogenic role in IBD

To test whether *ETS2* contributes to macrophage phenotypes in disease, we compared the effects of overexpressing *ETS2* in resting macrophages with a single-cell RNA-seq (scRNA-seq) signature from intestinal macrophages in Crohn's disease[31]. *ETS2* overexpression induced a transcriptional state that closely resembled disease macrophages, with core (leading edge) enrichment of most signature genes,

including several therapeutic targets (Fig. 3d). Similar enrichment was observed with myeloid signatures from other chr21q22-associated diseases and, to a lesser extent, from active bacterial infection, but not for signatures from influenza and tumour macrophages, suggesting that ETS2 was not simply inducing generic activation (Fig. 3e).

Given the central role of *ETS2* in inflammatory macrophages and the importance of these cells in disease, we hypothesized that other genetic associations would also implicate this pathway. A major goal of GWAS was to identify disease pathways, but this has proven to be challenging due to a paucity of confidently identified causal genes and variants[9]. To determine whether the macrophage ETS2 pathway was enriched for disease genetics, we focused on IBD as this has more GWAS hits than any other chr21q22-associated disease. Encouragingly, a network of 33 IBD-associated genes in intestinal mucosa was previously found to be enriched for predicted ETS2 motifs[32]. Examining the genes that were consistently downregulated in *ETS2*-edited macrophages (adjusted *P* ($P_{adj}$) < 0.05 for both gRNAs), we identified over

20 IBD-risk-associated genes, including many thought to be causal at their respective loci[3,33] (Extended Data Table 1). These included genes that are known to affect macrophage biology (such as *SP140*, *LACC1*, *CCL2*, *CARD9*, *CXCL5*, *TLR4*, *SLAMF8* and *FCGR2A*) and some that are highly expressed in macrophages but not linked to specific pathways (such as *ADCY7*, *PTPRC*, *TAGAP*, *PTAFR* and *PDLIM5*). A polygenic risk score comprising these variants associated with features of more severe IBD across 18,249 patients, including earlier disease onset, increased the need for surgery, and stricturing or fistulating complications in Crohn's disease (Extended Data Fig. 6a–h). To better test the enrichment of IBD GWAS hits in ETS2-mediated inflammation, and compare this with known disease pathways, we used SNPsea[34]—a method to identify pathways affected by disease loci. In total, 241 IBD loci were tested for enrichment in 7,658 GO Biological Pathways and 2 overlapping lists of ETS2-regulated genes (either those downregulated by *ETS2* disruption or upregulated by *ETS2* overexpression). Statistical significance was computed using 5 million matched null SNP sets, and pathways implicated by IBD genetics were extracted for comparison. Notably, IBD-associated SNPs were more significantly enriched in the macrophage ETS2 pathway than in many IBD pathways, with not a single null SNP set being more enriched in either ETS2-regulated gene list (Fig. 3f and Extended Data Fig. 6i). SNPs associated with primary sclerosing cholangitis (PSC), ankylosing spondylitis and Takayasu's arteritis were also enriched in ETS2-target genes (Extended Data Fig. 6j). Collectively, this suggests that macrophage ETS2 signalling has a central role in multiple inflammatory diseases.

## ETS2 has distinct inflammatory effects

We next investigated how *ETS2* might control such diverse macrophage functions. Studying ETS2 biology is challenging because no ChIP-grade antibodies exist, precluding direct identification of its transcriptional targets. We therefore first used a guilt-by-association approach to identify genes that were co-expressed with *ETS2* across 67 human macrophage activation conditions (comprising 28 stimuli and various durations of exposure)[16]. This identified *PFKFB3*—encoding the rate-limiting enzyme of glycolysis—as the most highly co-expressed gene, with *HIF1A* also highly co-expressed (Fig. 4a). Together, these genes facilitate a 'glycolytic switch' that is required for myeloid inflammatory responses[35]. We therefore hypothesized that *ETS2* might control inflammation through metabolic reprogramming—a possibility supported by OXPHOS genes being negatively correlated with *ETS2* (Fig. 4a) and upregulated after *ETS2* disruption (Fig. 2f). To assess the metabolic consequences of disrupting *ETS2*, we quantified label incorporation from [13]C-glucose in edited and unedited TPP macrophages using gas chromatography coupled with mass spectrometry (GC–MS). Widespread modest reductions in labelled and total glucose metabolites were detected after *ETS2* disruption (Fig. 4b and Extended Data Fig. 7a–c). This affected both glycolytic and tricarboxylic acid (TCA) cycle metabolites, with significant reductions in lactate, a hallmark of anaerobic glycolysis, and succinate, a key inflammatory metabolite[36]. These results are consistent with glycolytic suppression, with reductions in TCA metabolites being due to reduced flux into TCA and increased consumption by mitochondrial OXPHOS[37]. To determine whether metabolic changes accounted for ETS2-mediated inflammatory effects, we treated *ETS2*-edited macrophages with roxadustat—a HIF1α stabilizer that promotes glycolysis. This had the predicted effect on glycolysis and OXPHOS genes, but did not rescue the effects of *ETS2* disruption, either transcriptionally or functionally (Fig. 4c and Extended Data Fig. 7d,e). Thus, while disrupting *ETS2* impairs macrophage glycometabolism, this does not fully explain the differences in inflammation.

We therefore revisited whether we could directly identify ETS2-target genes. As ChIP–seq involves steps that can alter protein epitopes and prevent antibody binding (such as fixation) we tested whether any anti-ETS2 antibodies might work for cleavage under targets and release using nuclease (CUT&RUN), which does not require these steps. One antibody identified multiple significantly enriched genomic regions (peaks), of which 6,560 were reproducibly detected across two biological replicates with acceptable quality metrics[38] (Fig. 4d). These peaks were mostly located in active regulatory regions (90% in promoters or enhancers; Fig. 4d,e) and were highly enriched for both a canonical ETS2 motif (4.02-fold versus global controls; Fig. 4f) and for motifs of known ETS2 interactors, including FOS, JUN and NF-κB[39] (Extended Data Fig. 7f). After combining the biological replicates to improve peak detection, we identified ETS2 binding at genes involved in multiple inflammatory functions, including *NCF4* (ROS production), *NLRP3* (inflammasome activation) and *TLR4* (bacterial pattern recognition) (Fig. 4g). Overall, 48.3% (754 out of 1,560) of genes dysregulated after *ETS2* disruption and 50.3% (1,078 out of 2,153) of genes dysregulated after *ETS2* overexpression contained an ETS2-binding peak within their core promoter or *cis*-regulatory elements (Fig. 4h). Notably, ETS2 targets included *HIF1A*, *PFKFB3* and other glycolytic genes (such as *GPI*, *HK2* and *HK3*), consistent with the observed metabolic changes being directly induced as part of this complex inflammatory programme. Notably, we also detected ETS2 binding at the chr21q22 enhancer (Fig. 4i). This is consistent with reports that PU.1 and ETS2 can interact synergistically[40], and suggests that ETS2 might contribute to the activity of its own enhancer. Indeed, manipulating ETS2 expression altered enhancer activity in a manner consistent with positive autoregulation (Extended Data Fig. 7g–i). Together, these data implicate ETS2 as a central regulator of monocyte and macrophage inflammatory responses that is able to direct a multifaceted effector programme and create a metabolic environment that is permissive for inflammation.

## Targeting the ETS2 pathway in disease

To assess how ETS2 affects macrophage heterogeneity in diseased tissue, and whether this could be targeted therapeutically, we examined intestinal scRNA-seq data from patients with Crohn's disease and healthy control individuals[41]. Within myeloid cells, seven clusters were detected and identified using established markers and/or previous literature (Fig. 5a,b). Inflammatory macrophages (cluster 1, expressing CD209, CCL4, IL1B and FCGR3A) and inflammatory monocytes (cluster 2, expressing S100A8/A9, TREM1, CD14 and MMP9) were expanded in disease, as previously described[42], and expressed *ETS2* and ETS2-regulated genes more highly than other clusters, including tissue-resident macrophages (cluster 0, expressing C1QA, C1QB, FTL and CD63) and conventional dendritic cells (cluster 5, expressing CLEC9A, CADM1 and XCR1) (Fig. 5a,b and Extended Data Fig. 8a). Using spatial transcriptomics, a similar increase in inflammatory macrophages was observed in PSC liver tissue, with these cells being closely apposed to cholangiocytes—the main target of pathology (Fig. 5c–e). Notably, expression of ETS2-regulated genes was higher the closer macrophages were to cholangiocytes (Fig. 5f and Extended Data Fig. 8b). Indeed, using bulk RNA-seq data, we found that the transcriptional footprint of ETS2 was detectable in affected tissues from multiple chr21q22-associated diseases (Extended Data Fig. 8c).

We next examined whether this pathway could be targeted pharmacologically. Specific ETS2 inhibitors do not exist and structural analyses indicate that there is no obvious allosteric inhibitory mechanism[43]. We therefore used the NIH LINCS database to identify drugs that might modulate ETS2 activity[7]. This contains over 30,000 differentially expressed gene lists from cell lines exposed to around 6,000 small molecules. Using fGSEA, 906 signatures mimicked the effect of disrupting *ETS2* ($P_{adj} < 0.05$), including several approved IBD therapies. The largest class of drugs was MEK inhibitors (Fig. 5g), which are licensed

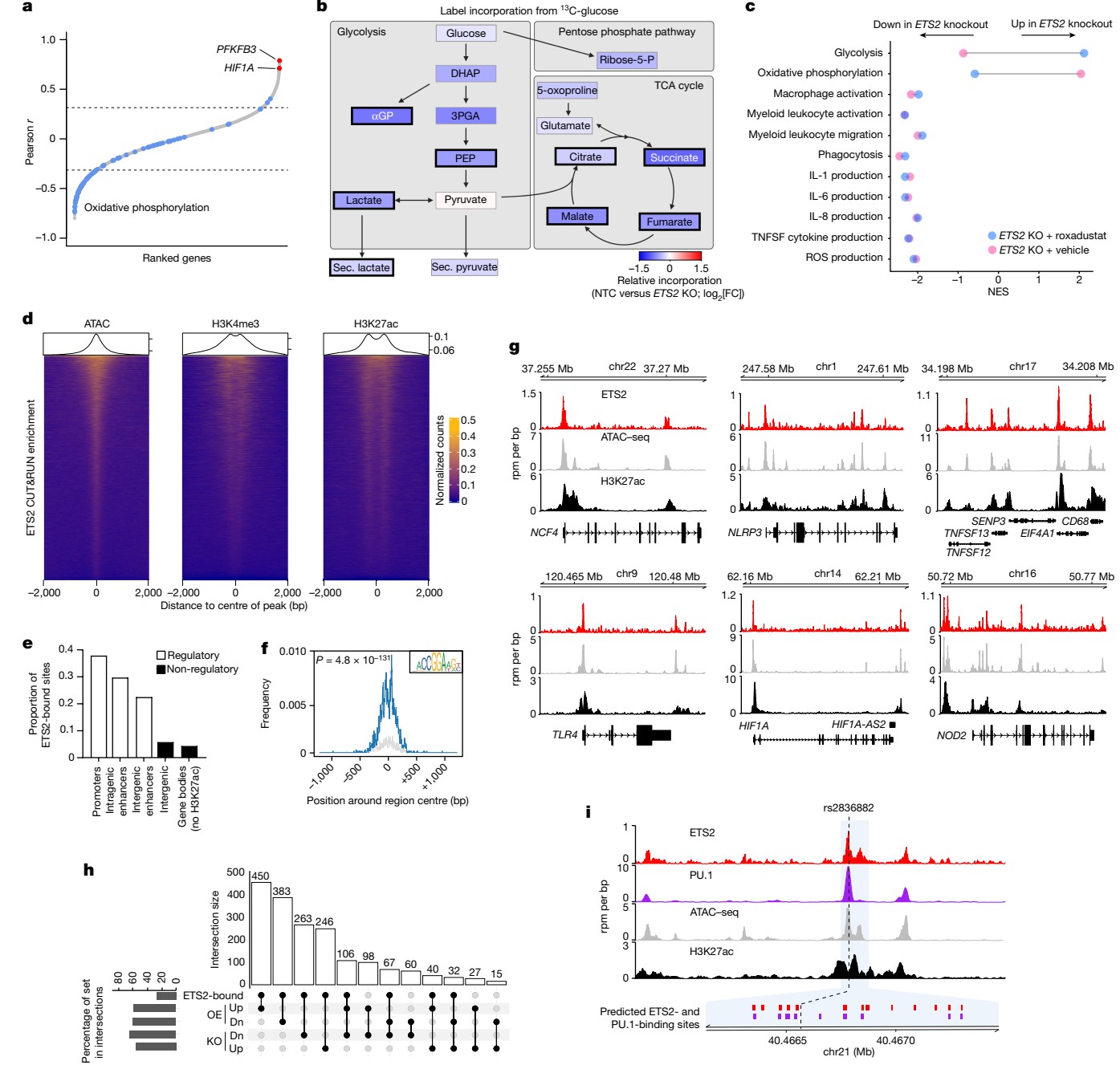

**Fig. 4 | ETS2 directs macrophage responses through transcriptional and metabolic effects. a**, Genes co-expressed with *ETS2* across 67 monocyte/macrophage activation conditions. The dotted lines denote FDR-adjusted *P* < 0.05. **b**, The effect of *ETS2* disruption on glucose metabolism. The colour denotes median log$_2$-transformed fold change in label incorporation from $^{13}$C-glucose in *ETS2*-edited versus unedited cells. The bold black border denotes *P* < 0.05 (Wilcoxon matched-pairs, two-sided). *n* = 6. Sec., secreted. **c**, fGSEA analysis of differentially expressed genes between *ETS2*-edited and unedited macrophages that were treated with roxadustat or vehicle. Results shown for pathways downregulated by ETS2 disruption. **d**, Enrichment heat maps of macrophage *ETS2* CUT&RUN peaks (IDR cut-off 0.01, *n* = 2) in 4 kb peak-centred regions from ATAC–seq (accessible chromatin), H3K4me3 ChIP–seq (active promoters) and H3K27ac ChIP–seq (active regulatory elements). **e**, Functional annotations of ETS2-binding sites (using gene coordinates and TPP macrophage H3K27ac ChIP–seq data). **f**, ETS2 motif enrichment in CUT&RUN peaks (hypergeometric *P* value, two-sided). **g**, ETS2 binding, chromatin accessibility (ATAC–seq) and regulatory activity (H3K27ac) at selected loci. **h**, Intersections between genes with ETS2 peaks in their core promoters or *cis*-regulatory elements and genes upregulated (Up) or downregulated (Dn) after *ETS2* editing (KO) or overexpression (OE). The vertical bars denote the size of overlap for lists indicated by connected dots in the bottom panel. The horizontal bars denote the percentage of gene list within intersections. **i**, ETS2 binding, PU.1 binding, chromatin accessibility and enhancer activity at chr21q22. Predicted ETS2-binding sites (red) and PU.1-binding sites (purple) shown below. The dashed line is positioned at rs2836882.

for non-inflammatory human diseases (such as neurofibromatosis). This result was not due to a single compound, but rather a class effect with multiple MEK1/2 inhibitors downregulating ETS2-target genes (Fig. 5h). This made biological sense, as MEK1/2, together with several

other targets identified, are known regulators of ETS-family transcription factors (Fig. 5g). Some of these compounds have shown benefit in animal colitis models[44], although this is often a poor indicator of clinical efficacy, as several IBD treatments are ineffective in mice and many

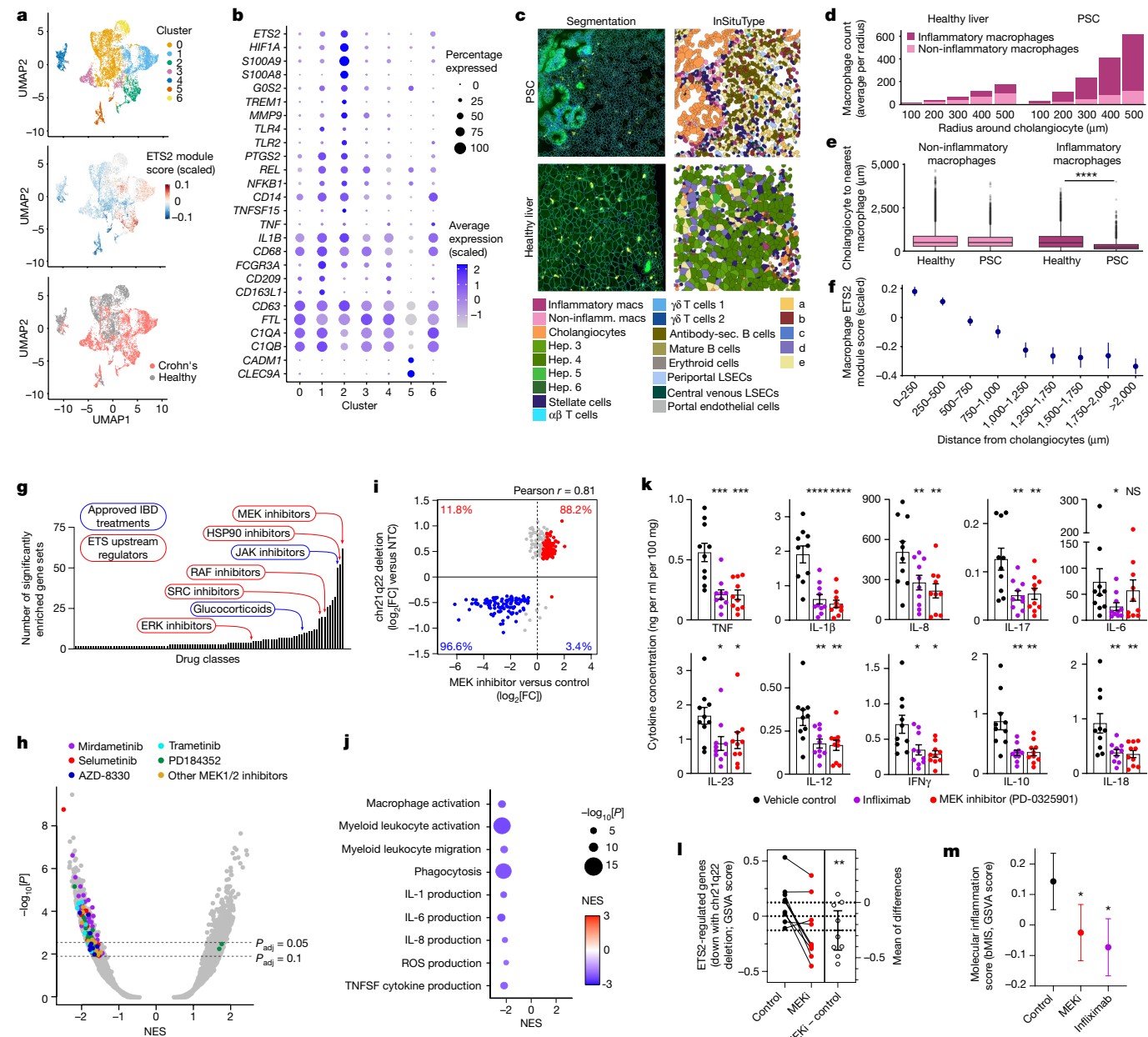

**Fig. 5 | ETS2-driven inflammation is evident in disease and can be therapeutically targeted. a**, Myeloid cell clusters in intestinal scRNA-seq from Crohn's disease and health (top). Middle, scaled expression of ETS2-regulated genes (downregulated by *ETS2* disruption). Bottom, the source of cells (disease or health). **b**, Scaled expression of selected genes. **c**, Spatial transcriptomics of PSC and healthy liver. *n* = 4. The images show representative fields of view (0.51 mm × 0.51 mm) with cell segmentation and semisupervised clustering. The main key (left and middle below images) denotes InSituType cell types; clusters a–e (far right key) are unannotated cell populations. Hep., hepatocyte; LSECs, liver sinusoidal endothelial cells; non-inflamm. macs, non-inflammatory macrophages. **d**, The number of macrophages within the indicated distances of cholangiocytes. **e**, The distance from cholangiocytes to the nearest macrophage. Data are shown as Tukey box and whisker plots. Statistical analysis was performed using two-tailed Mann–Whitney *U*-tests. Data in **d** and **e** are from 10,532 PSC and 13,322 control cholangiocytes. **f**, Scaled expression of ETS2-regulated genes in 21,067 PSC macrophages at defined distances from cholangiocytes (excluding genes used to define macrophage

subsets). **g**, Classes of drugs that phenocopy *ETS2* disruption (from the NIH LINCS database). **h**, fGSEA results for NIH LINCS drug signatures. Significant MEK inhibitor signatures are coloured by molecule. **i**, The log$_2$[fold change] of differentially expressed genes after chr21q22 enhancer deletion, plotted against their fold change after MEK inhibition. The percentages indicate the proportion of upregulated (red) and downregulated genes (blue). The coloured points (blue or red) were differentially expressed after MEK inhibition (FDR < 0.1). **j**, fGSEA of differentially expressed genes between MEK-inhibitor-treated and control TPP macrophages. Results are shown for pathways downregulated by *ETS2* disruption. The dot size denotes the unadjusted *P* value (two-sided) and the colour denotes the NES. **k**, IBD biopsy cytokine release with PD-0325901, infliximab or vehicle control. **l**, GSVA enrichment scores for chr21q22-downregulated genes in IBD biopsies after MEK inhibition. **m**, GSVA enrichment scores of a biopsy-derived molecular inflammation score (bMIS). Data are mean ± 95% CI (**f** and **l**) and mean ± s.e.m. (**k** and **m**). Statistical analysis was performed using two-sided paired *t*-tests. *n* = 10 (**k**), *n* = 9 (**l**). **\*\*P* < 0.01, \*\*\*P* < 0.001, \*\*\*\*P* < 0.0001.

compounds that improve mouse models are ineffective in humans[45]. To test whether MEK inhibition abrogates ETS2-driven inflammation in human macrophages, we treated TPP macrophages with PD-0325901, a

selective non-ATP competitive MEK inhibitor. Potent anti-inflammatory activity was observed that phenocopied the effects of disrupting *ETS2* or the chr21q22 enhancer (Fig. 5i,j and Extended Data Fig. 9a–c).

To further assess the therapeutic potential, we cultured intestinal biopsies from active, untreated IBD with either a MEK inhibitor or a negative or positive control (Methods). MEK inhibition reduced inflammatory cytokine release to similar levels as infliximab (an anti-TNF antibody that is widely used for IBD; Fig. 5k). Moreover, ETS2-regulated gene expression was reduced (Fig. 5l and Extended Data Fig. 9d) and there was improvement in a transcriptional inflammation score[46] (Fig. 5m). Together, these data show that targeting an upstream regulator of *ETS2* can abrogate pathological inflammation in a chr21q22-associated disease, and may be useful therapeutically.

## Discussion

Arguably the greatest challenge in modern genetics is to translate the success of GWAS into a better understanding of disease. Here, by studying a pleiotropic disease locus, we identify a central regulator of human macrophage inflammation and a pathogenic pathway that is potentially druggable. These findings also provide clues to the gene–environment interactions at this locus, highlighting a potential role for ETS2 in macrophage responses to bacteria. This would provide a balancing selection pressure that might explain why the risk allele remains so common (frequency of around 75% in Europeans and >90% in Africans) despite first being detected in archaic humans over 500,000 years ago (Extended Data Fig. 10).

Although ETS2 was reported to have pro-inflammatory effects on individual genes[24,25], the full extent of its inflammatory programme—with effects on ROS production, phagocytosis, glycometabolism and macrophage activation—was unclear. Moreover, without direct proof of ETS2 targets, nor studies in primary human cells, it was difficult to reconcile reports of anti-inflammatory effects at other genes[23,26]. By systematically characterizing the effects of *ETS2* disruption and overexpression in human macrophages, we identify an essential role in inflammation, delineate the mechanisms involved and show how ETS2 can induce pathogenic macrophage phenotypes. Increased *ETS2* expression may also contribute to other human pathology. For example, Down's syndrome (trisomy 21) was recently described as a cytokinopathy[47], with basal increases in multiple inflammatory cytokines, including several ETS2 targets (such as IL-1β, TNF and IL-6). Whether the additional copy of *ETS2* contributes to this phenotype is unknown, but warrants further study.

Blocking individual cytokines is a common treatment strategy in inflammatory disease[14], but emerging evidence suggests that targeting several cytokines at once may be a better approach[48]. Blocking ETS2 signalling through MEK1/2 inhibition affects multiple cytokines, including TNF and IL-23, which are targets of existing therapies, and IL-1β, which is linked to treatment resistance[49] and not directly modulated by other small molecules (such as JAK inhibitors). However, long-term MEK inhibitor use may not be ideal owing to the physiological roles of MEK in other tissues, with multiple side-effects having been reported[50]. Targeting ETS2 directly—for example, through PROTACs—or selectively delivering MEK inhibitors to macrophages through antibody–drug conjugates could overcome this toxicity, and provide a safer means of blocking ETS2-driven inflammation.

In summary, using an intergenic GWAS hit as a starting point, we have identified a druggable pathway that is both necessary and sufficient for human macrophage inflammation. Moreover, we show how genetic dysregulation of this pathway—through perturbation of pioneer factor binding at a critical long-range enhancer—predisposes to multiple diseases. This highlights the considerable, yet largely untapped, opportunity to resolve disease biology from non-coding genetic associations.

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

# Methods

## Analysis of existing data relating to chr21q22

IBD GWAS summary statistics[3] were used to perform multiple causal variant fine-mapping using susieR[51], with reference minor allele and LD information calculated from 503 European samples from 1000 Genomes phase 3 (ref. 52). All R analyses used v.4.2.1. Palindromic SNPs (A/T or C/G) and any SNPs that did not match by position or alleles were pruned before imputation using the ssimp equations reimplemented in R. This did not affect any candidate SNP at chr21q22. SuSiE fine-mapping results were obtained for *ETS2* (identifier ENSG00000157557 or ILMN_1720158) in monocyte/macrophage datasets from the eQTL Catalogue[53]. Co-localization analyses were performed comparing the chr21q22 IBD association with summary statistics from other chr21q22-associated diseases[3–6] and monocyte/macrophage eQTLs[54–58] to determine whether there was a shared genetic basis for these different associations. This was performed using coloc (v.5.2.0)[59] using a posterior probability of H4 (PP.H4.abf) > 0.5 to call co-localization.

Raw H3K27ac ChIP–seq data from primary human immune cells were downloaded from Gene Expression Omnibus (GEO series GSE18927 and GSE96014) and processed as described previously[60] (code provided in the 'Code availability' section).

Processed promoter-capture Hi-C data[61] from 17 primary immune cell types were downloaded from OSF (https://osf.io/u8tzp) and cell type CHiCAGO scores for chr21q22-interacting regions were extracted.

## Monocyte-derived macrophage differentiation

Leukocyte cones from healthy donors were obtained from NHS Blood and Transplant (Cambridge Blood Donor Centre, Colindale Blood Centre or Tooting Blood Donor Centre). Peripheral blood mononuclear cells (PBMCs) were isolated by density centrifugation (Histopaque 1077, Sigma-Aldrich) and monocytes were positively selected using CD14 Microbeads (Miltenyi Biotec). Macrophage differentiation was performed either using conditions that model chronic inflammation (TPP)[16]: 3 days GM-CSF (50 ng ml$^{-1}$, Peprotech) followed by 3 days GM-CSF, TNF (50 ng ml$^{-1}$, Peprotech), PGE$_2$ (1 µg ml$^{-1}$, Sigma-Aldrich) and Pam$_3$CSK4 (1 µg ml$^{-1}$, Invivogen); or, to produce resting (M0) macrophages: 6 days M-CSF (50 ng ml$^{-1}$, Peprotech). All cultures were performed at 37 °C under 5% CO$_2$ in antibiotic-free RPMI1640 medium containing 10% FBS, GlutaMax and MEM non-essential amino acids (all Thermo Fisher Scientific). Cells were detached using Accutase (BioLegend).

## Identifying a model of chronic inflammatory macrophages

Human monocyte/macrophage gene expression data files ($n = 314$) relating to 28 different stimuli with multiple durations of exposure (collectively comprising 67 different activation conditions) were downloaded from the GEO (GSE47189) and quantile normalized. Data from biological replicates were summarized to the median value for every gene. Gene set variation analysis[62] (using the GSVA package in R) was performed to identify the activation condition that most closely resembled CD14$^+$ monocytes/macrophages from active IBD using disease-associated lists of differentially expressed genes[63].

## CRISPR–Cas9 editing of primary human monocytes

gRNA sequences were designed using CRISPick and synthesized by IDT (Supplementary Table 3). Alt-R CRISPR–Cas9 negative control crRNA 1 (IDT) was used as a non-targeting control. Cas9–gRNA ribonucleoproteins were assembled as described previously[60] and nucleofected into $5 × 10^6$ monocytes in 100 µl nucleofection buffer (Human Monocyte Nucleofection Kit, Lonza) using a Nucleofector 2b (Lonza, program Y-001). After nucleofection, monocytes were immediately transferred into 5 ml of prewarmed culture medium in a six-well plate, and differentiated into macrophages under TPP conditions. The editing efficiency was quantified by PCR amplification of the target region in extracted DNA. All primer sequences are provided in Supplementary Table 3. The editing efficiency at the chr21q22 locus was measured by quantification of amplified fragments (2100 Bioanalyzer, Agilent) as previously described[60]. The editing efficiency for individual gRNAs was assessed using the Inference of CRISPR Edits tool[64] (ICE, Synthego).

## PrimeFlow RNA assay

RNA abundance was quantified by PrimeFlow (Thermo Fisher Scientific) in chr21q22-edited and unedited (NTC) cells on days 0, 3, 4, 5 and 6 of TPP differentiation. Target probes specific for *ETS2* (Alexa Fluor 647), *BRWD1* (Alexa Fluor 568) and *PSMG1* (Alexa Fluor 568) were used according to the manufacturer's instructions. Data were collected using FACS Diva software and analysed using FlowJo v10 (BD Biosciences).

## MPRA

Overlapping oligonucleotides containing 114 nucleotides of genomic sequence were designed to tile the region containing chr21q22 candidate SNPs (99% credible set) at 50 bp intervals. Six technical replicates were designed for every genomic sequence, each tagged by a unique 11-nucleotide barcode. Additional oligonucleotides were included to test the expression-modulating effect of every candidate SNP in the 99% credible set. Allelic constructs were designed as described previously[60] and tagged by 30 unique 11-nucleotide barcodes. Positive and negative controls were included as described previously[60]. 170-nucleotide oligonucleotides were synthesized as part of a larger MPRA pool (Twist Biosciences) containing the 16-nucleotide universal primer site ACTGG CCGCTTCACTG, 114-nucleotide variable genomic sequence, KpnI and XbaI restriction sites (TGGACCTCTAGA), an 11-nucleotide barcode and the 17-nucleotide universal primer site AGATCGGAAGAGCGTCG. Cloning into the MPRA vector was performed as described previously[60]. A suitable promoter for the MPRA vector (RSV) was identified by testing promoter activities in TPP macrophages. The MPRA vector library was nucleofected into TPP macrophages (5 µg vector into $5 × 10^6$ cells) in 100 µl nucleofection buffer (Human Macrophage Nucleofection Kit, Lonza) using a Nucleofector 2b (program Y-011). To ensure adequate barcode representation, a minimum of $2 × 10^7$ cells was nucleofected for every donor ($n = 8$). After 24 h, RNA was extracted and sequencing libraries were made from mRNA or DNA input vector as described previously[60]. Libraries were sequenced on the Illumina HiSeq2500 high-output flow-cell (50 bp, single-end reads). Data were demultiplexed and converted to FASTQ files using bcl2fastq and preprocessed as previously described using FastQC[60]. To identify regions of enhancer activity, a paired *t*-test was first performed to identify genomic sequences that enhanced transcription and a sliding-window analysis (300 bp window) was then performed using the les package in R. Expression-modulating variants were identified using QuASAR-MPRA[65], as described previously[60].

## BaalChIP

Publicly available PU.1 ChIP–seq datasets from human macrophages were downloaded from GEO, and BAM files were examined (IGV genome browser) to identify heterozygous samples (that is, files containing both A and G allele reads at chr21:40466570; hg19). Two suitable samples were identified (GSM1681423 and GSM1681429) and used for a Bayesian analysis of allelic imbalances in PU.1 binding (implemented in the BaalChIP package[66] in R) with correction for biases introduced by overdispersion and biases towards the reference allele.

## Allele-specific PU.1 ChIP genotyping

A 100 ml blood sample was taken from five healthy rs2836882 heterozygotes (assessed by Taqman genotyping; Thermo Fisher Scientific). All of the participants provided written informed consent. Ethical approval was provided by the London–Brent Regional Ethics Committee (21/LO/0682). Monocytes were isolated from PBMCs using

CD14 Microbeads (Miltenyi Biotec) and differentiated into inflammatory macrophages using TPP conditions[16]. After differentiation, macrophages were detached and cross-linked for 10 min in fresh medium containing 1% formaldehyde. Cross-linking was quenched with glycine (final concentration 0.125 M, 5 min). Nucleus preparation and shearing were performed as described previously[60] with 10 cycles sonication (30 s on/30 s off, Bioruptor Pico, Diagenode). PU.1 was immunoprecipitated overnight at 4 °C using a polyclonal anti-PU.1 antibody (1:25; Cell Signaling) using the SimpleChIP Plus kit (Cell Signaling). The ratio of rs2836882 alleles in the PU.1-bound DNA was quantified in duplicate by TaqMan genotyping (assay C 2601507_20). A standard curve was generated using fixed ratios of geneblocks containing either the risk or non-risk allele (200-nucleotide genomic sequence centred on rs2836882; Genewiz).

## PU.1 MPRA ChIP–seq

The MPRA vector library was transfected into TPP macrophages from six healthy donors. Assessment of PU.1 binding to SNP alleles was performed as described previously[60], with minimal sonication (to remove contaminants without chromatin shearing). Immunoprecipitation was performed as described above. Sequencing libraries were prepared as for MPRA and sequenced on the MiSeq system (50 bp, single-end reads).

## ATAC–seq analysis

ATAC–seq in *ETS2*-edited and unedited TPP macrophages was performed using the Omni-ATAC protocol[67] with the following modifications: the cell number was increased to 75,000 cells; the cell lysis time was increased to 5 min; the volume of Tn5 transposase in the transposition mixture was doubled; and the duration of the transposition step was extended to 40 min. Amplified libraries were cleaned using AMPure XP beads (Beckman Coulter) and sequenced on the NovaSeq6000 system (100 bp paired-end reads). Data were processed as described previously[68]. Differential ATAC–seq analysis was performed as described previously using edgeR and TMM normalization[69]. Allele-specific ATAC–seq analysis was performed in 16 heterozygous monocyte datasets from healthy controls and patients with ankylosing spondylitis[70] and in 2 deeply sequenced heterozygous TPP macrophage samples. For these analyses, sequencing reads at rs2836882 were extracted from preprocessed data using splitSNP (https://github.com/astatham/splitSNP) (see the 'Code availability' section).

## H3K27ac ChIP–seq

H3K27ac ChIP–seq was performed as described previously[60] using an anti-H3K27ac antibody (1:250, Abcam) or an isotype control (1:500, rabbit IgG, Abcam). Sequencing libraries from TPP macrophages from major and minor allele homozygotes at rs2836882 (identified through the NIHR BioResource, $n = 4$) were sequenced on the HiSeq4000 system (50 bp, single-end reads). Sequencing libraries from *ETS2*-edited and unedited TPP macrophages ($n = 3$) or resting M0 macrophages overexpressing *ETS2* or control mRNA ($n = 3$) were sequenced on the NovaSeq6000 system (100 bp, paired-end reads). Raw data were processed, quality controlled and analysed as described previously using the Burrows-Wheeler Aligner[60]. Unpaired differential ChIP–seq analysis, to compare rs2836882 genotypes, was performed using MEDIPS[71] by dividing the 560 kb region around rs2836882 (chr21:40150000–40710000, hg19) into 5 kb bins. Paired differential ChIP–seq analyses, to assess the effect of perturbing *ETS2* expression on enhancer activity, were performed using edgeR with TMM normalization[69,72] (with donor as covariate). Genome-wide analyses used consensus MACS2 peaks. Superenhancer activity was evaluated using Rank-Ordering of Super-Enhancers (ROSE). Chr21q22-based analyses used the enhancer coordinates that exhibited allele-specific activity (chr21:40465000–40470000, hg19). Code is provided for all data analysis (see the 'Code availability' section).

## Assays of macrophage effector functions

**Flow cytometry.** Expression of myeloid markers was assessed using flow cytometry (BD LSRFortessa X-20) with the following panel: CD11b PE/Dazzle 594 (BioLegend), CD14 evolve605 (Thermo Fisher Scientific), CD16 PerCP (BioLegend), CD68 FITC (BioLegend), Live/Dead Fixable Aqua Dead Cell Stain (Thermo Fisher Scientific) and Fc Receptor Blocking Reagent (Miltenyi). All antibodies were used at a dilution of 1:40; Live/Dead stained was used at 1:400 dilution. Data were collected using FACS Diva and analysed using FlowJo v.10 (BD Biosciences).

**Cytokine quantification.** Supernatants were collected on day 6 of TPP macrophage culture and frozen. Cytokine concentrations were quantified in duplicate by electrochemiluminescence using assays (Meso Scale Diagnostics, DISCOVERY WORKBENCH v.4.0).

**Phagocytosis.** Phagocytosis was assessed using fluorescently labelled Zymosan particles (Green Zymosan, Abcam) according to the manufacturer's instructions. Cells were seeded at $10^5$ cells per well in 96-well round-bottom plates. Cytochalasin D (10 µg ml$^{-1}$, Thermo Fisher Scientific) was used as a negative control. Phagocytosis was quantified by flow cytometry, and a phagocytosis index was calculated (the proportion of positive cells multiplied by their mean fluorescence intensity).

**Extracellular ROS production.** Extracellular ROS production was quantified using the Diogenes Enhanced Superoxide Detection Kit (National Diagnostics) according to the manufacturer's protocol. Cells were seeded at a density of $10^5$ cells per well and prestimulated with PMA (200 ng ml$^{-1}$, Sigma-Aldrich).

**Western blotting.** Western blotting was performed as described previously[73] using the following primary antibodies: mouse anti-gp91phox (1:2,000), mouse anti-p22phox (1:500; both Santa Cruz), rabbit anti-C17ORF62/EROS (1:1,000; Atlas), mouse anti-vinculin (Sigma-Aldrich). Loading controls were run on the same gel. Secondary antibodies were as follows: goat anti-rabbit IgG-horseradish or goat anti-mouse IgG-horseradish peroxidase (both 1:10,000; Jackson Immuno). Chemiluminescence was recorded on the ChemiDoc Touch imager (Bio-Rad) after incubation of the membrane with ECL (Thermo Fisher Scientific) or SuperSignal West Pico PLUS (Thermo Fisher Scientific) reagent. Densitometry analysis was performed using ImageJ.

## RNA-seq analysis

RNA was isolated from macrophage lysates (AllPrep DNA/RNA Micro Kit, Qiagen) and sequencing libraries were prepared from 10 ng RNA using the SMARTer Stranded Total RNA-Seq Kit v2 Pico Input Mammalian (Takara) according to the manufacturer's instructions. Libraries were sequenced on either the NextSeq 2000 (50 bp paired-end reads: CRISPR, roxadustat and PD-0325901 experiments) or NovaSeq 6000 (100 bp paired-end reads: overexpression experiments) system and preprocessed using MultiQC. Reads were trimmed using Trim Galore (Phred score 24) and filtered to remove reads <20 bp. Ribosomal reads (mapping to human ribosomal DNA complete repeating unit; GenBank: U13369 .1) were removed using BBSplit (https://sourceforge.net/projects/bbmap/). Reads were aligned to the human genome (hg38) using HISAT2 (ref. 74) and converted to BAM files, sorted and indexed using SAMtools[75]. Gene read counts were obtained using the featureCounts program[76] from Rsubread using the GTF annotation file for GRCh38 (v.102). Differential expression analysis was performed in R using limma[77] with voom transformation and including donor as a covariate. Differential expression results are shown in Supplementary Tables 1 and 2.

## GSEA

GSEA was performed using fGSEA[78] in R with differentially expressed gene lists ranked by $t$-statistic. Gene sets were obtained from GO Biological Pathways (MSigDB), experimentally derived based on differential expression analysis or sourced from published literature[31,42,70,79–86]. Specific details of disease macrophage signatures (Fig. 3f) are provided as source data. GO pathways shown in Figs. 2–5 are as follows: GO:0002274, GO:0042116, GO:0097529, GO:0006909, GO:0071706, GO:0032732, GO:0032755, GO:0032757, GO:2000379, GO:0009060, GO:0006119 and GO:0045649. Statistical significance was calculated using the adaptive multilevel split Monte Carlo method.

## IBD BioResource recall-by-genotype study

IBD patients who were rs2836882 major or minor allele homozygotes ($n = 11$ of each) were identified through the NIHR IBD BioResource. Patients were matched for age, sex, treatment and disease activity, and all provided written informed consent. Ethical approval was provided by the London–Brent Regional Ethics Committee (21/LO/0682). A 50 ml blood sample was taken from all patients and M0 monocyte-derived macrophages were generated as described. After 6 days, cells were collected, lysed and RNA was extracted. Quantitative PCR analysis of a panel of ETS2-regulated genes was performed in triplicate after reverse transcription (SuperScript IV VILO, Thermo Fisher Scientific) using the Quantifast SYBR Green PCR kit (Qiagen) on the Roche Light-Cycler 480. Primer sequences are provided in Supplementary Table 3 and *PPIA* and *RPLP0* were used as housekeeping genes. Expression values for each gene ($2^{\Delta c_T}$) were scaled to a minimum 0 and maximum 1 to enable intergene comparison.

## In vitro transcription

The cDNA sequence for *ETS2* (NCBI Reference Sequence Database NM005239.5) preceded by a Kozak sequence was synthesized and cloned into a TOPO vector. This was linearized and a PCR amplicon generated, adding a T7 promoter and an AG initiation sequence (Phusion, NEB). A reverse complement (control) amplicon was also generated. These amplicons were used as templates for in vitro transcription using the HiScribe T7 mRNA Kit with CleanCap Reagent AG kit (NEB) according to the manufacturer's instructions, but with substitution of N1-methyl-pseudouridine for uridine and methylcytidine for cytidine (both Stratech) to minimize non-specific cellular activation by the transfected mRNA. mRNA was purified using the MEGAclear Kit (Thermo Fisher Scientific) and polyadenylated using an *Escherichia coli* poly(A) polymerase (NEB) before further clean-up (MEGAclear), quantification and analysis of the product size (NorthernMax-Gly gel, Thermo Fisher Scientific). For optimizing overexpression conditions, *GFP* mRNA was produced using the same method. All primer sequences are provided in Supplementary Table 3.

## mRNA overexpression

Lipofectamine MessengerMAX (Thermo Fisher Scientific) was diluted in Opti-MEM (1:75 v/v), vortexed and incubated at room temperature for 10 min. IVT mRNA was then diluted in a fixed volume of Opti-MEM (112.5 µl per transfection), mixed with an equal volume of diluted Lipofectamine MessengerMAX and incubated for a further 5 min at room temperature. The transfection mix was then added dropwise to $2.5 \times 10^6$ M0 macrophages (precultured for 6 days in a six-well plate in antibiotic-free RPMI1640 macrophage medium containing M-CSF (50 ng ml$^{-1}$, Peprotech), with medium change on day 3). For GFP overexpression, cells were detached using Accutase 18 h after transfection and GFP expression was measured using flow cytometry. For *ETS2*/control overexpression, either 250 ng or 500 ng mRNA was transfected and low-dose LPS (0.5 ng ml$^{-1}$) was added 18 h after transfection, and cells were detached using Accutase 6 h later. Representative *ETS2* expression in untransfected macrophages was obtained from

previous data (GSE193336). Differential H3K27ac ChIP–seq analysis in *ETS2*-overexpressing macrophages was performed using 500 ng RNA transfection (see the 'Code availability' section).

## PRS

Plink1.9 (https://www.cog-genomics.org/plink/1.9/) was used to calculate a polygenic risk score (PRS) for patients in the IBD BioResource using 22 ETS2-regulated IBD-associated SNPs ($\beta$ coefficients from a previous study[3]). Linear regression was used to compare PRSs with age at diagnosis, and logistic regression to estimate the effect of PRSs on IBD subphenotypes, including anti-TNF primary non-response (PNR), CD behaviour (B1 versus B2/B3), perianal disease and surgery. For variables with more than two levels (for example, CD location or UC location), ANOVA was used to investigate the relationship with PRS. For analyses of age at diagnosis, anti-TNF response and surgery, IBD diagnosis was included as a covariate.

## SNPsea

Pathway analysis of 241 IBD-associated GWAS hits[3] was performed using SNPsea v.1.0.4 (ref. 34). In brief, linkage intervals were defined for every lead SNP based on the furthest correlated SNPs ($r^2 > 0.5$ in 1000 Genomes, European population) and were extended to the nearest recombination hotspots with recombination rate > 3 cM per Mb. If no genes were present in this region, the linkage interval was extended up- and downstream by 500 kb (as long-range regulatory interactions usually occur within 1 Mb). Genes within linkage intervals were tested for enrichment within 7,660 pathways, comprising 7,658 GO Biological Pathways and two lists of ETS2-regulated genes (either those significantly downregulated after *ETS2* disruption with gRNA1 or those significantly upregulated after *ETS2* overexpression, based on a consensus list obtained from differential expression analysis including all samples and using donor and mRNA quantity as covariates). The analysis was performed using a single score mode: assuming that only one gene per linkage interval is associated with the pathway. A null distribution of scores for each pathway was performed by sampling identically sized random SNP sets matched on the number of linked genes (5,000,000 iterations). A permutation $P$ value was calculated by comparing the score of the IBD-associated gene list with the null scores. An enrichment statistic was calculated using a standardized effect size for the IBD-associated score compared to the mean and s.e.m. of the null scores. Gene sets relating to the following IBD-associated pathways were extracted for comparison: NOD2 signalling (GO:0032495), integrin signalling (GO:0033627, GO:0033622), TNF signalling (GO:0033209, GO:0034612), intestinal epithelium (GO:0060729, GO:0030277), Th17 cells (GO:0072539, GO:0072538, GO:2000318), T cell activation (GO:0046631, GO:0002827), IL-10 signalling (GO:0032613, GO:0032733) and autophagy (GO:0061919, GO:0010506, GO:0010508, GO:1905037, GO:0010507). SNPs associated with PSC[5,87], ankylosing spondylitis[4,87], Takayasu arteritis[6,88,89] and schizophrenia[90] (as a negative control) were collated from the indicated studies and tested for enrichment in ETS2-regulated gene lists.

## ETS2 co-expression

Genes co-expressed with ETS2 across 67 human monocyte/macrophage activation conditions (normalized data from GSE47189) were identified using the rcorr function in the Hmisc package in R.

## $^{13}$C-glucose GC–MS

*ETS2*-edited or unedited TPP macrophages were generated in triplicate for each donor and on day 6, the medium was removed, cells were washed with PBS, and new medium with labelled glucose was added. Labelled medium was as follows: RPMI1640 medium, no glucose (Thermo Fisher Scientific), 10% FBS (Thermo Fisher Scientific), GlutaMax (Thermo Fisher Scientific), $^{13}$C-labelled glucose (Cambridge Isotope Laboratories). After 24 h, a timepoint selected from

a time-course to establish steady-state conditions, the supernatants were snap-frozen and macrophages were detached by scraping. Macrophages were washed three times with ice-cold PBS, counted, resuspended in 600 µl ice-cold chloroform:methanol (2:1, v/v) and sonicated in a waterbath (3 times for 8 min). All of the extraction steps were performed at 4 °C as previously described[91]. The samples were analysed on the Agilent 7890B-7000C GC–MS system. Spitless injection (injection temperature of 270 °C) onto a DB-5MS (Agilent) was used, using helium as the carrier gas, in electron ionization mode. The initial oven temperature was 70 °C (2 min), followed by temperature gradients to 295 °C at 12.5 °C per min and to 320 °C at 25 °C per min (held for 3 min). The scan range was $m/z$ 50–550. Data analysis was performed using in-house software MANIC (v.3.0), based on the software package GAVIN[92]. Label incorporation was calculated by subtracting the natural abundance of stable isotopes from the observed amounts. Total metabolite abundance was normalized to the internal standard (scyllo-inositol[91]).

## Roxadustat in TPP macrophages

*ETS2*-edited or unedited TPP macrophages were generated as described previously. On day 5 of culture, cells were detached (Accutase) and replated at a density of $10^5$ cells per well in 96-well round-bottom plates in TPP medium containing roxadustat (FG-4592, 30 µM). After 12 h, cells were collected for functional assays and RNA-seq as described.

## CUT&RUN

Precultured TPP macrophages were collected and processed immediately using the CUT&RUN Assay kit (Cell Signaling) according to the manufacturer's instructions but omitting the use of ConA-coated beads. In brief, $5 \times 10^5$ cells per reaction were pelleted, washed and resuspended in antibody binding buffer. Cells were incubated with antibodies: anti-ETS2 (1:100, Thermo Fisher Scientific) or IgG control (1:20, Cell Signaling) for 2 h at 4 °C. After washing in digitonin buffer, cells were incubated with pA/G-MNase for 1 h at 4 °C. Cells were washed twice in digitonin buffer, resuspended in the same buffer and cooled for 5 min on ice. Calcium chloride was added to activate pA/G-MNase digestion (30 min, 4 °C) before the reaction was stopped and cells incubated at 37 °C for 10 min to release cleaved chromatin fragments. DNA was extracted from the supernatants using spin columns (Cell Signaling). Library preparation was performed using the NEBNext Ultra II DNA Library Prep Kit according to a protocol available at protocols. io (https://doi.org/10.17504/protocols.io.bagaibse). Size selection was performed using AMPure XP beads (Beckman Coulter) and the fragment size was assessed using the Agilent 2100 Bioanalyzer (High Sensitivity DNA kit). Indexed libraries were sequenced on the NovaSeq 6000 system (100 bp paired-end reads). Raw data were analysed using guidelines from the Henikoff laboratory[93]. In brief, paired-end reads were trimmed using Trim Galore and aligned to the human genome (GRCh37/hg19) using Bowtie2. BAM files were sorted, merged (technical and, where indicated, biological replicates), resorted and indexed using SAMtools. Picard was used to mark unmapped reads and SAMtools to remove these, re-sort and re-index. Bigwig files were created using the deepTools bamCoverage function. Processed data were initially analysed using the nf-core CUT&RUN pipeline v.3.0, using CPM normalization and default MACS2 parameters for peak calling. This analysis yielded acceptable quality metrics (including an average FRiP score of 0.23) but there was a high number of peaks with low fold enrichment (<4) over the control. More stringent parameters were therefore applied for peak calling (--qvalue 0.05 -f BAMPE --keep-dup all -B --nomodel) and we applied an irreproducible discovery rate (IDR; cut-off 0.001) to identify consistent peaks between replicates, implemented in the idr package in R (see the 'Code availability' section). Enrichment of binding motifs for ETS2 and other transcription factors expressed in TPP macrophages (cpm > 0.5) within consensus IDR peaks was calculated using TFmotifView[94] using global genomic controls.

The overlap between consensus IDR peaks and the core promoter (−250bp to +35 bp from the transcription start site) and/or putative *cis*-regulatory elements of ETS2-regulated genes was assessed using differentially expressed gene lists after *ETS2* disruption (gRNA1) or *ETS2* overexpression (based on a consensus across mRNA doses, as described earlier). Putative *cis*-regulatory elements were defined as shared interactions (CHiCAGO score > 5) in monocyte and M0 and M1 macrophage samples from publicly available promoter-capture Hi-C data[61]. Predicted ETS2- and PU.1-binding sites were identified at the rs2836882 locus (chr21:40466150–40467450) using CisBP[95] (database 2.0, PWMs log odds motif model, default settings).

## Intestinal scRNA-seq

Raw count data from colonic immune cells[41] (including healthy controls and Crohn's disease) were downloaded from the Single Cell Portal (https://singlecell.broadinstitute.org/single_cell). Myeloid cell data were extracted for further analysis using the cell annotation provided. Raw data were preprocessed, normalized and variance-stabilized using Seurat (v.4)[96]. PCA and UMAP clustering was performed and clusters annotated using established markers and/or previous literature. Marker genes were identified using the FindAllMarkers function. Modular expression of ETS2-regulated genes (downregulated after *ETS2* editing, gRNA1) was measured using the AddModuleScore function.

## Spatial transcriptomics

Formalin-fixed paraffin-embedded sections (thickness, 5 µm) were cut from two PSC liver explants and two controls (healthy liver adjacent to tumour metastases), baked overnight at 60 °C and prepared for CosMx according to manufacturer's instructions using 15 min target retrieval and 30 min protease digestion. Tissue samples were obtained through Tissue Access for Patient Benefit (TAP-B, part of the UCL-RFH Biobank) under research ethics approval: 16/WA/0289 (Wales Research Ethics Committee 4). One case and one control were included on each slide. The Human Universal Cell Characterization core panel (960 genes) was used, supplemented with 8 additional genes to improve identification of cells of interest: *CD1D*, *EREG*, *ETS2*, *FCN1*, *GOS2*, *LYVE1*, *MAP2K1*, *MT1G*. Segmentation was performed using the CosMx Human Universal Cell Segmentation Kit (RNA), Human IO PanCK/CD45 Kit (RNA) and Human CD68 Marker, Ch5 (RNA). Fields of view (FOVs) were tiled across all available regions (221 control, 378 PSC) and cyclic fluorescence in situ hybridization was performed using the CosMx SMI (Nanostring) system. Data were preprocessed on the AtoMx Spatial Informatics Platform, with images segmented to obtain cell boundaries, transcripts assigned to single cells, and a transcript by cell count matrix was obtained[97]. Expression matrices, transcript coordinates, polygon coordinates, FOV coordinates and cell metadata were exported, and quality control, normalization and cell-typing were performed using InSituType[98]— an R package developed to extract all the information available in every cell's expression profile. A semi-supervised strategy was used to phenotype cells, incorporating the Liver Human Cell Atlas reference matrix. Spatial analysis of macrophage phenotypes was performed according to proximity from cholangiocytes (anchor cell type). Radius and nearest-neighbour analyses were performed using PhenoptR (https://akoyabio.github.io/phenoptr/) with macrophage distribution from cholangiocytes binned in 100 µm increments up to 500 µm. Nearest-neighbour analysis was performed to determine the distance from cholangiocytes to the nearest inflammatory and non-inflammatory macrophage and vice versa.

To generate overlay images, raw transcript and image (morphology 2D) data were exported from AtoMx. Overlays of selected ETS2-target genes (*CXCL8*, *S100A9*, *CCL2*, *CCL5*) and fluorescent morphology markers were generated using napari (v.0.4.17, https://napari.org/stable/index.html) on representative FOVs: FOV287 (PSC with involved duct), FOV294 (PSC background liver) and FOV55 (healthy liver).

## Chr21q22 disease datasets

Publicly available raw RNA-seq data from the affected tissues of chr21q22-associated diseases (and controls from the same experiment) were downloaded from the GEO: IBD macrophages (GSE123141), PSC liver (GSE159676), ankylosing spondylitis synovium (GSE41038). Reads were trimmed, filtered and aligned as described earlier. For each disease dataset, a ranked list of genes was obtained by differential expression analysis between cases and controls using limma with voom transformation. For IBD macrophages, only IBD samples with active disease were included. fGSEA using ETS2-regulated gene lists was performed as described.

## LINCS signatures

A total of 31,027 lists of downregulated genes after exposure of a cell line to a small molecule was obtained from the NIH LINCS database[7] (downloaded in January 2021). These were used as gene sets for fGSEA (as described) with a ranked list of genes obtained by differential expression analysis between *ETS2*-edited and unedited TPP macrophages (gRNA1) using limma with voom transformation and donor as a covariate. Drug classes for gene sets with FDR-adjusted *P* < 0.05 were manually assigned on the basis of known mechanisms of action.

## MEK inhibition in TPP macrophages

TPP macrophages were generated as described previously. On day 4 of culture, PD-0325901 (0.5 µM, Sigma-Aldrich) or vehicle (DMSO) was added. Cells were collected on day 6 and RNA was extracted and sequenced as described.

## Colonic biopsy explant culture

During colonoscopy, intestinal mucosal biopsies (6 per donor) were collected from ten patients with IBD (seven patients with ulcerative colitis, three patients with Crohn's disease). All had endoscopically active disease and were not receiving immunosuppressive or biologic therapies. All biopsies were collected from a single inflamed site. All patients provided written informed consent. Ethical approval was provided by the London–Brent Regional Ethics Committee (21/LO/0682). Biopsies were collected into Opti-MEM and, within 1 h, were weighed and placed in pairs onto a Transwell insert (Thermo Fisher Scientific), designed to create an air–liquid interface[99], in a 24-well plate. Each well contained 1 ml medium and was supplemented with either DMSO (vehicle control), PD-0325901 (0.5 µM) or infliximab (10 µg ml⁻¹; MSD). Medium was as follows: Opti-MEM I (Gibco), GlutaMax (Thermo Fisher Scientific), 10% FBS (Thermo Fisher Scientific), MEM non-essential amino acids (Thermo Fisher Scientific), 1% sodium pyruvate (Thermo Fisher Scientific), 1% penicillin–streptomycin (Thermo Fisher Scientific) and 50 µg ml⁻¹ gentamicin (Merck). After 18 h, the supernatants and biopsies were snap-frozen. The supernatant cytokine concentrations were quantified using the LEGENDplex Human Inflammation Panel (BioLegend). RNA was extracted from biopsies and libraries were prepared as described earlier (*n* = 9, RNA from one donor was too degraded). Sequencing was performed on the NovaSeq 6000 system (100 bp paired-end reads). Data were processed as described earlier and GSVA was performed for ETS2-regulated genes and biopsy-derived signatures of IBD-associated inflammation[46].

## Chr21q22 genotypes in archaic humans

Using publicly available genomes from seven Neanderthal individuals[100–103], one Denisovan individual[104], and one Neanderthal and Denisovan F1 individual[105], genotypes were called at the disease-associated chr21q22 candidate SNPs from the respective BAM files using bcftools mpileup with base and mapping quality options -q 20 -Q 20 -C 50 and using bcftools call -m -C alleles, specifying the two alleles expected at each site in a targets file (-T option). From the resulting .vcf file, the number of reads supporting the reference and alternative alleles was extracted and stored in the 'DP4' field.

## Inference of Relate genealogy at rs2836882

Genome-wide genealogies, previously inferred for samples of the Simons Genome Diversity Project[106] using Relate[107,108] (https://reichdata. hms.harvard.edu/pub/datasets/sgdp/), were downloaded from https:// www.dropbox.com/sh/2gjyxe3kqzh932o/AAAQcipCHnySgEB873t9EQ jNa?dl=0. Using the inferred genealogies, the genealogy at rs2836882 (chr21:40466570) was plotted using the TreeView module of Relate.

## Data presentation

The following R packages were used to create figures: GenomicRanges[109], EnhancedVolcano[110], ggplot2 (ref. 111), gplots[112], karyoploteR[113].

## Statistical methodology

Statistical methods used in MPRA analysis, fGSEA and SNPsea are described above. For other analyses, comparison of continuous variables between two groups was performed using Wilcoxon matched-pairs tests (paired) or Mann–Whitney *U*-tests (unpaired) for nonparametric data or a *t*-tests for parametric data. Comparison against a hypothetical value was performed using Wilcoxon signed-rank tests for nonparametric data or one-sample *t*-tests for parametric data. A Shapiro–Wilk test was used to confirm normality. Two-sided tests were used as standard unless a specific hypothesis was being tested. Sample sizes are provided in the main text and figure captions.

## Reporting summary

Further information on research design is available in the Nature Portfolio Reporting Summary linked to this article.

## Data availability

The datasets produced in this study are accessible at the following repositories: MPRA (GEO: GSE229472), RNA-seq data of *ETS2* or chr21q22-edited TPP macrophages (EGA: EGAD00001011338), RNA-seq data of *ETS2* overexpression (EGA: EGAD00001011341), RNA-seq data of MEK-inhibitor-treated TPP macrophages (EGA: EGAD00001011337), H3K27ac ChIP–seq data in TPP macrophages (EGA: EGAD00001011351), ATAC–seq and H3K27ac ChIP–seq data in *ETS2*-overexpressing or -edited macrophages (EGA: EGAD50000000154), ETS2 CUT&RUN data (EGA: EGAD00001011349), biopsy RNA-seq data (EGA: EGAD00001011333). MetaboLights: Metabolomics (MTBLS7665). The counts table for CosMx is provided at Zenodo (https://zenodo. org/records/10707942)[114]. The phenotype and genotype data used for the PRS analysis are available on application to the IBD Bioresource (https://www.ibdbioresource.nihr.ac.uk/). Source data are provided with this paper.

## Code availability

Code to reproduce analyses are available at GitHub (https://github. com/JamesLeeLab/chr21q22_manuscript; https://github.com/ chr1swallace/ibd-ets2-analysis; https://github.com/qzhang314/PRS_ IBD_subpheno)[114]. Final code is deposited at Zenodo (https://zenodo. org/records/10707942).

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

**Acknowledgements** We thank the members of the Lee laboratory, K. Slowikowski and A. Kaser for discussions; G. Stockinger, C. Vinuesa, C. Swanton, R. Patani and C. Reis e Sousa for reading the manuscript; C. Cheshire and the staff at the Francis Crick Institute Advanced Sequencing Facility and Flow Cytometry STP for technical support; L. Lucaciu for help with patient recruitment; RFH PITU nurses for assistance obtaining infliximab; the members of Tissue Access for Patient Benefit (TAP-B) for providing liver samples; NIHR BioResource volunteers for their participation; and the NIHR BioResource centres, NHS Blood and Transplant, and NHS staff for their contributions. This work was supported by Crohn's and Colitis UK (M2018-3), the Wellcome Trust (Sir Henry Wellcome Fellowship to L.S., 220457/Z/20/Z; Investigator Award to P.S., 217223/Z/19/Z; Senior Fellowship to C.W., WT220788; Clinical Research Career Development Fellowship to M.Z.C., 222056/Z/20/Z; Wellcome-Beit Prize Clinical Career Development Fellowship to D.C.T., 206617/A/17/A; and Intermediate Clinical Fellowship to J.C.L., 105920/Z/14/Z), and the Francis Crick Institute, which receives its core funding from Cancer Research UK (CC2219, FC001595), the UK Medical Research Council (CC2219, FC001595) and the Wellcome Trust (CC2219, FC001595). L.M.H. is supported by the Charité–Universitätsmedizin Berlin and the Berlin Institute of Health Charité (Clinician-Scientist Program); A.J.C. by the Medical Research Council (MR/V029711/1); A.L. by a Lord Kelvin/Adam Smith Leadership Grant; A.H.S. by the National Institute of Arthritis and Musculoskeletal and Skin Diseases (NIH, R01:AR070148); N.B.J. by Cancer Research UK (C55370/A25813); T.Z. by the Chinese Scholarship Council (202308060128); A.Q. by the NIHR UCLH/UCL BRC; J.C.K. by Versus Arthritis (program grant, 20773), Janssen Oxford Translational fellowships and NIHR Oxford BRC; P.S. by the European Molecular Biology Organisation, the Vallee Foundation and the European Research Council (852558); C.W. by the Medical Research Council (MC UU 00002/4), GSK, MSD and the NIHR Cambridge BRC (BRC-1215-20014); and D.C.T. by the Sidharth Burman endowment. J.C.L. is a Lister Institute Prize Fellow. The funders had no role in study design, data collection and analysis, decision to publish or preparation of the manuscript. Experimental schematics in Figs. 1d, 2a and 3a and Extended Data Figs. 3a, 4a,b,e and 7g,h were created using BioRender. For the purpose of open access, the authors have applied a CC BY public copyright licence to any author accepted manuscript version arising from this submission.

**Author contributions** Conceptualization: J.I.M., N.B.J., P.S., M.Z.C., C.W., D.C.T. and J.C.L. Methodology: C.T.S., C.B., M.S.d.S., M.G., E.N., J.I.M., C.W. and J.C.L. Software: C.B., M.S.d.S., Q.Z., A.J.C., A.L., T.Z., C.S.W., L.S., J.I.M., N.B.J., P.S., C.W. and J.C.L. Investigation: C.T.S., C.B., L.M.H., T.T.-S., A.P.P., I.P., M.S.d.S., L.O.R., A.C.B., E.C.P., W.E., M.G., C.D.M. and J.C.L. Resources: C.T.S., C.B., C.P.-J., A.H., F.S., A.Q., D.T., A.P.R., C.D.M. and J.C.L. Formal analysis: C.T.S., C.B., M.S.d.S., Q.Z., A.J.C., A.L., T.Z., F.N.N., L.S., P.D., C.W. and J.C.L. Writing—original draft: C.T.S., C.B. and J.C.L. Writing—review and editing: all of the authors. Funding acquisition: J.C.L. Supervision: J.C.K., J.I.M., N.B.J., P.S., C.W., D.C.T. and J.C.L.

**Funding** Open Access funding provided by The Francis Crick Institute.

**Competing interests** C.T.S., C.B. and J.C.L. are listed as co-inventors on a patent application related to this work. C.W. holds a part-time position at GSK. GSK had no role in the design or conduct of this study. F.N.N. and P.D. are employees and shareholders of NanoString Technologies. NanoString had no role in the design or conduct of this study. The other authors declare no competing interests.

**Additional information**
**Correspondence and requests for materials** should be addressed to J. C. Lee.

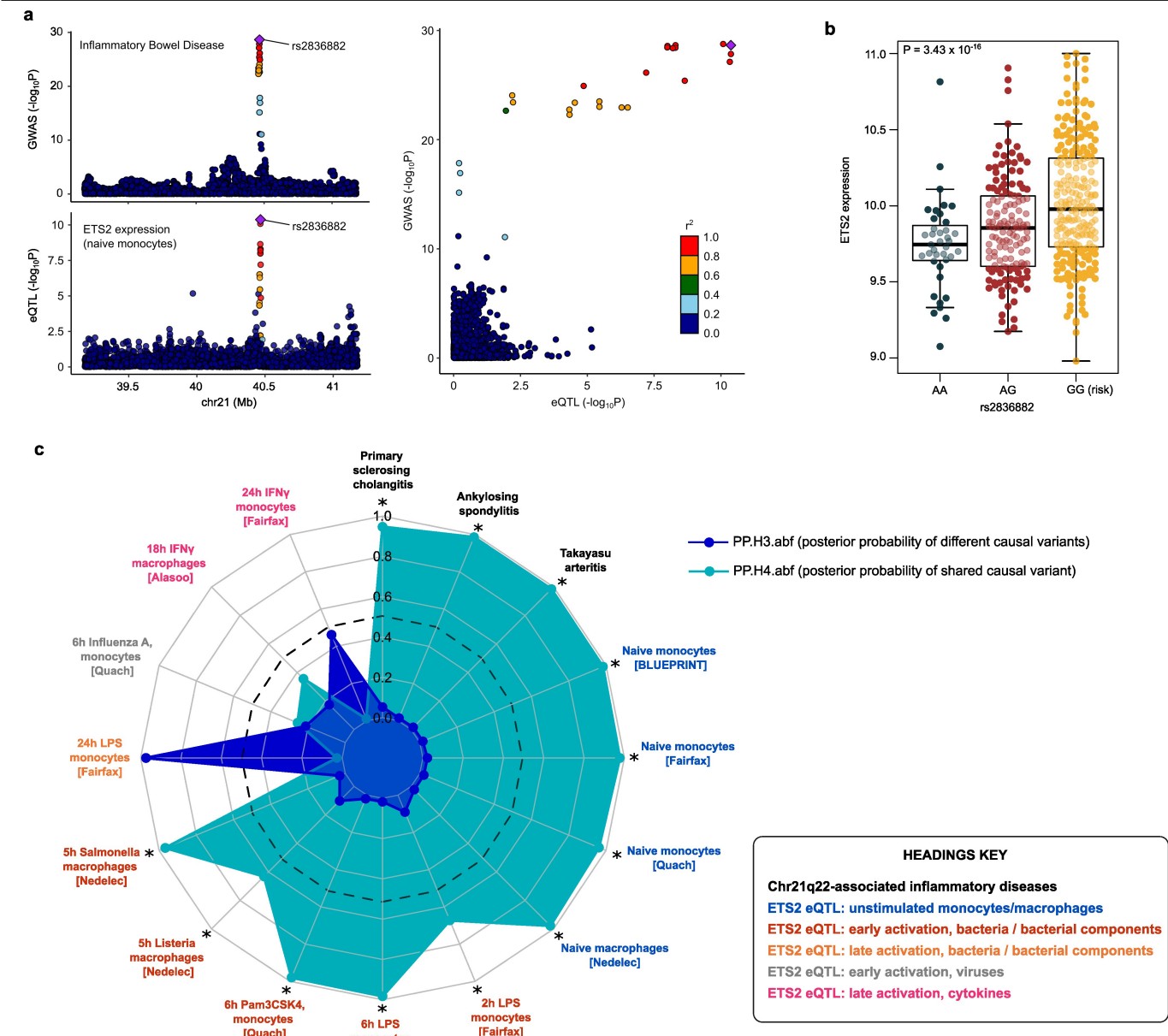

**Extended Data Fig. 1 | Colocalisation between genetic associations at chr21q22. a.** Example comparison of genetic associations at chr21q22: IBD and *ETS2* eQTL in unstimulated monocytes. Plot adapted from locuscomparer. **b.** Tukey box-and-whisker plot depicting *ETS2* expression stratified by rs2836882 genotype in unstimulated monocytes (AA, n = 39; AG, n = 142; GG, n = 233)[54]. *P*-value is as reported in index study. **c.** Radar plot of representative colocalization results for the indicated genetic associations compared to IBD.

Posterior probability of independent causal variants, PP.H3, dark blue; posterior probability of shared causal variant, PP.H4, light blue. PP.H4 > 0.5 was used to call colocalisation (denoted by dashed line). Labels are coloured according to class of data (indicated in the key). Asterisks denote colocalisation. Data sources are: IBD[3], PSC[5], AS[4], Takayasu Arteritis[6], BLUEPRINT[56], Fairfax[54], Quach[55], Nedelec[57], Alasoo[58].

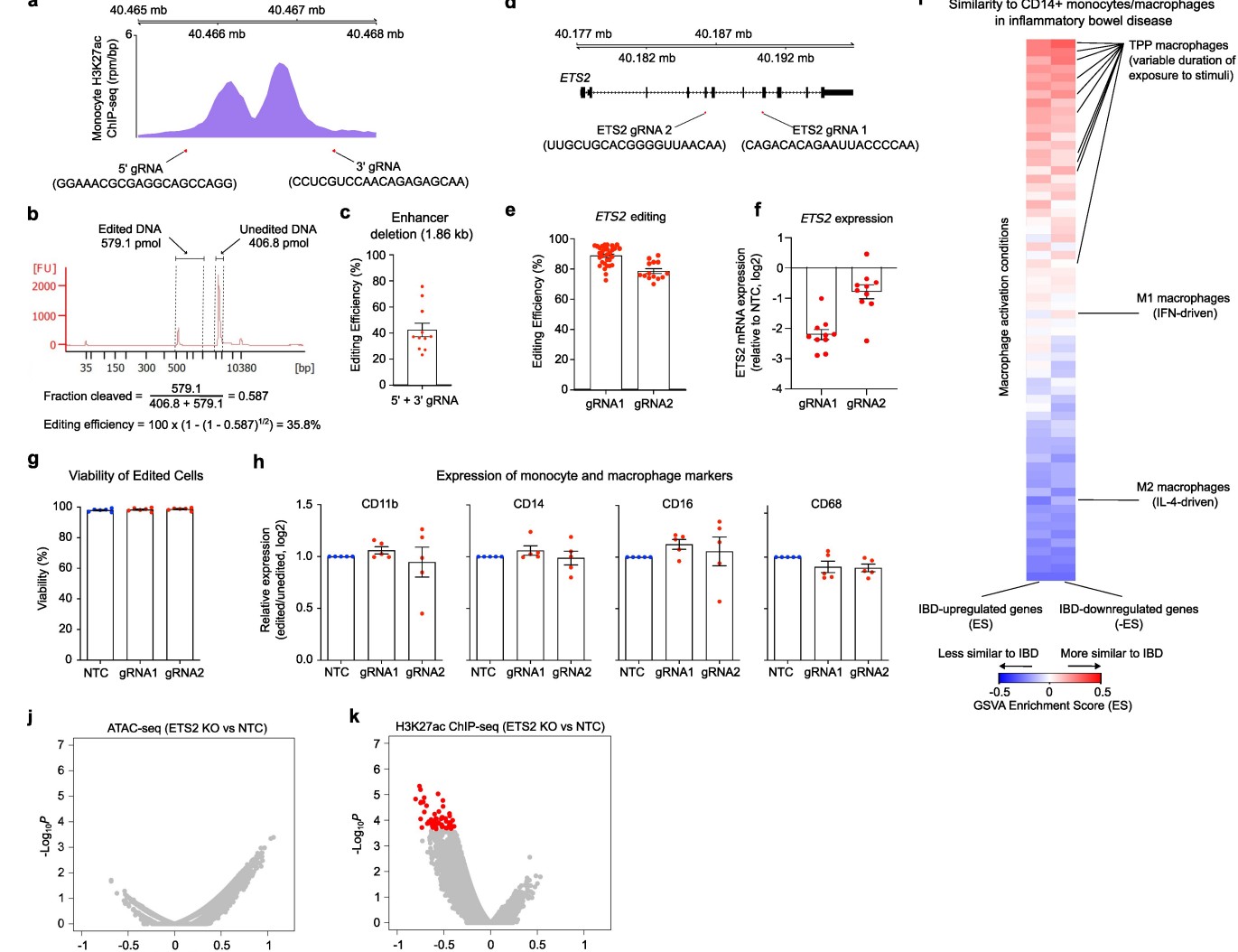

**Extended Data Fig. 2 | CRISPR-Cas9 editing of the chr21q22 locus and ETS2 in monocytes. a**. Cas9 gRNAs were designed to flank the chr21q22 enhancer region at the indicated sites. **b**. Representative bioanalyzer trace of PCR-amplified target region following monocyte CRISPR/Cas9 editing with an equimolar mix of RNPs containing 5′ and 3′ chr21q22 gRNAs. Example editing efficiency calculation shown. **c**. Editing efficiency at the chr21q22 locus. Mean enhancer deletion: 42.4% (n = 11). **d**. Location and sequence of gRNAs used to disrupt *ETS2*. **e**. *ETS2* editing efficiency. gRNA1 (mean), 89.7% (n = 31); gRNA2 (mean), 78.6% (n = 14). **f**. *ETS2* expression (relative to NTC) following CRISPR/Cas9 editing, measured by qPCR (housekeeping gene *PPIA*; equivalent results with other housekeeping genes; n = 10). **g**. Viability following

monocyte nucleofection with Cas9 RNPs and macrophage differentiation. Mean values: NTC, 97.9%; gRNA1: 98.3%; gRNA2, 98.6% (n = 6). **h**. Expression of myeloid lineage markers following *ETS2* editing and TPP differentiation (n = 5). Gating strategy shown in Supplementary Information Fig. 2. **i**. GSVA enrichment scores for 67 different monocyte/macrophage activation conditions to identify stimuli that phenocopy CD14+ monocytes/macrophages from IBD patients. **j**. Chromatin accessibility in ETS2-edited versus unedited inflammatory macrophages (n = 3). **k**. Enhancer activity (H3K27ac) in ETS2-edited versus unedited inflammatory macrophages (n = 3). P values calculated using edgeR (two-sided) in **j**, **k**. Red points denote adjusted *P*-value ($P_{adj}$) < 0.1, grey points NS. Error bars are mean±SEM in **c**, **e-h**. * P < 0.05. NTC: non-targeting control.

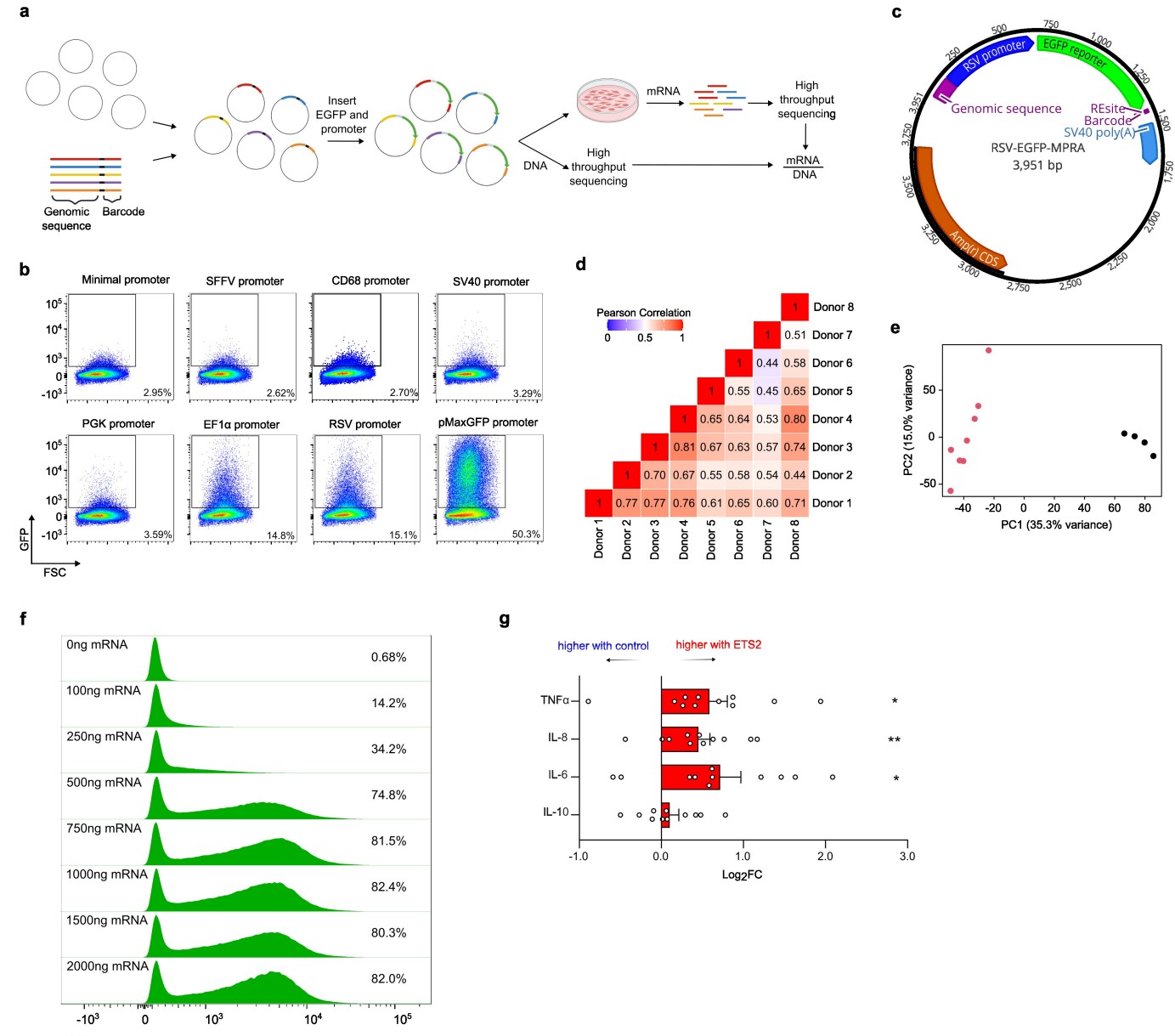

**Extended Data Fig. 3 | Optimization of MPRA and mRNA overexpression in primary human macrophages. a**. Schematic of MPRA. A library of oligonucleotides (each containing a genomic sequence and unique barcode, separated by restriction enzyme sites) is cloned into a pGL4.10 M cloning vector. A promoter and reporter gene are inserted using directional cloning. The resulting plasmids are transfected into primary human macrophages (TPP) and RNA is extracted after 24 h. Barcode abundance in cellular mRNA and input DNA library are quantified by high-throughput sequencing, and mRNA barcode counts are normalized to corresponding counts in DNA library to assess expression-modulating activity. **b**. Identification of suitable promoters for MPRA in TPP macrophages. TPP macrophages were transfected with reporter vectors, each with GFP expression under the control of a different promoter. GFP expression was quantified by flow cytometry after 24 h.

**c**. Adapted MPRA vector for use in primary human macrophages, containing RSV promoter. **d**. Heatmap showing pairwise correlation of expression-modulating activity of all constructs between donors. **e**. Principal component analysis of element counts (sum of barcodes tagging same genomic sequence) in mRNA from TPP macrophages (n = 8 donors; red) and four replicates of DNA vector (black). **f**. Primary human macrophages (M0) were transfected with different quantities of GFP mRNA using Lipofectamine MessengerMAX. GFP expression was quantified by flow cytometry 18 h after transfection. **g**. Cytokine secretion following *ETS2* overexpression. Plot shows relative cytokine concentrations in macrophage supernatants (*ETS2* relative to control) following transfection with 500 ng mRNA (n = 11). Error bars are mean±SEM. One-sample *t*-test (two-tailed) * P < 0.05, ** P < 0.01. The diagram in **a** was created using BioRender.

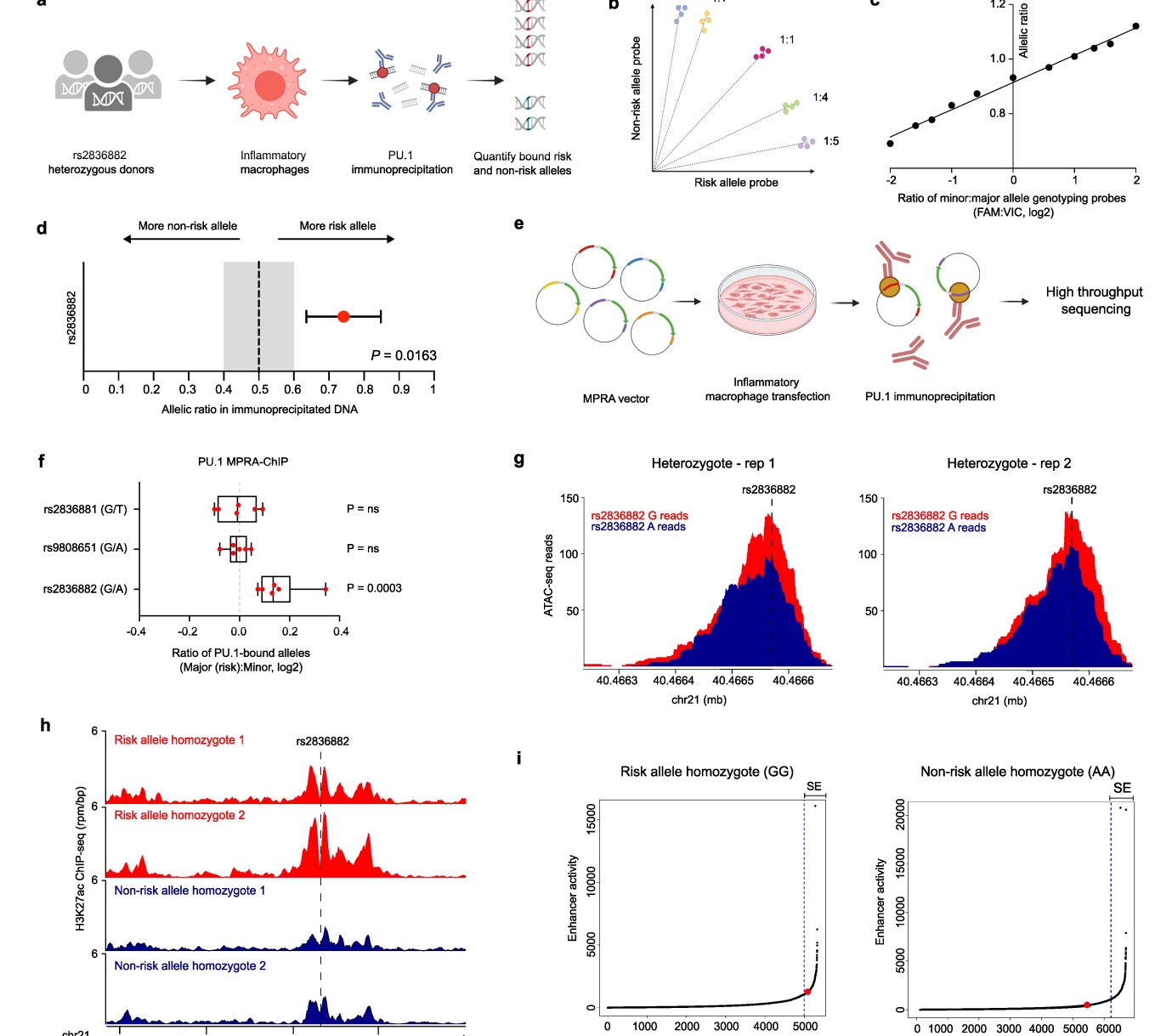

**Extended Data Fig. 4 | Molecular effects of allelic variation at rs2836882.**
**a**. Schematic of PU.1 ChIP-genotyping assay to assess allele-specific PU.1 binding at rs2836882 in human macrophages. **b**. Schematic of standard curve generation by TaqMan genotyping various pre-defined ratios of risk and non-risk containing DNA sequences. **c**. Standard curve generated using different allelic ratios of 200-nt DNA geneblocks centred on either the major (risk) or minor (non-risk) rs2836882 allele. **d**. Allele-specific PU.1 binding at rs2836882 in TPP macrophages (one-sample *t*-test, two-sided, n = 5). Error bars represent mean±95%CI. **e**. Schematic of PU.1 MPRA-ChIP assay to assess allele-specific PU.1 binding at individual SNPs within chr21q22 enhancer. **f**. Allele-specific PU.1 binding at SNPs within chr21q22 enhancer in TPP macrophages. Data represents the allelic ratio of normalized PU.1 binding for constructs centred on the SNP allele from the MPRA library (fixed-effects meta-analysis of QuASAR-MPRA results, two-sided, n = 6). Box represents median (IQR), whiskers represent minima and maxima. **g**. Allele-specific ATAC-seq reads at rs2836882 in two deeply sequenced heterozygous TPP macrophage datasets (left: 154.7 million non-duplicate paired-end reads, right: 165.4 million non-duplicate paired-end reads). **h**. H3K27ac ChIP-seq data from risk (red) or non-risk (blue) allele homozygotes at rs2836882 (n = 4). **i**. Rank Ordering of Super-Enhancers (ROSE) analysis of H3K27ac ChIP-seq data from TPP macrophages from major (left) and minor (right) allele homozygotes. Dashed line denotes inflection point of curve, with enhancers above this point being denoted as super-enhancers. Red points indicate rs2836882-containing chr21q22 enhancer. SE, super-enhancer. The diagrams in **a**, **b** and **e** were created using BioRender.

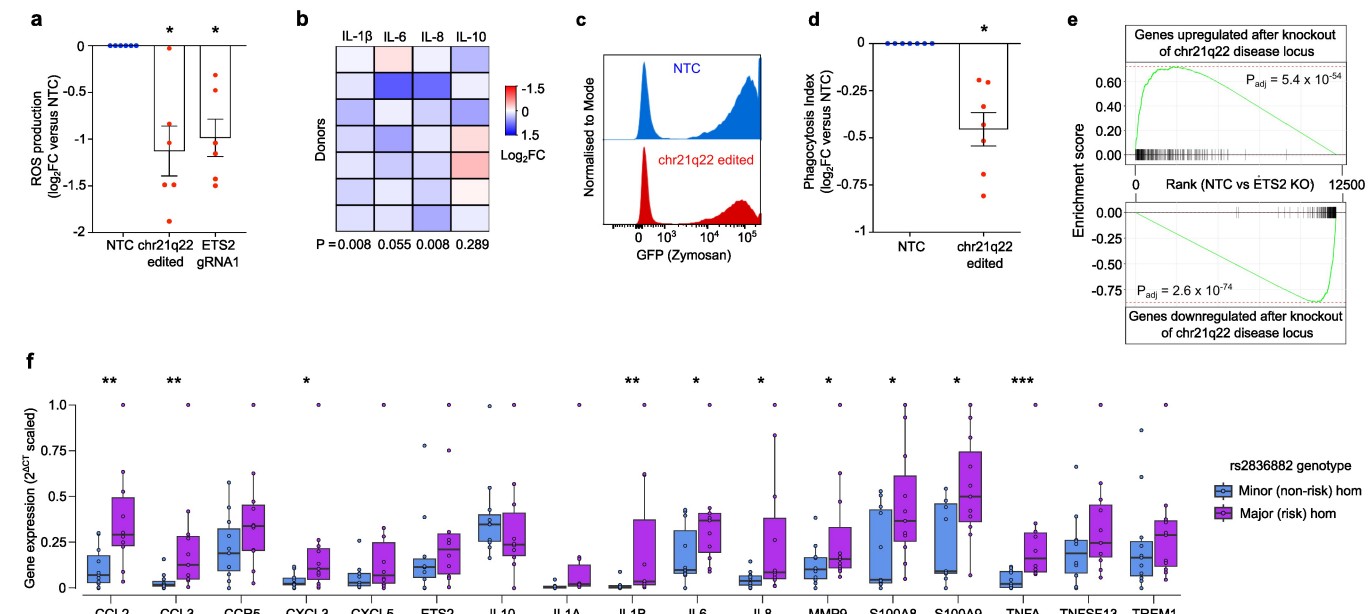

**Extended Data Fig. 5 | Functional effects of the chr21q22 enhancer.**
**a**. Extracellular ROS production by unedited (NTC), chr21q22-edited, and *ETS2* g1-edited TPP macrophages, quantified by chemiluminescence. Points represent relative area under curve for edited versus unedited cells (Wilcoxon signed-rank test, two-sided; n = 6). **b**. Cytokine secretion from inflammatory macrophages following deletion of the chr21q22 enhancer. Heatmap shows relative cytokine concentrations in the supernatants of chr21q22-edited TPP macrophages versus unedited (NTC) cells (Wilcoxon signed rank test, one-sided; n = 7). **c**. Representative flow cytometry histograms demonstrating phagocytosis of fluorescently-labelled zymosan particles by chr21q22-edited and unedited (NTC) TPP macrophages. **d**. Phagocytosis index for unedited and chr21q22-edited TPP macrophages, calculated as proportion of positive cells multiplied by mean fluorescence intensity of positive cells. Plot shows relative phagocytosis index for chr21q22-edited cells versus unedited cells (Wilcoxon signed-rank test two-sided; n = 7). **e**. Enrichment of differentially-expressed genes following deletion of the disease-associated chr21q22 locus (upregulated genes, top; downregulated genes, bottom) in *ETS2*-edited versus unedited macrophages. $P_{adj}$, FDR-adjusted *P*-value (two-sided). **f**. Tukey box-and-whisker plot depicting quantitative PCR of selected ETS2-target genes in resting (M0) macrophages from minor and major allele homozygote IBD patients (n = 22, expression normalized to *PPIA* and scaled to minimum 0, maximum 1). Mann-Whitney test (one-sided). * *P* < 0.05, ** *P* < 0.01, *** *P* < 0.001.

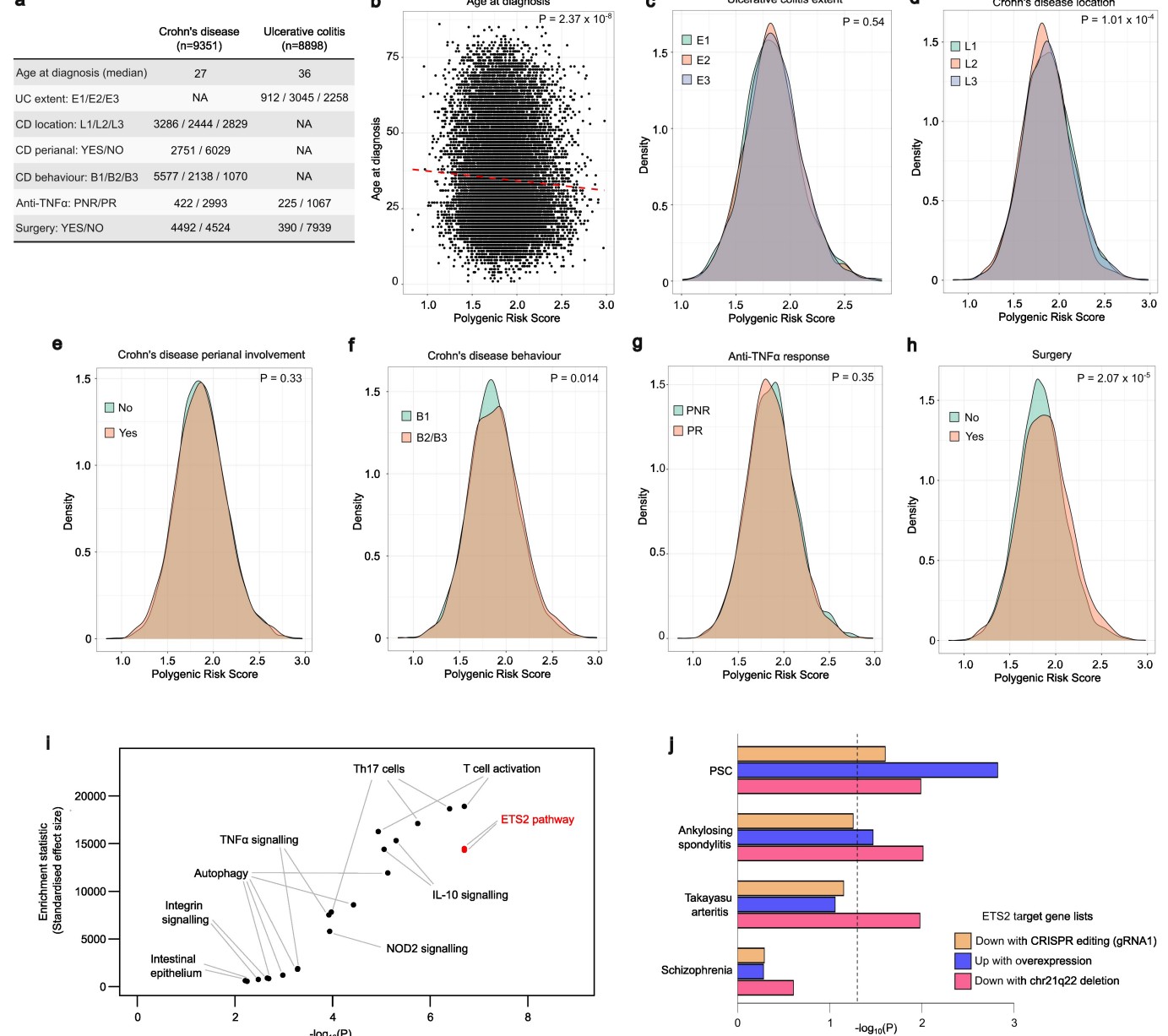

**Extended Data Fig. 6 | Polygenic Risk Score of 22 ETS2-regulated IBD-associated genes. a**. Summary of IBD BioResource cohorts used for PRS analysis. **b**. Association between PRS and age at diagnosis. **c**. Association between PRS and extent of ulcerative colitis (E1, proctitis; E2, left-sided; E3, extensive colitis). **d**. Association between PRS and Crohn's disease location (L1, ileal; L2, colonic; L3, ileocolonic). L2 is associated with a milder disease phenotype. **e**. Association between PRS and perianal involvement in Crohn's disease. **f**. Association between PRS and Crohn's disease behaviour (B1, inflammatory; B2, stricturing; B3, fistulating). B2 and B3 represent more aggressive, complicated forms of Crohn's disease. **g**. Association between PRS and response to anti-TNFα in Crohn's disease and ulcerative colitis (PR, primary responder; PNR, primary non-responder). **h**. Association between PRS and need for surgery in Crohn's disease and ulcerative colitis. Overall, higher PRS

was associated with: earlier age at diagnosis, ileal or ileocolonic forms of Crohn's disease, B2/B3 Crohn's disease behaviour, and increased need for surgery in IBD. Analysis in **b** performed using linear regression. Analyses in **c-h** performed using logistic regression (with diagnosis as covariate in **g** and **h**). SNPs included in PRS are listed in Extended Data Table 1. **i**. Plot of enrichment statistic (standardized effect size) against statistical significance from SNPsea analysis of genes tagged by 241 IBD SNPs within *ETS2*-regulated genes (red) and known IBD pathways (black). **j**. SNPsea analyses of SNPs associated with PSC, ankylosing spondylitis, Takayasu's arteritis or Schizophrenia (negative control) within lists of ETS2-regulated genes–either upregulated by *ETS2* overexpression, downregulated by *ETS2* disruption, or downregulated following chr21q22 deletion (all FDR < 0.05). Dashed line denotes P < 0.05.

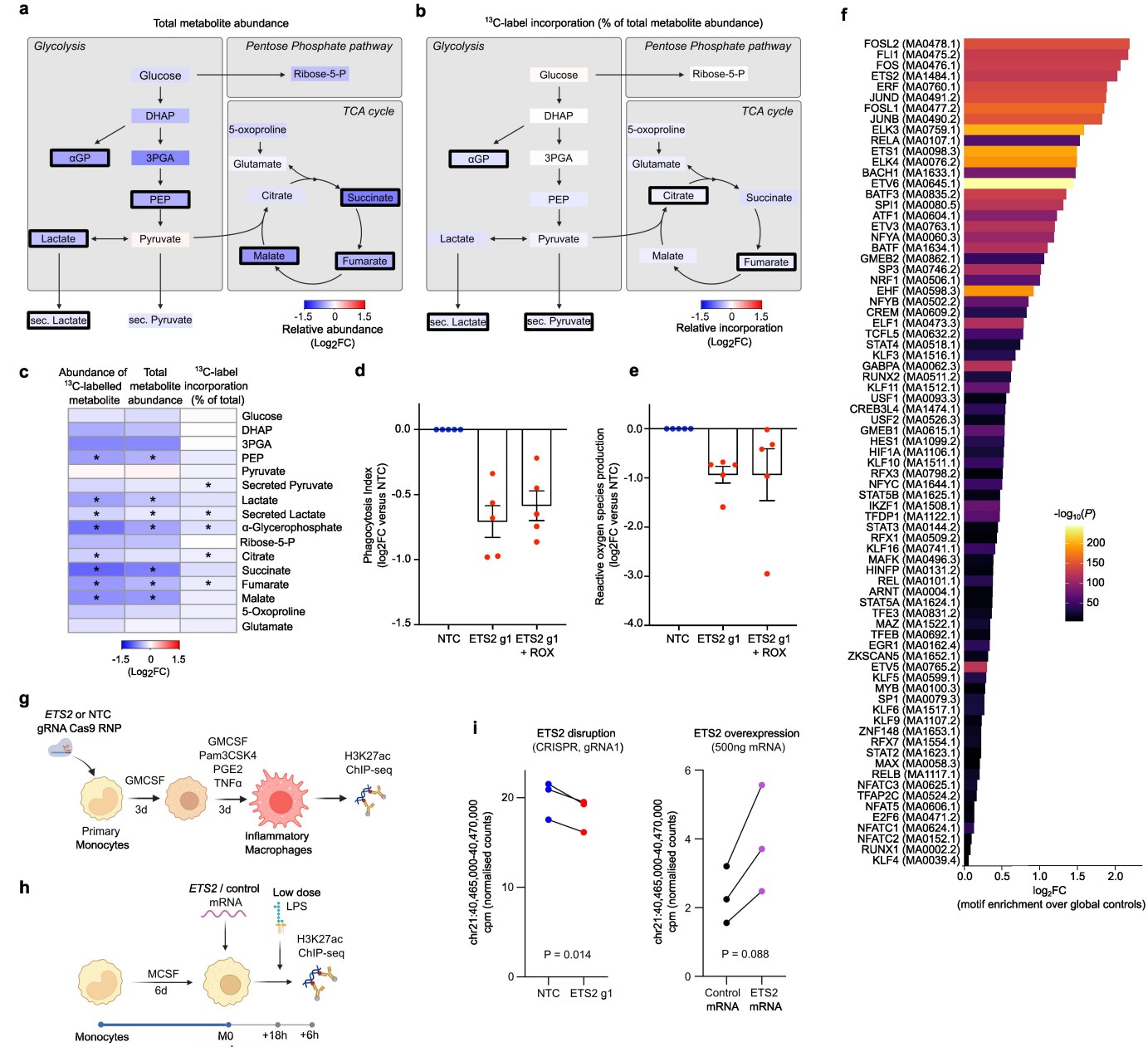

**Extended Data Fig. 7 | Effects of modulating *ETS2*. a** and **b**. Changes in total metabolite abundance (**a**) and percentage of label incorporation from $^{13}$C-glucose (**b**) following *ETS2* editing in TPP macrophages (n = 6). Colour depicts median log2 fold-change in *ETS2*-edited macrophages relative to unedited macrophages (transfected with non-targeting control RNPs; NTC). Bold black border indicates *P* < 0.05 (Wilcoxon signed rank test, two-sided). **c**. Heatmap summarizing metabolic changes following *ETS2* disruption. Colour depicts median log2 fold-change in *ETS2* g1-edited cells relative to unedited cells (Wilcoxon signed rank test, two-sided, * *P* < 0.05). **d**. Phagocytosis index in unedited (NTC) and *ETS2*-edited TPP macrophages treated with roxadustat (ROX) or vehicle. Phagocytosis index is calculated as proportion of positive cells multiplied by mean fluorescence intensity of positive cells (488 nm channel). Data normalized to phagocytosis index in unedited cells (n = 5). **e**. Extracellular ROS production by unedited (NTC) and *ETS2*-edited TPP macrophages treated

with ROX or vehicle – quantified using a chemiluminescence assay. Data represent log2 fold-change of area under curve (AUC) normalized to unedited (NTC) TPP macrophages (n = 5). **f**. TFmotifView enrichment results for motifs of transcription factors expressed in TPP macrophages (CPM > 0.5) within ETS2 CUT&RUN peaks. Results shown for all significantly enriched transcription factors (Bonferroni P value < 0.05, two-sided) with motifs in more than 10% peaks. **g**. Schematic of experiment to assess how ETS2 disruption affects the activity of the chr21q22 ETS2 enhancer in inflammatory (TPP) macrophages. **h**. Schematic of experiment to assess how ETS2 overexpression affects the activity of the chr21q22 ETS2 enhancer in resting (M0) macrophages. **i**. Normalized H3K27ac ChIP-seq read counts (edgeR fitted values) from chr21:40,465,000-40,470,000 in experiments depicted in **g** (left) and **h** (right) (edgeR P values, two-sided, n = 3 for each). Error bars in **d** and **e** represent mean±SEM. The diagrams in **g** and **h** were created using BioRender.

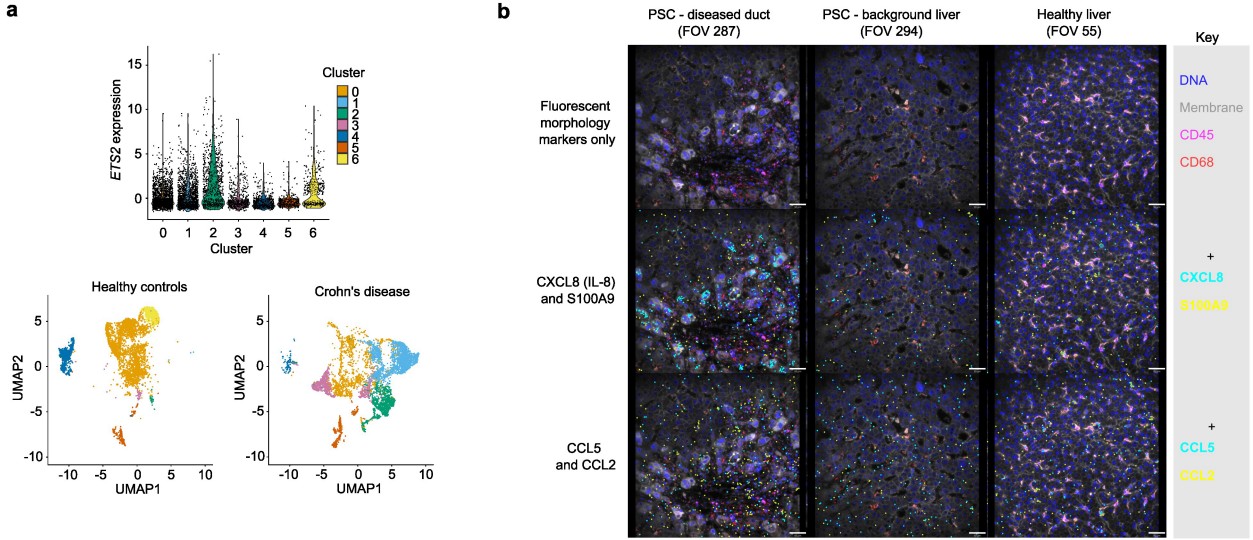

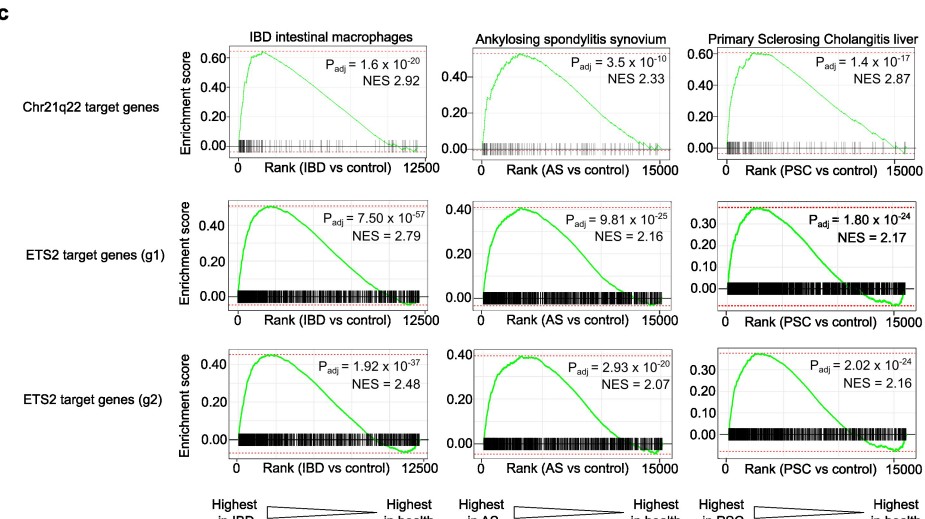

**Extended Data Fig. 8 | The transcriptional signature of *ETS2* is detectable in affected tissues from chr21q22-linked diseases. a**. ETS2 expression in scRNA-seq clusters of myeloid cells from Crohn's disease and healthy controls (upper panel). Relative contributions of single cells from Crohn's disease or healthy controls to individual clusters (same UMAP dimensions as for combined analysis). **b**. Overlay of CosMx morphology 2D image data and raw transcripts of selected ETS2 target genes. Fluorescent morphology markers alone (top row), CXCL8 (cyan) and S1009A (yellow) transcripts (middle row), CCL5 (cyan) and CCL2 (yellow) transcripts (bottom row). Columns are representative

examples of PSC with diseased ducts (left), PSC with uninflamed background liver (centre), and healthy liver (right). Size marker (white) on every field of view (FOV) denotes 50 μm. **c**. Gene set enrichment analysis (fGSEA) of genes downregulated following chr21q22 enhancer deletion or *ETS2* disruption (gRNA1 or gRNA2) within intestinal macrophages from patients with active IBD (compared to control intestinal macrophages, n = 20; left), ankylosing spondylitis synovium (compared to control synovium, n = 15; centre), and PSC liver biopsies (compared to control liver biopsies, n = 17; right). $P_{adj}$, FDR-adjusted *P*-value (two-sided).

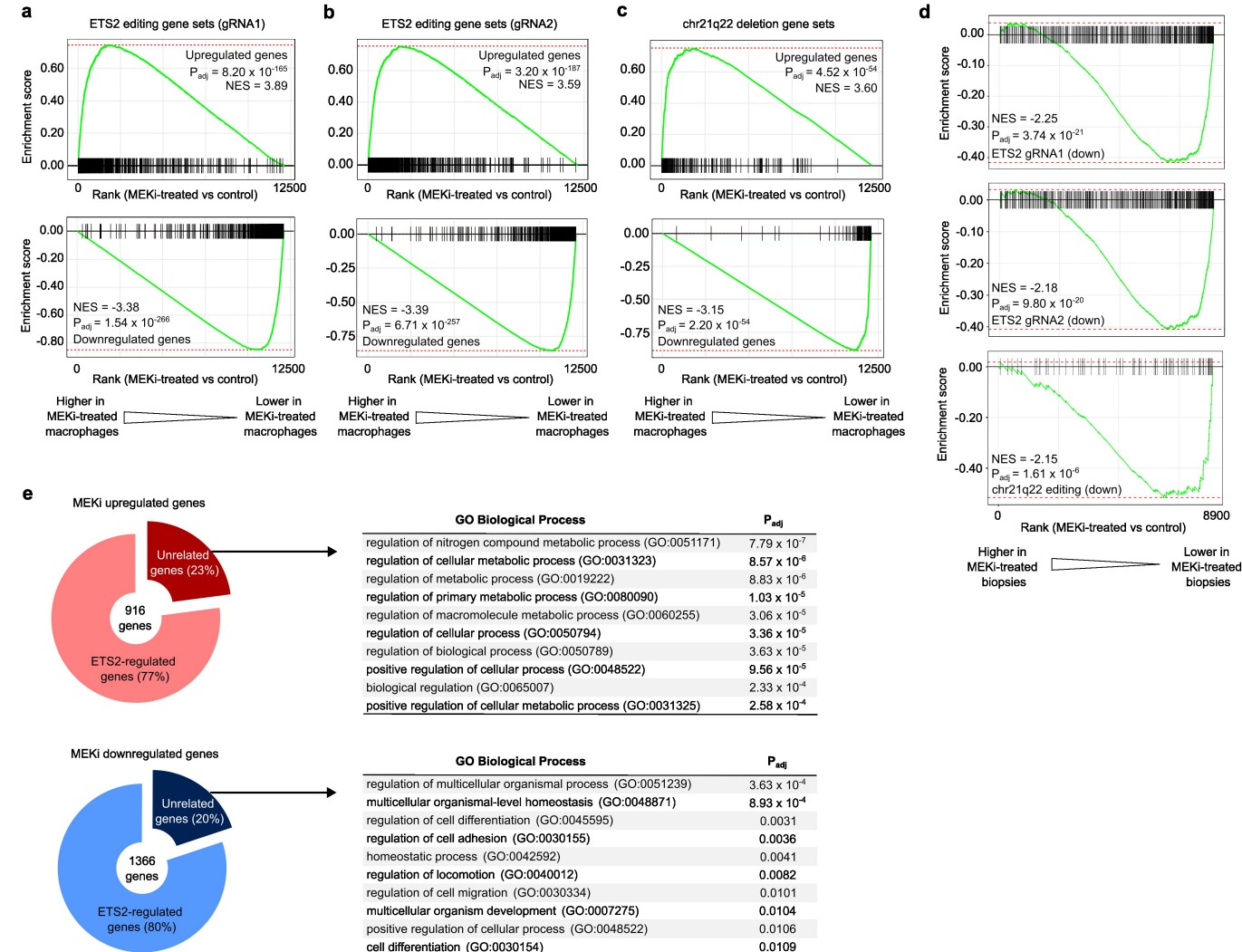

**Extended Data Fig. 9 | Effect of MEK1/2 inhibition on *ETS2*-regulated genes.**
**a-c.** Gene set enrichment analysis (fGSEA) in MEK1/2 inhibitor-treated TPP macrophages showing enrichment of gene sets upregulated (upper panel) or downregulated (lower panel) following *ETS2* or chr21q22 editing (MEK1/2 inhibited using PD-0325901, 0.5 μM). Gene sets obtained from differential gene expression analysis (limma using voom transformation) following ETS2 disruption with gRNA1 (**a**), gRNA2 (**b**), or following chr21q22 deletion (**c**). **d**. fGSEA in intestinal biopsies from IBD patients showing enrichment of gene sets downregulated following *ETS2* or chr21q22 editing in MEK inhibitor-treated biopsies. Upregulated gene sets were not enriched. **e**. Proportion and pathway analysis of MEK inhibitor-induced differentially expressed genes that have no evidence for being ETS2 targets in macrophages (incorporating differential expression from knockout or overexpression experiments and promoter / regulatory element binding from ETS2 CUT&RUN). $P_{adj}$, FDR-adjusted *P*-value (two-sided).

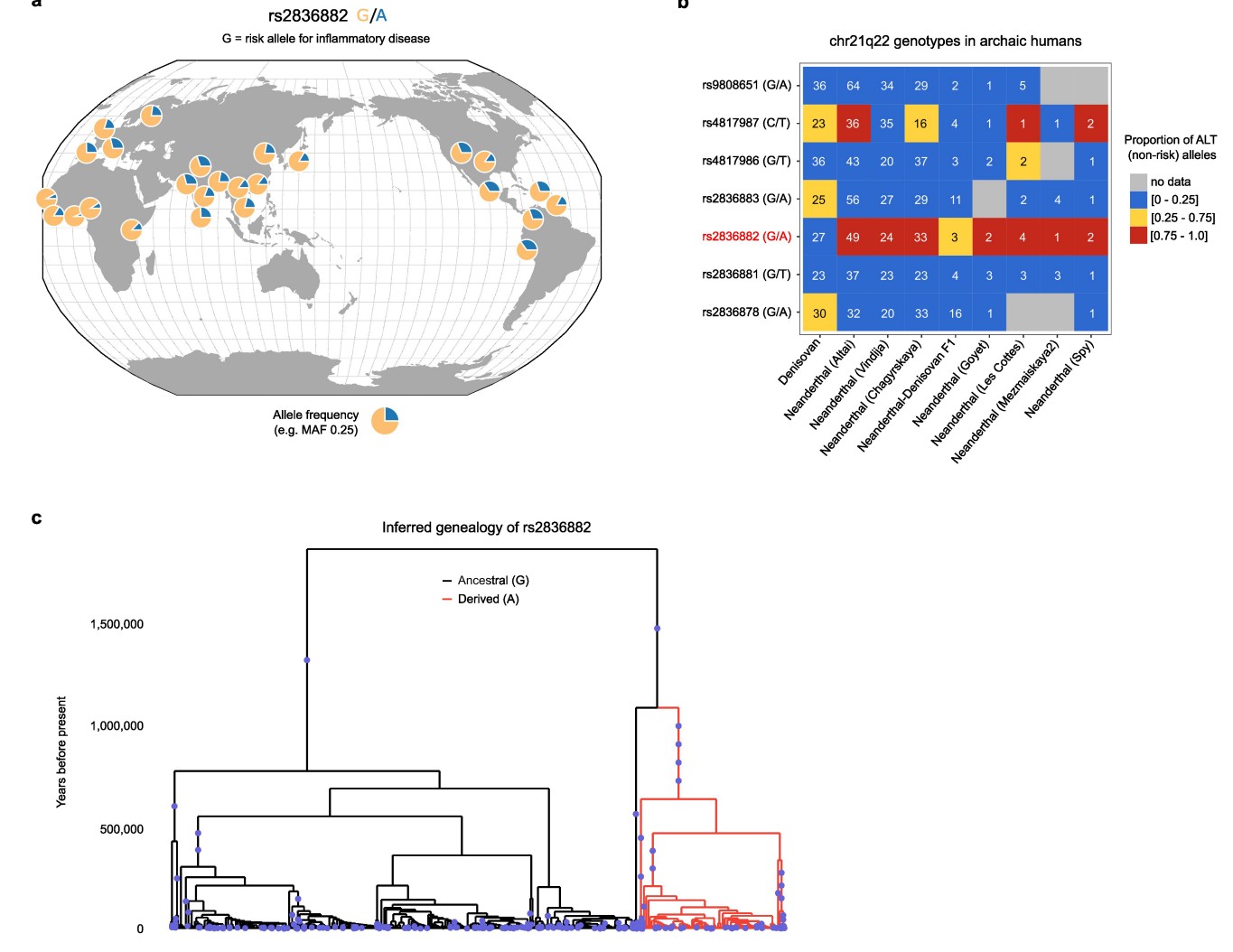

**Extended Data Fig. 10 | Geographic distribution and history of rs2836882.**
**a**. rs2836882 allele frequency in modern global populations (data from 1000 Genomes Project, plotted using Geography of Genetic Variants browser: https://popgen.uchicago.edu/ggv/). **b**. Genotypes of candidate SNPs at chr21q22 (99% credible set) in archaic humans (Neanderthals and Denisovans). Colour depicts the proportion of reads containing ALT alleles, with a value close to 0 consistent with a homozygous REF (risk) genotype, a value close to 1 consistent with a homozygous ALT (non-risk) genotype, and an intermediate value indicating a potential heterozygous genotype. Number in each cell indicates the number of reads at that SNP in the indicated sample. Putative causal variant highlighted in red. **c**. Inferred genealogy of the age of the rs2836882 polymorphism – analysed using Relate. The diagram in **a** was created using the Geography of Genetic Variants browser.

**Extended Data Table 1 | IBD risk genes downregulated following *ETS2* disruption**

| Ensembl ID | Gene ID | PMID | ETS2 gRNA 1 | | | ETS2 gRNA 2 | | |
|---|---|---|---|---|---|---|---|---|
| | | | logFC | *P* | adj.P.Val | logFC | *P* | adj.P.Val |
| ENSG00000164691 | *TAGAP* | 23128233 | -0.91 | 7.55E-08 | 8.48E-05 | -1.00 | 1.53E-08 | 6.18E-05 |
| ENSG00000005844 | ***ITGAL*** | 28658209 | -0.77 | 3.32E-07 | 1.29E-04 | -0.72 | 8.99E-07 | 5.09E-04 |
| ENSG00000179630 | ***LACC1*** | 23128233 | -0.73 | 4.04E-07 | 1.29E-04 | -0.65 | 2.18E-06 | 6.71E-04 |
| ENSG00000163735 | *CXCL5* | 23128233 | -1.14 | 1.10E-06 | 2.78E-04 | -0.90 | 3.17E-05 | 2.29E-03 |
| ENSG00000172575 | ***RASGRP1*** | 23128233 | -0.70 | 5.85E-05 | 2.88E-03 | -0.70 | 5.90E-05 | 3.10E-03 |
| ENSG00000166501 | ***PRKCB*** | 23128233 | -0.34 | 6.23E-05 | 3.01E-03 | -0.36 | 3.35E-05 | 2.35E-03 |
| ENSG00000197943 | ***PLCG2*** | 36038634 | -0.25 | 1.49E-03 | 1.94E-02 | -0.33 | 7.02E-05 | 3.51E-03 |
| ENSG00000081237 | ***PTPRC*** | 26192919 | -0.54 | 7.11E-05 | 3.32E-03 | -0.51 | 1.52E-04 | 5.46E-03 |
| ENSG00000158714 | ***SLAMF8*** | 36038634 | -0.56 | 1.16E-04 | 4.53E-03 | -0.54 | 1.80E-04 | 5.86E-03 |
| ENSG00000108691 | *CCL2* | 23128233 | -0.83 | 1.21E-04 | 4.60E-03 | -0.79 | 2.10E-04 | 6.51E-03 |
| ENSG00000163110 | ***PDLIM5*** † | 36038634 | -0.24 | 1.77E-03 | 2.17E-02 | -0.29 | 2.22E-04 | 6.71E-03 |
| ENSG00000136869 | ***TLR4*** | 26974007 | -0.49 | 6.73E-04 | 1.20E-02 | -0.54 | 2.22E-04 | 6.71E-03 |
| ENSG00000134242 | ***PTPN22*** | 28658209 | -0.40 | 2.66E-03 | 2.77E-02 | -0.48 | 5.70E-04 | 1.16E-02 |
| ENSG00000079263 | ***SP140*** | 26192919 | -0.42 | 6.09E-03 | 4.72E-02 | -0.52 | 9.96E-04 | 1.64E-02 |
| ENSG00000169403 | ***PTAFR*** † | 36038634 | -0.71 | 4.50E-04 | 9.70E-03 | -0.65 | 1.08E-03 | 1.71E-02 |
| ENSG00000150637 | *CD226* | 23128233 | -0.43 | 2.05E-03 | 2.36E-02 | -0.46 | 1.20E-03 | 1.82E-02 |
| ENSG00000143226 | ***FCGR2A*** | 23128233 | -0.99 | 2.01E-04 | 6.18E-03 | -0.79 | 1.93E-03 | 2.38E-02 |
| ENSG00000138821 | *SLC39A8* | 26192919 | -0.71 | 3.90E-04 | 8.98E-03 | -0.57 | 2.74E-03 | 2.93E-02 |
| ENSG00000100365 | ***NCF4*** | 36038634 | -0.59 | 5.26E-04 | 1.05E-02 | -0.47 | 3.77E-03 | 3.54E-02 |
| ENSG00000187796 | ***CARD9*** | 23128233 | -0.51 | 1.18E-03 | 1.67E-02 | -0.44 | 3.89E-03 | 3.61E-02 |
| ENSG00000121281 | ***ADCY7*** | 28067910 | -0.32 | 3.67E-03 | 3.42E-02 | -0.32 | 3.96E-03 | 3.66E-02 |
| ENSG00000144802 | ***NFKBIZ*** * | 26192919 | -0.28 | 1.55E-03 | 2.00E-02 | -0.23 | 7.10E-03 | 5.26E-02 |
| ENSG00000167207 | ***NOD2*** * | 28658209 | -0.78 | 3.94E-04 | 9.01E-03 | -0.53 | 9.66E-03 | 6.32E-02 |

Results shown for IBD-associated genes that were differentially expressed between *ETS2*-edited and unedited (NTC) TPP macrophages (n = 9). Log2 fold-change is with respect to expression in unedited cells. Genes in bold have been denoted as causal at their respective loci or are the only candidate gene at the locus. *P*-value (two-sided) adjusted for multiple testing using Benjamini-Hochberg method. PMID denotes PubMed ID of study with strongest IBD association for each gene. *Consistent effect but adjusted *P* value (adj.P.val) <0.05 for one gRNA only. †Not included in PRS (rare variant that was not genotyped in IBD BioResource).

# Reporting Summary

## Statistics

For all statistical analyses, confirm that the following items are present in the figure legend, table legend, main text, or Methods section.

| n/a | Confirmed | |
|---|---|---|
| ☐ | ☒ | The exact sample size (*n*) for each experimental group/condition, given as a discrete number and unit of measurement |
| ☐ | ☒ | A statement on whether measurements were taken from distinct samples or whether the same sample was measured repeatedly |
| ☐ | ☒ | The statistical test(s) used AND whether they are one- or two-sided<br>*Only common tests should be described solely by name; describe more complex techniques in the Methods section.* |
| ☐ | ☒ | A description of all covariates tested |
| ☐ | ☒ | A description of any assumptions or corrections, such as tests of normality and adjustment for multiple comparisons |
| ☐ | ☒ | A full description of the statistical parameters including central tendency (e.g. means) or other basic estimates (e.g. regression coefficient) AND variation (e.g. standard deviation) or associated estimates of uncertainty (e.g. confidence intervals) |
| ☐ | ☒ | For null hypothesis testing, the test statistic (e.g. *F*, *t*, *r*) with confidence intervals, effect sizes, degrees of freedom and *P* value noted<br>*Give P values as exact values whenever suitable.* |
| ☐ | ☒ | For Bayesian analysis, information on the choice of priors and Markov chain Monte Carlo settings |
| ☒ | ☐ | For hierarchical and complex designs, identification of the appropriate level for tests and full reporting of outcomes |
| ☐ | ☒ | Estimates of effect sizes (e.g. Cohen's *d*, Pearson's *r*), indicating how they were calculated |

*Our web collection on statistics for biologists contains articles on many of the points above.*

## Software and code

Policy information about availability of computer code

**Data collection**

FACS Diva v.6.0 was used to collect flow cytometry data. Next-generation sequencing data was obtained using in-built software on Illumina Sequencers and demultiplexed and converted to FASTQ files using bcl2fastq (v.1.8.4) or bcl2fastq2 (v.2.20). U-PLEX data was obtained using DISCOVERY WORKBENCH Software (v.4.0). NanoString CosMx SMI was used to collect spatial transcriptomics data.

**Data analysis**

Code to reproduce analyses are available at https://github.com/JamesLeeLab/chr21q22_manuscript, https://github.com/chr1swallace/ibd-ets2-analysis, and https://github.com/qzhang314/PRS_IBD_subpheno. Flow cytometry data was gated with FlowJo v10.8.1. Metabolomic GC-MS analysis was performed using MANIC (v.3.0), an adaptation of GAVIN. Densitometry analysis was performed using ImageJ (v1.53g). Next-generation sequencing data was pre-processed using FastQC (v.0.11.8), MultiQC (v.0.9), Trim Galore (v.0.6.0), Picard (v.2.1.1), deepTools (v.3.3.1), BBSplit (from BBMap v.36.20), Burrows-Wheeler Aligner (ChIP-seq, ATC-seq) or Bowtie2 (v.2.4.4; CUT&RUN) or HISAT2 (v.2.1.0; RNA-seq), Rsubread (v.2.12.3), and splitSNP (ATAC-seq). Polygenic risk score analysis was run using Plink (v.1.9). Spatial transcriptomics data were pre-processed using the AoMx Spatial Informatics Platform (NanoString Technologies). The following R (v.4.2.1) packages were used in this work: fine-mapping: susieR (v.0.12.35), ssimp (v.0.5.6), co-localisation: coloc (v.5.2.0), allele-specific binding: BaalChIP (v.1.12.0), gene-set variation analysis: GSVA (v.1.44.5), gene-set enrichment analysis: fgsea (v.1.23.2), differential expression analysis: limma (v.3.52.4), irreproducible discovery rate: idr (v.1.3), GenomicRanges (v.1.48.0), MPRA analysis: QuASAR (v.0.1) and les (v.1.46.0), co-expression analysis: Hmisc (v5.0.1), single cell RNA-seq: Seurat (v.4), spatial transcriptomics: InsituType, plots were made using EnhancedVolcano (v.1.14.0), EnhancedHeatmap (v.1.26.0), ggplot2 (v.3.4.0), gplots (v.3.1.3), karyoploteR (v.1.22.0). The following Python packages were used: peak-calling: MACS2 (v.2.2.5), genetic enrichment: SNPsea (v.1.0.3), Rank-Ordering of Super-Enhancers: ROSE, ChIP-seq pre-processing: SAMtools (v.1.11), spatial transcriptomics: napari (v0.4.17).

For manuscripts utilizing custom algorithms or software that are central to the research but not yet described in published literature, software must be made available to editors and reviewers. We strongly encourage code deposition in a community repository (e.g. GitHub). See the Nature Portfolio guidelines for submitting code & software for further information.

## Data

Policy information about availability of data

All manuscripts must include a data availability statement. This statement should provide the following information, where applicable:
- Accession codes, unique identifiers, or web links for publicly available datasets
- A description of any restrictions on data availability
- For clinical datasets or third party data, please ensure that the statement adheres to our policy

The datasets produced in this study are accessible via the following repositories:
MPRA (GEO: GSE229472)
RNA-seq of ETS2 or chr21q22-edited TPP macrophages (EGA: EGAD00001011338)
RNA-seq of ETS2 overexpression (EGA: EGAD00001011341)
RNA-seq of MEK inhibitor-treated TPP macrophages (EGA: EGAD00001011337)
H3K27ac ChIP-seq in TPP macrophages (EGA: EGAD00001011351)
ATAC-seq and H3K27ac ChIP-seq in ETS2-overexpressing or –edited macrophages (EGA: EGAD50000000154)
ETS2 CUT&RUN (EGA: EGAD00001011349)
Biopsy RNA-seq data (EGA: EGAD00001011333)
MetaboLights: Metabolomics (MTBLS7665)
The phenotype and genotype data used for the PRS analysis are available upon application to the IBD Bioresource (https://www.ibdbioresource.nihr.ac.uk/).

## Research involving human participants, their data, or biological material

Policy information about studies with human participants or human data. See also policy information about sex, gender (identity/presentation), and sexual orientation and race, ethnicity and racism.

| | |
|---|---|
| Reporting on sex and gender | Information on sex and gender on leukocyte cones were anonymised at the point of collection and not provided to the research team. Sex information was provided for inflammatory bowel disease patient samples for matching. |
| Reporting on race, ethnicity, or other socially relevant groupings | We do not report on race, ethnicity, or other socially relevant groupings. |
| Population characteristics | Population characteristics of human research participants were not used as covariates. |
| Recruitment | Leukocyte apheresis cones were collected by NHS Blood and Transplant from healthy platelet donors. Patients with active inflammatory bowel disease, who were not receiving immunosuppressive or biologic therapies, were recruited by clinicians at the Royal Free Hospital London. Inflammatory bowel disease patients who were homozygous for either the rs2836882 risk or non-risk allele were recruited via the NIHR BioResource. |
| Ethics oversight | Ethical approval to obtain blood from healthy individuals and samples from inflammatory bowel disease patients was provided by the London - Brent Regional Ethics Committee (REC: 21/LO/0682). Liver samples were provided by the Tissue Access for Patient Benefit (TAP-B, part of the UCL-RFH BioBank) with approval from the Wales Research Ethics Committee 4 (REC 16/WA/0289). |

Note that full information on the approval of the study protocol must also be provided in the manuscript.

# Field-specific reporting

Please select the one below that is the best fit for your research. If you are not sure, read the appropriate sections before making your selection.

☒ Life sciences    ☐ Behavioural & social sciences    ☐ Ecological, evolutionary & environmental sciences

For a reference copy of the document with all sections, see nature.com/documents/nr-reporting-summary-flat.pdf

# Life sciences study design

All studies must disclose on these points even when the disclosure is negative.

| | |
|---|---|
| Sample size | Where relevant, sample sizes were pre-determined using power calculations based on anticipated effect sizes and variability observed in previous similar experiments. Sample size is stated in each panel and in the Methods. |
| Data exclusions | Data were only excluded if they failed pre-defined quality control metrics or if the sample had insufficient material to perform the experiment (applied to one biopsy RNA-seq sample). Otherwise results represent all data points collected from the experiments indicated. |
| Replication | The number of biological replicates for individual experiments are stated in figure legends and methods. Technical replicates were summarised before statistical analyses. All experimental findings were reproducible - as indicated by the statistical analysis described. |

| | |
|---|---|
| Randomization | Biopsy culture experiments were performed by randomly allocating 2 biopsies to each experimental condition. H3K27ac ChIP-seq in minor and major allele homozygous samples and RT-qPCR in inflammatory bowel disease patient samples could not be randomised due to genotyping. All other experiments included cells from the same donor collected at the same time in different experimental groups e.g. transfection of non-targeting control RNPs for CRISPR, reverse complement ETS2 for overexpression, IgG pulldowns for CUT&RUN, vehicle controls for MEK inhibitor samples. |
| Blinding | Blinding was not performed as all experiments were based on objective quantitative measurements (e.g. flow cytometry, CUT&RUN, RNA-seq) and analyses were automated and applied equally to all samples. |

# Reporting for specific materials, systems and methods

We require information from authors about some types of materials, experimental systems and methods used in many studies. Here, indicate whether each material, system or method listed is relevant to your study. If you are not sure if a list item applies to your research, read the appropriate section before selecting a response.

## Materials & experimental systems

| n/a | Involved in the study |
|---|---|
| ☐ | ☒ Antibodies |
| ☒ | ☐ Eukaryotic cell lines |
| ☐ | ☒ Palaeontology and archaeology |
| ☒ | ☐ Animals and other organisms |
| ☒ | ☐ Clinical data |
| ☒ | ☐ Dual use research of concern |
| ☒ | ☐ Plants |

## Methods

| n/a | Involved in the study |
|---|---|
| ☐ | ☒ ChIP-seq |
| ☐ | ☒ Flow cytometry |
| ☒ | ☐ MRI-based neuroimaging |

## Antibodies

| | |
|---|---|
| Antibodies used | PU.1 antibody, #2266S, Cell Signaling<br>anti-H3K27ac antibody, ab4729, Abcam<br>Recombinant rabbit IgG isotype control, ab172730, clone EPR25A, Abcam<br>gp91phox antibody, sc-130543, Santa Cruz<br>p22phox antibody, sc-20781, Santa Cruz<br>anti-C17ORF62/EROS antibody, HPA045696, Atlas Antibodies<br>anti-vinculin, V4505, Sigma<br>goat anti-mouse IgG-horseradish peroxidase, 115-036-071, Jackson Immuno<br>goat anti-rabbit IgG-horseradish peroxidase, 111-036-046, Jackson Immuno<br>ETS2 antibody, PA528053, ThermoFisher<br>rabbit IgG isotype control, #66362, clone DA1E, Cell Signaling<br>PE/Dazzle 594 CD11b antibody, #301347, clone ICRF44, BioLegend<br>evolve405 CD14 antibody, #83-0149-42, clone 61D3, ThermoFisher<br>PerCP CD16 antibody, #302029, clone 3G8, BioLegend<br>FITC CD68 antibody, #333805, clone Y1/82A, BioLegend |
| Validation | PU.1 antibody, #2266S, Cell Signaling: validated for ChIP-sequencing: https://www.cellsignal.com/products/primary-antibodies/pu-1-antibody/2266<br>anti-H3K27ac antibody, Ab4729, Abcam: validated for ChIP-sequencing: https://www.abcam.com/products/primary-antibodies/histone-h3-acetyl-k27-antibody-chip-grade-ab4729.html<br>Recombinant rabbit IgG isotype control, ab172730, Abcam: validated for ChIP-sequencing: https://www.abcam.com/products/primary-antibodies/rabbit-igg-monoclonal-epr25a-isotype-control-ab172730.html<br>gp91phox antibody, sc-130543, Santa Cruz: validated for western blot: https://datasheets.scbt.com/sc-130543.pdf<br>p22phox antibody, sc-20781, Santa Cruz: validated for western blot: https://datasheets.scbt.com/sc-20781.pdf<br>anti-C17ORF62/EROS antibody, HPA045696, Atlas Antibodies: previously validated to bind to EROS in PMID: 36421765<br>anti-vinculin, V4505, Sigma: validated for western blot: https://www.sigmaaldrich.com/GB/en/product/sigma/v4505<br>goat anti-mouse IgG-horseradish peroxidase, 115-036-071, Jackson Immuno - validated for western blot: https://www.jacksonimmuno.com/catalog/products/115-036-071<br>goat anti-rabbit IgG-horseradish peroxidase, 111-036-046, Jackson Immuno - validated for western blot: https://www.jacksonimmuno.com/catalog/products/111-036-046<br>ETS2 antibody, PA528053, ThermoFisher: validated for Western blot: https://www.thermofisher.com/antibody/product/ETS2-Antibody-Polyclonal/PA5-28053; produced reproducible CUT&RUN peaks that met acceptable quality metrics<br>rabbit IgG isotype control, #66362, Cell Signaling; validated for CUT&RUN: https://www.cellsignal.com/products/primary-antibodies/rabbit-da1e-mab-igg-xp-isotype-control-cut-amp-run/66362<br>PE/Dazzle 594 CD11b antibody, #301347, BioLegend: validated for flow cytometry: https://www.biolegend.com/en-gb/explore-new-products/pe-dazzle-594-anti-human-cd11b-antibody-10195<br>evolve605 CD14 antibody, #83-0149-42, ThermoFisher: validated for flow cytometry in PMID: 28813661<br>PerCP CD16 antibody, #302029, BioLegend: validated for flow cytometry; https://www.biolegend.com/de-at/products/percp-anti-human-cd16-antibody-4340<br>FITC CD68 antibody, #333805, BioLegend: validated for flow cytometry: https://www.biolegend.com/nl-be/products/fitc-anti-human-cd68-antibody-4844 |

# Palaeontology and Archaeology

| | |
|---|---|
| Specimen provenance | Genotypes at the disease-associated chr21q22 candidate SNPs were determined using publicly available genomes from seven Neanderthal individuals, one Denisovan individual, and one Neanderthal and Denisovan F1 individual. |
| Specimen deposition | Publicly available genomes. |
| Dating methods | No new dates are provided. |

☒ Tick this box to confirm that the raw and calibrated dates are available in the paper or in Supplementary Information.

| | |
|---|---|
| Ethics oversight | Because the genomes were publicly available, no ethical approval or guidance was necessary. |

Note that full information on the approval of the study protocol must also be provided in the manuscript.

# Plants

| | |
|---|---|
| Seed stocks | *Report on the source of all seed stocks or other plant material used. If applicable, state the seed stock centre and catalogue number. If plant specimens were collected from the field, describe the collection location, date and sampling procedures.* |
| Novel plant genotypes | *Describe the methods by which all novel plant genotypes were produced. This includes those generated by transgenic approaches, gene editing, chemical/radiation-based mutagenesis and hybridization. For transgenic lines, describe the transformation method, the number of independent lines analyzed and the generation upon which experiments were performed. For gene-edited lines, describe the editor used, the endogenous sequence targeted for editing, the targeting guide RNA sequence (if applicable) and how the editor was applied.* |
| Authentication | *Describe any authentication procedures for each seed stock used or novel genotype generated. Describe any experiments used to assess the effect of a mutation and, where applicable, how potential secondary effects (e.g. second site T-DNA insertions, mosiacism, off-target gene editing) were examined.* |

# ChIP-seq

## Data deposition

☒ Confirm that both raw and final processed data have been deposited in a public database such as GEO.

☒ Confirm that you have deposited or provided access to graph files (e.g. BED files) for the called peaks.

| | |
|---|---|
| Data access links<br>*May remain private before publication.* | Data have been deposited in EGA:<br>H3K27ac ChIP-seq in TPP macrophages (EGA: EGAD00001011351)<br>H3K27ac ChIP-seq in ETS2-overexpressing or -edited macrophages (EGA: EGAD50000000154)<br>ETS2 CUT&RUN (EGA: EGAD00001011349) |
| Files in database submission | H3K27ac ChIP-seq in TPP macrophages<br>- H3K27ac ChIP, rs2836882 MajorAlleleHom, rep1<br>- H3K27ac ChIP, rs2836882 MajorAlleleHom, rep2<br>- H3K27ac ChIP, rs2836882 MinorAlleleHom, rep1<br>- H3K27ac ChIP, rs2836882 MinorAlleleHom, rep2<br>- Input DNA, rs2836882 MajorAlleleHom, rep1<br>- Input DNA, rs2836882 MajorAlleleHom, rep2<br>- Input DNA, rs2836882 MinorAlleleHom, rep1<br>- Input DNA, rs2836882 MinorAlleleHom, rep2<br><br>ETS2 CUT&RUN in TPP macrophages<br>- TPP_donor1, ETS2 CUT&RUN<br>- TPP_donor1_IgG control CUT&RUN<br>- TPP_donor2_ETS2 CUT&RUN<br>- TPP_donor2_IgG control CUT&RUN<br><br>H3K27ac ChIP-seq in ETS2-overexpressing or ETS2-disrupted TPP macrophages<br>- H3K27ac ChIP, ETS2 500 ng, donor1<br>- H3K27ac ChIP, ETS2 REV 500 ng, donor1<br>- H3K27ac ChIP ETS2 500 ng, donor2<br>- H3K27ac ChIP ETS2 REV 500 ng, donor2<br>- H3K27ac ChIP ETS2 500 ng, donor3<br>- H3K27ac ChIP ETS2 REV 500 ng, donor3<br>- Input DNA, ETS2 500 ng, donor1<br>- Input DNA, ETS2 REV 500 ng, donor1<br>- Input DNA, ETS2 500 ng, donor2<br>- Input DNA, ETS2 REV 500 ng, donor2 |

- Input DNA, ETS2 500 ng, donor3
- Input DNA, ETS2 REV 500 ng, donor3
- H3K27ac ChIP, NTC, donor1
- H3K27ac ChIP, ETS2KO, donor1
- H3K27ac ChIP, NTC, donor2
- H3K27ac ChIP, ETS2KO, donor2
- H3K27ac ChIP, NTC, donor3
- H3K27ac ChIP, ETS2KO, donor3
- Input DNA, NTC, donor1
- Input DNA, ETS2KO, donor1
- Input DNA, NTC, donor2
- Input DNA, ETS2KO, donor2
- Input DNA, NTC, donor3
- Input DNA, ETS2KO, donor3

**Genome browser session**
(e.g. UCSC)

IGV session including tracks from ETS2 CUT&RUN, ATAC-seq in TPP macrophages, and H3K27ac ChIP-seq in TPP macrophages from rs2836882 minor and major allele homozygotes available at https://tinyurl.com/23g9h3bn

## Methodology

**Replicates**

Biological replicates (healthy donors) for H3K27ac ChIP:
2 replicates: rs2836882 homozygote major (H3K27ac ChIP-seq and Input)
2 replicates: rs2836882 homozygote minor (H3K27ac ChIP-seq and Input)

Biological replicates (healthy donors) for ETS2 CUT&RUN:
2 replicates for ETS2 CUT&RUN in TPP macrophages with corresponding IgG controls

Biological replicates (healthy donors) for H3K27ac ChIP-seq in ETS2-overexpressing or ETS2-disrupted TPP macrophages:
3 replicates: ETS2 500 ng and REV ETS2 500 ng (H3K27ac ChIP-seq and Input)
3 replicates: NTC and ETS2 KO (H3K27ac ChIP-seq and Input)

**Sequencing depth**

H3K27ac ChIP. All reads are single-end, 50bp in length.

| Sample | Read depth (M) | Uniquely mapped reads |
|---|---|---|
| - H3K27ac ChIP, MajorAlleleHom, rep1 | 28.9 | 16336754 |
| - H3K27ac ChIP, MajorAlleleHom, rep2 | 30.7 | 20058550 |
| - H3K27ac ChIP, MinorAlleleHom, rep1 | 27.4 | 15622745 |
| - H3K27ac ChIP, MinorAlleleHom, rep2 | 24.9 | 16204315 |
| - Input DNA, MajorAlleleHom, rep1 | 21.4 | 12576448 |
| - Input DNA, MajorAlleleHom, rep2 | 22 | 14679612 |
| - Input DNA, MinorAlleleHom, rep1 | 26.6 | 14253281 |
| - Input DNA, MinorAlleleHom, rep2 | 25.5 | 16069327 |

ETS2 CUT&RUN. All reads are paired-end, 100bp in length.

| Sample | Read depth | Read Pair Unique | Read Pair Not Optical Duplicates |
|---|---|---|---|
| - TPP_donor1, ETS2 CUT&RUN | 44,822,163 | 23,406,807 | 6,269,062 |
| - TPP_donor1_IgG control CUT&RUN | 46,996,015 | 17,294,478 | 5,400,929 |
| - TPP_donor2_ETS2 CUT&RUN | 50,848,338 | 20,508,435 | 6,612,914 |
| - TPP_donor2_IgG control CUT&RUN | 37,626,720 | 14,520,763 | 4,443,194 |

H3K27ac ChIP-seq in ETS2-overexpressing or ETS2-disrupted TPP macrophages. All reads are paired-end, 100bp in length.

| Sample | Uniquely mapped read pairs |
|---|---|
| H3K27ac_TPP_D1_ETS2_500 | 62821646 |
| H3K27ac_TPP_D1_REV_500 | 54400167 |
| H3K27ac_TPP_D1_ETS2input_500 | 62539567 |
| H3K27ac_TPP_D1_REVinput_500 | 55850854 |
| H3K27ac_TPP_D2_ETS2_500 | 59509120 |
| H3K27ac_TPP_D2_REV_500 | 57390645 |
| H3K27ac_TPP_D2_ETS2input_500 | 59945178 |
| H3K27ac_TPP_D2_REVinput_500 | 58958544 |
| H3K27ac_TPP_D3_ETS2_500 | 59798070 |
| H3K27ac_TPP_D3_REV_500 | 59256158 |
| H3K27ac_TPP_D3_ETS2input_500 | 54136028 |
| H3K27ac_TPP_D3_REVinput_500 | 54702503 |
| H3K27ac_TPP_NCI106NTC | 58039063 |
| H3K27ac_TPP_NCI106KO | 58709735 |
| H3K27ac_TPP_NCI106NTC_input | 57860350 |
| H3K27ac_TPP_NCI106KO_input | 65846125 |
| H3K27ac_TPP_NCI107NTC | 58746162 |
| H3K27ac_TPP_NCI107KO | 58965593 |
| H3K27ac_TPP_NCI107NTC_input | 64677003 |
| H3K27ac_TPP_NCI107KO_input | 58014156 |
| H3K27ac_TPP_NCI109NTC | 67028007 |
| H3K27ac_TPP_NCI109KO | 59109486 |
| H3K27ac_TPP_NCI109NTC_input | 61411094 |
| H3K27ac_TPP_NCI109KO_input | 59967525 |

| | |
|---|---|
| **Antibodies** | anti-H3K27ac antibody, ab4729, abcam<br>rabbit IgG, ab172730, abcam<br><br>anti-ETS2 antibody, PA528053, ThermoFisher Scientific<br>Rabbit (DA1E) mAb IgG XP® Isotype Control, #66362, Cell Signaling Technology |

**Peak calling parameters**

H3K27ac ChIP-seq:
macs2 callpeak -t sample1.bam -c input1.bed -g hs -n H3k27ac_1_hg19 -f AUTO --outdir PEAKS --qvalue 0.01 -B --nomodel --extsize=200

ETS2 CUT&RUN:
- initial peak calling: macs2 callpeak -t sample1.bam -c IgG1.bam -g hs -n CRETS2_Ther1_hg19 -f BAMPE --keep-dup all --qvalue 0.05 -B --nomodel
- idr on peak calls: mu = 3, sigma = 1, rho = 0.9, eps = 0.001, p = .5

Full code on GIthub (see below)

**Data quality**

H3K27ac ChIP-seq:
The data were trimmed with Trimgalore, with parameters: --phred33 -q 24 --illumina --length 30.
Unmapped , multi-mapped, chimeric and duplicate reads were excluded using Samtools, with parameters:  -b -h -F 4 -F 256 -F 1024 -F 2048 -q 15.
NSC and RSC values were obtained using phantompeakqualtools, with the SPP R package.
Peaks were called using a FDR threshold of 1% (cf. above).

| Sample | Number of peaks (FDR 1%) | Above 5-fold enrichment | NSC | RSC |
|---|---|---|---|---|
| H3K27ac, ChIP, MajorAlleleHom, rep1 | 40114 | 38820 | 1.3 | 1.1 |
| H3K27ac, ChIP, MajorAlleleHom, rep2 | 14604 | 14544 | 1.05 | 0.8 |
| H3K27ac, ChIP, MinorAlleleHom, rep1 | 34498 | 33594 | 1.3 | 1.1 |
| H3K27ac, ChIP, MinorAlleleHom, rep2 | 36419 | 36046 | 1.4 | 1.1 |

ETS2 CUT&RUN:
The data were trimmed with Trimgalore, with parameters: --phred33 -q 24 --illumina --length 25 --paired --stringency 6.
Reads were aligned with Bowtie2 following the guidelines by Skene & Henikoff (2017, Elife), with parameters: --local --very-sensitive-local --no-mixed --no-discordant --phred33 -I 10 -X 700.
Unmapped , multi-mapped and chimeric reads were excluded using Samtools, with parameters:  -b -h -F 4 -F 256 -F 2048 -q 15.
Peaks were called using a FDR threshold of 5% (cf. above) and IDR analysis was performed with a 1% cut-off (cf. main text and methods) to select the reproducible peaks between the 2 biological replicates.

| Sample | Number of peaks (FDR 5%) | Above 5-fold enrichment |
|---|---|---|
| TPP_donor1, ETS2 CUT&RUN | 46,813 | 38,183 |
| TPP_donor2_ETS2 CUT&RUN | 58,918 | 42,395 |

Irreproducible discovery rate: 6,560 reproducible peaks

H3K27ac ChIP-seq in ETS2-overexpressing or ETS2-disrupted TPP macrophages.
Data processed as described above.

| Sample | Number of peaks (FDR 1%) | Above 5-fold enrichment |
|---|---|---|
| H3K27ac_TPP_D1_ETS2_500 | 38930 | 18265 |
| H3K27ac_TPP_D1_REV_500 | 39781 | 16573 |
| H3K27ac_TPP_D2_ETS2_500 | 38915 | 15639 |
| H3K27ac_TPP_D2_REV_500 | 35716 | 13646 |
| H3K27ac_TPP_D3_ETS2_500 | 32279 | 14776 |
| H3K27ac_TPP_D3_REV_500 | 31847 | 15266 |
| H3K27ac_TPP_NCI106NTC | 50712 | 23417 |
| H3K27ac_TPP_NCI106KO | 48879 | 20446 |
| H3K27ac_TPP_NCI107NTC | 58789 | 25190 |
| H3K27ac_TPP_NCI107KO | 48588 | 23760 |
| H3K27ac_TPP_NCI109NTC | 53762 | 23990 |
| H3K27ac_TPP_NCI109KO | 52799 | 25044 |

**Software**

All code and software details are available at https://github.com/JamesLeeLab/chr21q22_manuscript/tree/main/ChIP-seq and https://github.com/JamesLeeLab/chr21q22_manuscript/tree/main/CUT%26RUN. Also deposited at https://zenodo.org/records/10707942.

# Flow Cytometry

## Plots

Confirm that:

☒ The axis labels state the marker and fluorochrome used (e.g. CD4-FITC).

☒ The axis scales are clearly visible. Include numbers along axes only for bottom left plot of group (a 'group' is an analysis of identical markers).

☒ All plots are contour plots with outliers or pseudocolor plots.

☒ A numerical value for number of cells or percentage (with statistics) is provided.

## Methodology

| | |
|---|---|
| Sample preparation | Monocytes were positively selected from leukocyte cones using CD14 Microbeads. Macrophage differentiation was performed using conditions that model chronic inflammation (TPP): 3 days GM-CSF (50ng/mL) followed by 3 days GM-CSF, TNFa (50ng/mL), PGE2 (1mg/mL), and Pam3CSK4 (1mg/mL).<br><br>RNA abundance was quantified by PrimeFlow (ThermoFisher) in TPP macrophages on days 0, 3, 4, 5, and 6 of TPP differentiation. Target probes specific for ETS2 (Alexa Fluor 647), BRWD1 (Alexa Fluor 568) and PSMG1 (Alexa Fluor 568) were used according to the manufacturer's instructions.<br><br>For assessment of myeloid marker expression, macrophages were detached with Accutase on day 6 of culture and were stained with CD11b PE/Dazzle 594, CD14 evolve405, CD16 PerCP, and CD68 FITC, along with Live/Dead Fixable Aqua Dead Cell Stain (ThermoFisher) and Fc Receptor Blocking Reagent (Miltenyi).<br><br>Phagocytosis was assayed by quantifying uptake of uptake of fluorescently-labelled Zymosan particles (Green Zymosan, Abcam) according to the manufacturer's instructions. Cells were stained with Live/Dead Fixable Aqua Dead Cell Stain (ThermoFisher) prior to flow cytometry. |
| Instrument | BD LSRFortessa X-20 |
| Software | FACS Diva was used to collect flow cytometry data. FlowJo v10 was used to for data analysis. |
| Cell population abundance | N/A - no sorting was performed |
| Gating strategy | Macrophages were gated by FSC-A/SSC-A and singlets were gated by FSC-A/FSC-H. Live cells were gated (and viability was quantified) using Live/Dead Fixable Aqua Dead Cell Stain. |

☒ Tick this box to confirm that a figure exemplifying the gating strategy is provided in the Supplementary Information.

