## [Peer Review File · Nature]

Manuscript Title: A disease-associated gene desert directs macrophage inflammation via ETS2

Reviewer Comments & Author Rebuttals

Reviewer Reports on the Initial Version:

Referees' comments:

Referee #1 (Remarks to the Author):

The manuscript by Stankey et al., entitled "" describes a functional genomics approach in humans to functionally and causally link a genetic alteration previously associated with several autoimmune diseases including IBD to the transcription factor ETS2, thereby elucidating the function of ETS2 in monocyte/macrophages in these diseases and finally also provide potential therapeutic strategies based on the biology of this TF. Overall, this manuscript is a conceptual advance, it illustrates how human genetics should move forward from descriptive GWAS and polygenic risk score science to address causation and function in a combined experimental and computational approach, which this author would like to term "functional genomics". Overall, the data, the approaches and the execution are of very high quality, yet, there is some room for improvements, which this reviewer has listed below (with some of the comments are more editorial):

Comments with the request to be addressed by the authors:

Title: The term "gene desert" is never really explained by the results. Why did the authors choose this word instead of making the point that mainly regulatory elements in intergenic regions of the genome are linked to diseases? Why not naming the identified function of the region instead of stating what it isn't (namely a gene region)?

Line 5: The statement about genetics is actually wrong: Genetics did not address this. The approach chosen by the authors is "functional genomics". As correctly stated by the authors GWAS itself (the tool of genetics) was incapable of being informative concerning the role of the SNP in question. The authors should change to "functional genomics"

Line 15: "identified" instead of "identify"

Line 15: Considering and as discussed by the authors in their discussion section that MEK inhibition will most likely not be easily applicable clinically, this statement should be altered to better reflect the principle of the finding (namely identifying potential therapeutic approaches) but without implying that this is already the clinical solution.

Line 17f: Again misleading. Common genetic associations per se cannot improve our understanding and treatment of human disease. They are a starting point, but so would be expression analysis combined with TF binding prediction, which would also have identified ETS in the conditions studied here. Correct would be therefore: "Together, this highlights the potential of the armamentarium of functional genomics applied directly in humans to improve both the understanding and treatment of human disease"

Line 23ff. Sentence contextually correct, but language not scientific enough. Re-phrase.

Line 35: Thi expression of "one million bases" needs to be optimized. Here, to mention a specific border is not correct, because the gene could even be in trans. Make a more applicable statement

Figure 1A: The upper schema should already contain the information about the gene locations, this

should not be introduced by the IBD GWAS panel already showing first data. Probably even better, make each of the three parts of current 1a into individual panels 1a-c. Also easier to reference in the text.

Line 53: Enhancer information is now introduced prior to the Hi-C data which in the figure is located above of the enhancer CHIP-seq information. Change order as data is described in the text.

Extended Data Fig. 1: several things need improvement: Panels b&c: introduce a schema, how the three different plots have been generated, from which dataset(s), why categorized into IBD, monocytes naive or LPS. Why is there no PP3 and PP4 for IBD? Why are the dots for monocytes LPS circles not at the same location? What does this actually mean for the overall results? In the extended figure legend, it would be probably good to introduce in more detail, what was really done. Main text is too short. This could be probably combined with a schema explaining what was done and what is shown, for example as panel a and then show the panels presented so far. What is meant here with co-localisation analysis and how was it done? How is the term "independent signals" defined in this context?

Line 63ff: Avoid terms like "interestingly". It is not explained, why this should be interesting. Provide further evidence that this is indeed the case (other datasets, additional experimental approach) and that the difference is real.

Line 66: Does this include both variants, the strong eQTL and the weaker eQTL, and if so, how do the authors now distinguish between the two?

Extended Data Fig.2: (see also question to line 66. In Panel a, two peaks are shown, are these associated with the two different eQTL loci?

Line 72: The terms M1 and M2 should be avoided as they are outdated (despite the fact that they are unfortunately still used), term could be IL-4-driven and IFN-driven macrophages, which would better reflect the stimulatory direction of these macrophage activation. Similarly, avoid the term polarization, use activation, as polarization would indicate only two major directions of activation, which is disproven for macrophages.

Line 145ff: TNF could be assessed by measuring transcriptional regulation. An experimental attempt should be made, since it is one of the major therapeutic targets in many autoimmune diseases and it would be good to know, whether it is also under ETS2 control.

Fig 2D: Provide quantification and statistics on the Western Blot data from the three donors. If too high variance: Add additional donors. Important finding.

Line 202ff and discussion: Albeit the reviewer completely agrees with this statement, it would also be good to provide at least a speculative statement in the discussion, under which conditions elevated ETS2 expression might have a benefit for an individual rather than being a burden when it comes to the development of autoimmune diseases. Further, it would be of interest to the readership to put this also into the perspective of the increasing number of patients during the last 50-80 years, which certainly is not explained by genetics alone. In other words, while genetics might favor such development, secondary environmental triggers are major drivers of disease and those individuals prone to react more strongly have the highest chance to develop disease.

Fig 3e.: This is a very exciting use of existing data from CD patients. Is there any additional dataset that could be used for computational validation? In addition, to exclude that this signature is not just the baseline activation signature of macrophages the authors need to also show that the signature is not enriched in every inflammatory condition. For example, what about monocytes/macrophages from acute infections, tumor-associated macrophages. This comparison would strengthen the overall results.

Line 223: The reviewer agrees that focusing on IBD first was the right thing to do. Nevertheless, since the findings could be very far-reaching, this reviewer strongly suggests to provide information for one additional autoimmune disease implicated at this particular locus. Either, the effect is specific to IBD or, it is an even broader concept for this particular transcription factor.

Line 250ff: The sentence about the ENCODE project is more to be seen as an information for the reviewing process. It can be deleted. Not necessary for this project.

Line 253: This reviewer thinks to remember that the referenced manuscript did not have 64 conditions, but less. Please check again and correct if necessary.

Line 275: It must state: "To further elucidate the mechanisms", because as presented, it seems that the metabolic and transcriptional regulation by ETS2 are independent and therefore, the term "further" would be correct.

Line 276ff and the whole last result section: The strong statement about no anti-ETS antibody working for ChIP-seq and then coming back that one worked for CUT&RUN is too dramatic. Make it more simple and more neutral scientific language. In fact, the discussion about no availability of any anti-ETS antibody for ChIP-seq could be avoided as follows: "As no anti-ETS antibody for ChIP-seq approaches could be identified, we applied two approaches to define ETS-driven transcriptional and functional programs on a global scale: 1. Co-expression, 2. CUT&RUN, not requiring fixation. " This reviewer would think that this would have less "drama".

Line 289ff: In the text, the authors state percentages, in the figure absolute genes counted. Link figure and text better. As provided, the reader needs to start to think, where the 48% and the 50% are derived from. Make it easier for the reader to link figure and text.

Line 293ff: What exactly is meant with the statement: "... rather than being solely attributable to differences in inflammation". Not clear to this reviewer.

Line 296: Avoid words such as "Interestingly", "Intriguingly", "Clearly". All interpretations.

Lines 296: The findings concerning binding of ETS2 to its own enhancer is of importance and interest, but the suggested feed-forward loop is speculative. Either erase such statements, at least move them to the discussion, or show experimentally that this is the case. The authors do have all tools at hand, to provide experimental evidence for such a feed-forward loop.

Line 300: The authors link all their findings to chronic inflammation. Some of the experimental conditions are most likely acute inflammatory conditions (short term activation of monocyte-derived macrophages via signals related to chronic inflammation). Can the authors make an attempt to address the role of ETS2 in acute inflammation or alternatively, make a statement about limitations of the study, suggesting that the role of ETS2 throughout the process of induction and maintenance of inflammation (early phases, late phases, chronic phases) requires additional work. E.g. what are the triggers that are necessary for a patient to acquire IBD beyond having a genetically determined elevation of ETS2, which by itself probably does not yet trigger IBD.

Fig 5f: This seems almost too good to be true. A better way for the reader to see, whether there was an almost 100% overlay would be to show FC/FC or ratio/ratio plots comparing the DE genes after chr21q22 deletion with the MEK inhibition. Also here, one can provide percentages of similar regulation, opposite regulation and strength differences of regulation. Please provide such information.

Lines 343ff: This reviewer is not arguing that the MEK inhibition experiments are not intriguing and promising, but since upstream regulators will have additional gene programs that are also

changed, it would be necessary to show the differences between upstream blockade (MEK inhibitor) and ETS2 blockade (as presented here by KO approaches) to see, whether there might be already off targets to be recognized. For precision medicine approaches, this would be similarly important and needs to be addressed, at least by providing a computational analysis predicting for MEK-related, ETS2 independent effects on macrophage function. The authors actually discuss the limitations of MEK inhibitors in the discussion, but it could be a very strong argument for the approach that even such potentially toxic effects are to be discovered by the chosen functional genomics approach.

Line 366f: This is another generalizing statement that is neither correct nor necessary. Even if this might be correct for the current SNP, it is not correct overall. For example, change in gene expression itself might be of great value for patient stratification irrespective of linking the patterns to disease mechanisms. Avoid such general opinion statements. The manuscript is better without.

Lines 375ff: The discussion about the allele frequency and its origins is interesting, but it does not help to understand, why autoimmune diseases show such a strong increase during the last decades. The authors should replace this nice intellectually stimulating discussion about origin with more solid discussion about the potential reasons that - in addition to the existing genetics - are most likely the major environmental drivers. This would be more important for developing tangible strategies for individuals at higher risk.

Line 389f: This is a very strong statement. If the authors want to claim that this locus confers susceptibility to multiple inflammatory diseases, then more data and results (as already mentioned above) about other autoimmune diseases throughout the manuscript is required. Otherwise focus on IBD here as well.

Line 390ff: Yet another very strong statement, but the authors should recognize that attempts to explain SNPs by other means (e.g. generating animal models with the same genetic variant) have been successfully performed before. The news here is that it is now presented solely in human studies, but the authors should either here or at the beginning of the introduction give credit to those who have done similar work in animals. Actually, this would also be an option to reduce the subliminal criticism of GWAs at some places in the manuscript. While the reviewer agrees that GWAS information urgently needs to be linked to work as presented here, a scientific original paper does not have to criticize (even if only subliminal) the GWAS approach and field. Not necessary (albeit correct).

Line 537: 100 Genomes phase 3 needs reference.

Line 541: eQTL Catalog: needs reference

Line 546: "... previously". Code needs to be provided.

Line 547: Downloading data is nice, but describing the computational approaches performed with the data is better. Authors need to be more precise. Overall: Authors are requested to provide ALL CODE for ALL analyses in one repository space (authors can choose), so that FAIR principles are respected (particularly R = reproducibility!). The links provided do not provide this information yet and it is therefore not possible for the reviewer to judge, whether reproducibility is following FAIR principles.

Line 569: the word "genes" is missing.

Lines 634: Accessibility path according to FAIR principles to these data is also not provided. This reviewer does not ask for open data, only for the accessibility path. (see also comment below).

Line 870: This is not a title for a methods subheading

Line 928: IMPORTANT! According to European GDPR regulations, human data cannot be uploaded to GEO anymore without breaching European law. The authors should use EGA instead and provide a clear accessibility path to the data.

Referee #2 (Remarks to the Author):

There is a lack of effective treatments for autoimmune/inflammatory diseases. Genetics has been shown to drastically improve the successful identification of novel drugs. However, the use of the vast amount of genetic data generated for the past 20 years in drug discovery programmes has been limited, due to a lack of understanding on the mechanisms by which genetic variants predispose to disease and the genes they affect. This currently remains a major challenge. Stankey et al address this critical challenge with an array of functional genomics, molecular biology, and immunology assays to study a risk locus on 21q22 that has been shown to be associated with several autoimmune diseases including IBD and AS. The authors robustly linked the risk locus to a causal gene, ETS2 (although this had already been suggested by previous CHI-C and eQTL studies), and a single causal variant, describe the mechanism by which the causal variant affect expression of the causal gene, describe the role that this gene has in macrophage inflammatory responses, and propose drugs that could target these mechanisms. The data is convincing, and the study elegantly takes a GWAS locus from just genetic association to potential translation into patient benefit.

One criticism may be that the role of the ETS2 gene, and the pathways it regulates, could have been explored further in disease. All experiments were performed in monocytes isolated from blood from healthy volunteers; could the authors explain if they would expect to see the same results in monocytes isolated from the affected tissues in IBD (intestine), AS (joints) and PSC (liver) patients? The only attempt to look at these mechanisms directly in disease is an interrogation of differential expression of genes in the ETS2 pathway from public datasets, but more details are needed to evaluate how robust this data is; how many patients were included in the transcriptomic analyses used?

The authors found that ETS2 genes were enriched for IBD risk loci. This is very interesting, and this reviewer wonders if this data could have been taken further, by performing a "ETS2 genetic risk score" or looking back at patients carrying genetic variants in this pathway to identify clinically meaningful monocyte-driven disease subgroups (would drugs targeting this pathway be more efficient in these patients?).

Referee #3 (Remarks to the Author):

This paper establishes a mechanism by which a common haplotype linked with a spectrum of autoimmune diseases can lead to functional disease phenotypes. The authors identified a SNP located in a novel enhancer of ETS2 that increases both PU.1 binding and downstream ETS2 transcription. They show that ETS2 is both necessary and sufficient for inflammatory responses in human macrophages and that ETS2 signaling is central to inflammatory bowel disease (IBD) pathogenesis. They identified a drug that targets upstream regulators of ETS2 and can abrogate ETS2 driven inflammation phenotypes in primary patient samples.

The findings are potentially important but also raise several questions. Specifically, the authors show there is a correlation between the risk haplotype and higher levels of ETS2 expression. However, the authors also note that the risk haplotype being studied is incredibly common (in ~75% of Europeans and ~90% of Africans). It remains unclear how the identified SNP leads to disease outcomes in some but not others.

Comments:

The authors should examine whether ETS2 expression is higher in individuals carrying the risk allele or is elevated specifically in patients with IBD symptoms. A critical question is whether ETS2 expression be used as indicator of whether or not someone with a risk haplotype exhibits IBD symptoms?

Line 105 PU.1 can bind to heterochromatin... Not sure that this indeed has been proven.

Figure 1g. Could the authors please show cis elements associated with genome-wide PU.1 occupancy? This is to validate the quality of the ChIP-seq data.

Could the authors please include ATAC-Seq data to confirm allelic differences in chromatin accessibility?

Figure 1i. Could the authors please show the patterns for all four donors?

Figure 2. It would be nice to perform a scRNA-seq analysis to examine whether subsets of populations were more affected upon removal of ETS2 expression. Skewing of the populations remains a bit of a worry in bulk RNA-Seq analysis. A more detailed analysis seems necessary. Same for the overexpression data.

Could the authors please perform ATAC-Seq analysis on wild-type cells and cells depleted for ETS expression?

The motif analysis associated with the CUT&RUN experiments should reveal p-values and other elements associated with ETS occupancy.

Tornado plots should be presented for CUT&RUN data in parallel with enhancer and promoter marks.

Referee #4 (Remarks to the Author):

A disease associated gene desert orchestrates macrophage inflammatory responses via ETS2

Stankey et al.

Stankey et al highlight an example of how a common genetic association can improve the understanding and reveal new therapeutic treatments for inflammatory / autoimmune disease. The paper focuses on the chr21q22 locus and the major (risk) allele haplotype which is associated with 5 different inflammatory diseases. They identify a monocyte/macrophage specific enhancer which contains the SNP rs2836882. This enhancer elevates ETS2 expression and the SNP colocalizes with a PU.1 binding site.

Loss of function for ETS2 results in a decrease in phagocytosis and a decrease in oxidative bursts. RNA-seq on ETS2 edited vs unedited inflammatory macrophages from multiple donors revealed a reduction in cytokines, chemokines, secreted effector molecules, cell surface receptors, pattern recognition receptors and signaling molecules with ETS2 loss of function. Gene-set enrichment analysis showed a reduction in macrophage activation, pro-inflammatory cytokine production, phagocytosis, ROS production following ETS2 disruption.

ETS2 is necessary and sufficient for inflammatory responses in human macrophages and acts as a master regulator. This is highlighted through the ETS2 over-expression experiments where it

drives the pathways identified in the ETS2 loss of function RNA-seq experiment in a dose-dependent manner. Furthermore, they showed that over-expressing ETS2 in resting macrophages induced a transcriptional state that closely resembled disease macrophages in Crohn's disease. They also showed that ETS2 target genes were more strongly enriched for IBD associated loci than almost all previously implicated pathways, suggesting that ETS2 signaling in macrophages plays a fundamental role in IBD pathogenesis.

The authors performed CUT&RUN to identify ETS2 binding sites across the genome and they found that ~50% of genes dysregulated by ETS2 over expression or disruption contained an ETS2 binding site, suggesting ETS2 directs macrophage responses at a transcriptional level.

The authors next used the NIH LINCS database to identify drugs that might modulate ETS2 activity (i.e. mimic the transcriptional effect of disrupting ETS2 in inflammatory macrophages). For example, MEK inhibitors are the most common class and an upstream regulator of ETS2. The authors treated differentiated monocytes with MEK inhibitor and saw an anti-inflammatory activity that phenocopied the effect of disrupting ETS2 or deleting the chr21q22 enhancer. Treating IBD mucosal biopsies with MEK inhibitor reduced the secreted cytokine levels to those which were similar to Infliximab.

Overall, this study presents a very thorough GWAS locus dissection / variant-to-function story at the ETS2 locus. The authors have linked a GWAS risk variant associated with multiple autoimmune and inflammatory diseases to effects on TF binding, enhancer chromatin state, and ETS2 expression. Then, they link ETS2 to changes in downstream gene expression, metabolic state, and macrophage phenotypes.

The overall experimental rigor is high, including challenging genetic experiments in primary human cell and explant models (e.g., MPRA transfected into inflammatory macrophages from multiple donors, PU.1 ChIP-seq from multiple heterozygous donors, ETS2 mRNA over-expression in primary human macrophages, MEK inhibition in human gut explants).

The paper is also very clear and well written — an enjoyable read!

One consideration around the novelty of this study is that the role of ETS2 in macrophages has been extensively studied previously. Several other individual findings reported in this paper are similar to previous observations, such as the importance of ETS2 in cytokine production in macrophages, that IBD GWAS genes are enriched for ETS2 motifs, that treatment of an experimental colitis mouse model with MEK inhibitors reduces severity, etc. The authors here have contributed many new excellent experiments to put together these pieces into a complete story about how the human genetic variant affects ETS2, including new evidence in primary human cell models. The authors could perhaps cite some of this prior work more thoroughly where applicable, and provide some additional explanations where relevant to highlight the new findings about ETS2 in the Discussion.

I do not have any other major comments about the paper. Some minor comments + corrections below:

Corrections:

1. Page 3, line 36 "which may lie up to one million bases away". Technically but rarely they can be located even further. I would suggest "which may lie up to millions of bases away"
2. Page 3, line 39 "none have led to new therapies". This is not correct, I think. What about TYK2, IL12/IL23R? RORC and JAK2 have also motivated drug development programs, although I am not up to date on the status of these programs.

3. For language reasons, suggest using “regulator” instead of “master regulator”
4. Extended data Figure 5. Labels for b,c and d have been swapped around (please fix)

Minor comments:

1. Fig 2: Could the authors please show some evidence for the ‘loss of function’ for ETS2 at a RNA or protein level (Sequencing data, expression data or Western blot?)
2. Fig 3f. Legend says “enrichments”, but figure shows p-values, not degree of enrichment. P-values depend on gene set size. Would be helpful for figure to show both enrichment and p-value
3. Could you clarify — Do the human gut explant models include inflammatory macrophages?
4. Regarding Figure 4i, is it possible that the rs2836882 variant falls within the ETS2 motif? The authors have said in Fig. 1 that the SNP lies outside the PU.1 motif but it would be good to have a figure showing where the SNP lies in relation to the PU.1 and ETS2 motif.
5. The link between the GWAS variant and cellular phenotypes could be made stronger by for example CRISPR editing the variant or by studying some of the cellular phenotypes in macrophages with different genotypes

Author Rebuttals to Initial Comments:

Referee #1 (Remarks to the Author):

The manuscript by Stankey et al., entitled "" describes a functional genomics approach in humans to functionally and causally link a genetic alteration previously associated with several autoimmune diseases including IBD to the transcription factor ETS2, thereby elucidating the function of ETS2 in monocyte/macrophages in these diseases and finally also provide potential therapeutic strategies based on the biology of this TF. Overall, this manuscript is a conceptual advance, it illustrates how human genetics should move forward from descriptive GWAS and polygenic risk score science to address causation and function in a combined experimental and computational approach, which this author would like to term "functional genomics". Overall, the data, the approaches and the execution are of very high quality, yet, there is some room for improvements, which this reviewer has listed below (with some of the comments are more editorial):

Thank you for these positive comments and for your detailed review of our manuscript. We have provided point-by-point responses below with relevant text excerpts for ease of reviewing.

Comments with the request to be addressed by the authors:

Title: The term "gene desert" is never really explained by the results. Why did the authors choose this word instead of making the point that mainly regulatory elements in intergenic regions of the genome are linked to diseases? Why not naming the identified function of the region instead of stating what it isn't (namely a gene region)?

Thank you for this question and apologies for not qualifying the term "gene desert" in the text. We chose this term because (1) it has been widely used in the scientific literature since the early 2000s, (2) we thought it would be readily understood by a non-specialist audience, and (3) this is what the locus was described as before we showed it contained a distal *ETS2* enhancer. Moreover, we wanted a title that would trigger curiosity in specialist and non-specialist readers alike – and thus encourage them to continue reading – and felt that the inherent conflict associated with a gene desert playing an important biological role should arouse interest.

In response to your comment, we have spent some time considering whether a more functional description of the locus would improve the title e.g. "distal enhancer". However, on balance, we felt that any gain in specificity would be outweighed by making the title less understandable to, and potentially less likely to trap the attention of, a non-specialist audience. We have discussed this point with the editor and have now qualified the use of this term in the main text:

"One notable example is an intergenic region on chr21q22, where the major (risk) allele haplotype has been independently associated with five different inflammatory diseases³⁻⁶. Such regions, which were originally termed "gene deserts" due to their lack of coding genes, frequently harbour disease associations but are not well understood. To determine whether a common pathogenic mechanism might be present, we performed colocalisation analyses and confirmed that the genetic basis for all of the diseases was the same, meaning that a single causal variant – and thus a shared molecular effect – was responsible for every association (**Fig.1a, Extended Data Fig.1**)."

Line 5: The statement about genetics is actually wrong: Genetics did not address this. The approach chosen by the authors is "functional genomics". As correctly stated by the authors GWAS itself (the tool of genetics) was incapable of being informative concerning the role of the SNP in question. The authors should change to "functional genomics"

We have now changed the description of our approach from "genetics" to "functional genomics":

Abstract: "Here we show how functional genomics could address this challenge... Together, this illustrates the power of functional genomics, applied directly in humans, to elucidate mechanisms of immune-mediated disease and potential therapeutic opportunities."

Line 15: "identified" instead of "identify"

We have now made this change and have also changed "validate" to "validated" in the same sentence for consistency:

"Using a database of cellular signatures⁸, we identified drugs that could potentially modulate this pathway and validated the potent anti-inflammatory activity of one class of small molecules *in vitro* and *ex vivo*."

Line 15: Considering and as discussed by the authors in their discussion section that MEK inhibition will most likely not be easily applicable clinically, this statement should be altered to better reflect the principle of the finding (namely identifying potential therapeutic approaches) but without implying that this is already the clinical solution.

We have now changed this sentence to better reflect the principle of our finding:

"Together, this illustrates the power of functional genomics, applied directly in humans, to elucidate mechanisms of immune-mediated disease and potential therapeutic opportunities."

Line 17f: Again misleading. Common genetic associations per se cannot improve our understanding and treatment of human disease. They are a starting point, but so would be expression analysis combined with TF binding prediction, which would also have identified ETS in the conditions studied here. Correct would be therefore: "Together, this highlights the potential of the armamentarium of functional genomics applied directly in humans to improve both the understanding and treatment of human disease"

This sentence has now been re-written – integrating the reviewer's suggestion, our response to their previous comment, and the word limit of the abstract:

"Together, this illustrates the power of functional genomics, applied directly in humans, to elucidate mechanisms of immune-mediated disease and potential therapeutic opportunities."

Line 23ff. Sentence contextually correct, but language not scientific enough. Re-phrase.

We have now re-phrased this sentence:

"This high failure rate is principally due to a lack of efficacy⁹ and reflects our incomplete understanding of disease mechanisms. Genetics provides a unique opportunity to address this, with hundreds of regions of the human genome now directly linked to the pathogenesis of one or more autoimmune or inflammatory disease¹⁰."

Line 35: The expression of "one million bases" needs to be optimized. Here, to mention a specific border is not correct, because the gene could even be in trans. Make a more applicable statement

The need to amend this sentence was also highlighted by one of the other reviewers. We have used their suggestion to re-phrase the sentence:

"Most non-coding variants are thought to affect gene regulation¹⁴, but the need to identify the causal gene(s) – which may lie millions of bases away – and the causal cell-type(s), which may only express the causal gene under specific conditions, have hindered attempts to discover disease mechanisms."

Figure 1A: The upper schema should already contain the information about the gene locations, this should not be introduced by the IBD GWAS panel already showing first data. Probably even better, make each of the three parts of current 1a into individual panels 1a-c. Also easier to reference in the text.

Thank you for this helpful suggestion, which also enabled us to present evidence that a shared causal variant – and thus a common molecular effect – accounts for susceptibility to all of the associated diseases. We have reproduced the new panels of Figure 1a-c below the next response.

Line 53: Enhancer information is now introduced prior to the Hi-C data which in the figure is located above of the enhancer CHIP-seq information. Change order as data is described in the text.

The new panels in Fig.1a-c, which we have reproduced below, should hopefully address this comment and are now consistent with the order in which the data are introduced in the text.

Legend:

a. Manhattan plots of inflammatory disease associations at a common chr21q22 haplotype. Red points indicate 99% credible set in IBD. Colocalisation results are shown for each disease compared to IBD. PP.H3: posterior probability of independent causal variants; PP.H4: posterior probability of shared causal variant.

b. H3K27ac ChIP-seq at the disease-associated locus in immune cells. IBD GWAS results shown for orientation.

c. Manhattan plot of *ETS2* eQTL in resting monocytes, with colocalisation results between eQTL and IBD association. Promoter-capture Hi-C data showing physical interactions of the disease-associated locus in macrophages.

Extended Data Fig. 1: several things need improvement: Panels b&c: introduce a schema, how the three different plots have been generated, from which dataset(s), why categorized into IBD, monocytes naive or LPS. Why are the dots for monocytes LPS circles not at the same location? What does this actually mean for the overall results? In the extended figure legend, it would be probably good to introduce in more detail, what was really done. Main text is too short. This could be probably combined with a schema explaining what was done and what is shown, for example as panel a and then show the panels presented so far.

We apologise that this figure was confusing. As part of our response to one of your later comments, we have considerably expanded our colocalisation analyses to incorporate many more monocyte/macrophage eQTL datasets and other chr21q22-associated diseases. This necessitated a different format for presenting the results – both due to space limitations and the lack of clarity highlighted. We think the revised figure is much clearer, and have added additional text and methods to explain the approach and reference the datasets used:

“To determine whether a common pathogenic mechanism might be present, we performed colocalisation analyses and confirmed that the genetic basis for all of these diseases was the same, meaning that a single causal variant – and thus a shared molecular effect – was responsible for every association (**Fig.1a, Extended Data Fig.1**).”

“Using publicly-available data from human monocytes, including promoter-capture Hi-C and eQTL datasets (**Methods**), we found that the disease-associated locus physically interacts with the promoter of *ETS2*, the most distant of the candidate genes (located 290-kb away), and that the risk haplotype correlates with higher expression of *ETS2* (**Fig.1c, Extended Data Fig.1**). Indeed, increased *ETS2* expression in monocytes/macrophages – either at rest or during early exposure to bacteria – was predicted to have the same genetic basis as the risk of inflammatory disease (**Fig.1c, Extended Data Fig.1**).”

Extended Data Figure 1. Colocalisation between genetic associations at chr21q22.

a. Example comparison of genetic associations at chr21q22: IBD and *ETS2* eQTL in unstimulated monocytes. Plot adapted from locuscomparer. **b.** Boxplot depicting *ETS2* expression stratified by rs2836882 genotype in unstimulated monocytes⁵⁶. **c.** Radar plot of representative colocalization results for the indicated genetic associations compared to IBD. Posterior probability of independent causal variants, PP.H3, dark blue; posterior probability of shared causal variant, PP.H4, light blue. PP.H4 > 0.5 was used to call colocalisation (denoted by dashed line). Labels are coloured according to class of data (indicated in the key). Asterisks denote colocalisation. Data sources are: IBD³, PSC⁵, AS⁴, Takayasu Arteritis⁶, BLUEPRINT⁵⁸, Fairfax⁵⁶, Quach⁵⁷, Nedelec⁵⁹, Alasoo⁶⁰.

Methods: "Colocalisation analyses were performed comparing the chr21q22 IBD association with summary statistics from other chr21q22-associated diseases³⁻⁶ and monocyte/macrophage eQTLs⁵⁶⁻⁶⁰ to determine whether there was a shared genetic basis for these different associations. This was performed using coloc v5.2.0⁶¹ using a posterior probability of H4 (PP.H4.abf) > 0.5 to call colocalisation."

What is meant here with co-localisation analysis and how was it done? How is the term "independent signals" defined in this context? Why is there no PP3 and PP4 for IBD?

Colocalisation analysis is a genetic test to determine whether two potentially related phenotypes share a common genetic basis at a particular locus. It is widely used in genetics and was developed by one of our co-authors (Chris Wallace). PP.H3 refers to the Bayesian posterior probability that the two phenotypes have different causal variants (i.e. are independent) and PP.H4 refers to the posterior probability that there is a shared causal variant. The IBD locus did not have a PP3 or PP4 since this was the reference association to which the other associations were being compared. We have elaborated on the details of this analysis in the methods, which also contains a link to the code required to reproduce the analysis (<https://github.com/chr1swallace/ibd-ets2-analysis>).

Fig.1 legend: "Manhattan plots of inflammatory disease associations at a common chr21q22 haplotype. Red points indicate 99% credible set in IBD. Colocalisation results are shown for each disease compared to IBD. PP.H3: posterior probability of independent causal variants; PP.H4: posterior probability of shared causal variant."

Methods: "Colocalisation analyses were performed comparing the chr21q22 IBD association with summary statistics from other chr21q22-associated diseases³⁻⁶ and monocyte/macrophage eQTLs⁵⁶⁻⁶⁰ to determine whether there was a shared genetic basis for these different associations. This was performed using coloc v5.2.0⁶¹ using a posterior probability of H4 (PP.H4.abf) > 0.5 to call colocalisation."

Line 63ff: Avoid terms like "interestingly". It is not explained, why this should be interesting. Provide further evidence that this is indeed the case (other datasets, additional experimental approach) and that the difference is real.

We have now considerably expanded our eQTL colocalisation analysis, adding 3 additional naïve monocyte/macrophage datasets and 9 other monocyte/macrophage datasets under various stimulation conditions. The results are discussed in detail after the next response and are shown in Extended Data Fig.1 (above). We have also removed the word "interestingly" and re-written this sentence:

"Indeed, increased *ETS2* expression in monocytes/macrophages – either at rest or during early exposure to bacteria – was predicted to have the same genetic basis as the risk of inflammatory disease (**Fig.1c, Extended Data Fig.1**)."

Line 66: Does this include both variants, the strong eQTL and the weaker eQTL, and if so, how do the authors now distinguish between the two?

This is a good question which made us revisit (and ultimately extend) the source data we used for the eQTL analyses. In turn, this has generated some new and important insights.

The purpose of our original experiment was to identify the gene regulated by the enhancer detected in H3K27ac ChIP-seq data (which directly overlaps the chr21q22 disease-associated variants – shown in Fig.1b). Guide RNAs were therefore designed based on the co-ordinates of the monocyte enhancer marks, rather than the location of the eQTL variants. On further examining the coordinates of the *ETS2* eQTLs (from Fairfax et al. and several other datasets) we found that the deleted region contains the entire haplotype of SNPs that are an *ETS2* eQTL in resting monocytes, but not the SNPs that are an eQTL following 24h LPS stimulation. Consistent with this, none of the 99% credible SNPs from the 24h LPS stimulation eQTL overlap the 99% credible SNPs from the IBD association.

However, to be certain that there was only overlap in resting monocytes, we decided to analyse additional eQTL datasets – especially since most transcriptional responses to stimulation occur early (within 4-8 hours) and thus 24 hours is a relatively late timepoint. This yielded two important results. First, we confirmed – in several independent monocyte/macrophage datasets – that an *ETS2* eQTL in resting cells overlaps the coordinates of the disease-associated enhancer. Indeed, colocalisation analysis revealed that this eQTL shares a common genetic basis with the risk of inflammatory disease (shown in Extended Data Fig.1 above). Second, we found that the disease-associated enhancer also contains the 99% credible SNPs from an *ETS2* eQTL in monocytes/macrophages during early exposure to bacteria or bacterial components (and that these similarly colocalised with inflammatory disease risk). There was no overlap with eQTLs following viral stimulation, nor during more prolonged exposure to cytokines or bacterial components. This suggests that this enhancer, which directly overlaps the pleiotropic disease association, regulates *ETS2* expression both at rest and during early responses to bacteria. We have updated the main text accordingly:

“Using publicly-available data from human monocytes, including promoter-capture Hi-C and eQTL datasets (**Methods**), we found that the disease-associated locus physically interacts with the promoter of *ETS2*, the most distant of the candidate genes (located 290-kb away), and that the risk haplotype correlates with higher expression of *ETS2* (**Fig.1c, Extended Data Fig.1**). Indeed, increased *ETS2* expression in monocytes/macrophages – either at rest or during early exposure to bacteria – was predicted to have the same genetic basis as the risk of inflammatory disease (**Fig.1c, Extended Data Fig.1**).”

Extended Data Fig.2: (see also question to line 66. In Panel a, two peaks are shown, are these associated with the two different eQTL loci?

This is an understandable question. Thanks to your previous suggestion, we now recognise that the disease-associated enhancer at chr21q22 functions in resting monocytes/macrophages and during early exposure to bacteria (based on colocalisation across multiple eQTL datasets). It was not possible to separate the resting and early activation eQTLs since their 99% credible sets completely overlap. In contrast, the eQTL in monocytes following 24h LPS activation lies outside of this region.

Importantly, the “peak-valley-peak” appearance of the H3K27ac marks – to which this comment refers – is a recognised feature of active regulatory elements and is not indicative of separate eQTL loci. The valley between the peaks represents an accessible nucleosome-free region, which is flanked by modified histones on either side (this is described in “Peak-valley-peak pattern of histone modifications delineates active regulatory elements and their directionality”. Pundhir et al. *Nucleic Acids Research* 2016. doi: 10.1093/nar/gkw250).

Line 72: The terms M1 and M2 should be avoided as they are outdated (despite the fact that they are unfortunately still used), term could be IL-4-driven and IFN-driven macrophages, which would better reflect the stimulatory direction of these macrophage activation. Similarly, avoid the term polarization,

use activation, as polarization would indicate only two major directions of activation, which is disproven for macrophages.

We have edited the manuscript to incorporate these suggestions:

“This model, termed “TPP”, was designed to mimic a chronic inflammatory environment¹⁷, and better recapitulates the state of patient-derived monocytes/macrophages than classical IFN γ -driven or IL-4-driven models (ref.18 and **Extended Data Fig.2**).”

We have also changed all uses of the word polarization to activation:

Figure 1 legend: “**d.** Schematic of experiment to functionally characterise the chr21q22 locus in monocyte-derived macrophages activated under chronic inflammatory (“TPP”) conditions. **e.** Histograms depicting the expression of *ETS2*, *BRWD1*, and *PSMG1* during inflammatory macrophage activation, measured using PrimeFlow RNA assays.”

“We therefore first used a “guilt-by-association” approach to identify genes that were co-expressed with *ETS2* across 67 human monocyte/macrophage activation conditions (comprising 28 different stimuli and several durations of exposure)¹⁷.”

Methods: “Human monocyte/macrophage gene expression data files (n=314) relating to 28 different stimuli with multiple durations of exposure (collectively comprising 67 different activation conditions) were downloaded from Gene Expression Omnibus (GSE47189) and quantile normalised... Gene set variation analysis⁶⁴ (using the *GSEA* package in R) was performed to identify the activation condition that most closely resembled CD14+ monocytes/macrophages from active IBD – using disease-associated lists of differentially-expressed genes⁶⁵.”

Methods: “Genes co-expressed with *ETS2* across 67 human monocyte/macrophage activation conditions (normalised data from GSE47189) were identified using the *rcorr* function in the *Hmisc* package in R.”

Line 145ff: TNF could be assessed by measuring transcriptional regulation. An experimental attempt should be made, since it is one of the major therapeutic targets in many autoimmune diseases and it would be good to know, whether it is also under *ETS2* control.

We agree that it is important to know whether TNF α is under *ETS2* control. At this point in the manuscript (line 145), we had not yet introduced the whole transcriptome RNA-sequencing experiments that we went on to perform – both in *ETS2*-disrupted and *ETS2*-overexpressing macrophages. TNF α was measured in those experiments, and later in the manuscript we provide evidence that TNF α is likely to be under *ETS2* control. For example, in Figure 3 we show that both TNF α mRNA and protein are upregulated by *ETS2* overexpression. All of the genome-wide transcriptional changes we detect are presented in Supplementary Tables 1 and 2:

c. Cytokine secretion following *ETS2* overexpression. Plot shows relative cytokine concentrations in macrophage supernatants (*ETS2* relative to control) following transfection with 500ng mRNA (n=11).

e. Enrichment of a disease-associated inflammatory macrophage gene signature, derived from single cell RNA-seq of Crohn's disease intestinal biopsies, in *ETS2*-overexpressing macrophages (relative to control; top). Heatmap of leading-edge genes showing relative gene expression in *ETS2*-overexpressing macrophages versus control (500ng mRNA; bottom).

Fig 2D: Provide quantification and statistics on the Western Blot data from the three donors. If too high variance: Add additional donors. Important finding.

Thank you for this suggestion. We have now performed densitometry on the Western blots (adding four additional donors) and present these data with comparative statistics in a revised Figure 2d (replacing the single gel image that was there previously). Of note, during this process we realised that beta actin may not be the best loading control since it is significantly upregulated by *ETS2* overexpression. All of the data have therefore now been re-generated using vinculin as a loading control instead. The raw gel images are now provided as a Supplementary Figure per Nature requirements:

d

d. Production of ROS by *ETS2*-edited and unedited inflammatory macrophages (measured in relative light units; left). Data representative of one of six donors. Expression of selected NADPH oxidase components in *ETS2*-edited and unedited macrophages quantified by Western blot densitometry (right, $n=7$, data represent mean \pm SEM). For gel source data, see Supplementary Figure 1.

Line 202ff and discussion: Albeit the reviewer completely agrees with this statement, it would also be good to provide at least a speculative statement in the discussion, under which conditions elevated *ETS2* expression might have a benefit for an individual rather than being a burden when it comes to the development of autoimmune diseases.

Thank you for this suggestion, which we have now incorporated into a re-written discussion. Importantly, the new data we have added enable us to form a specific hypothesis about this point, since the risk of inflammatory disease was shown to have the same genetic basis (i.e. be driven by the same genetic variant) as the increased expression of *ETS2* during early responses to bacteria (Extended Data Fig.1, above). Moreover, we have now shown that the transcriptional state of macrophages during serious bacterial infection is significantly enriched for genes induced by *ETS2* overexpression (Fig.3f, shown after a later response below). This is not simply a generic activation response since comparable enrichment was not observed in either dataset for responses to influenza

A, for example. This suggests that increased *ETS2* expression may have a beneficial role following initial exposure to bacteria. This is a critical time window in which effective pathogen killing and clearance can avert disseminated infection, and would thus represent a strong balancing selective pressure to maintain the inflammatory disease risk allele in the population. This may explain why this allele has not been selected against in the >500,000 years since it first appeared in humans (despite predisposing to multiple diseases that occur in young adults and are life-limiting) and would fit with the observation that this allele is commoner in populations with higher rates of tuberculosis and other endemic bacterial diseases:

“These discoveries also provide clues to the gene-environment interactions at this locus. For example, the shared genetic basis of inflammatory disease risk and increased *ETS2* expression during early exposure to bacteria highlights a situation in which *ETS2*-driven macrophage responses may be beneficial. This also suggests that *ETS2* may function in different phases of inflammation, consistent with *ETS2*-regulated genes being induced during serious bacterial infections. Swift bacterial killing upon first encounter is critical to avoid disseminated infection, and could provide a balancing selective pressure to maintain the chr21q22 ancestral allele in human populations. This may explain why this disease risk allele remains so common (frequency ~75% in Europeans, >90% in Africans) despite being an exceptionally old variant (>500,000 years) that was even polymorphic in archaic humans (**Extended Data Fig.9**).”

Further, it would be of interest to the readership to put this also into the perspective of the increasing number of patients during the last 50-80 years, which certainly is not explained by genetics alone. In other words, while genetics might favor such development, secondary environmental triggers are major drivers of disease and those individuals prone to react more strongly have the highest chance to develop disease.

Thank you for this comment. We have now mentioned the increasing rates of autoimmunity in the discussion, and highlighted the importance of environmental triggers in initiating disease in genetically-susceptible individuals. Our colocalisation analyses (discussed above) also provide clues to the gene-environment interactions that are likely to be involved at this locus, and specifically illustrate the point that stronger inflammatory reactions may be beneficial in some contexts but detrimental in others:

“This is particularly important since rates of autoimmunity continue to rise, yet how environmental factors trigger disease in genetically-susceptible individuals remains largely unknown. Here, by investigating a pleiotropic disease locus, we uncover a central regulator of human macrophage effector functions and a key pathogenic pathway that is potentially druggable. These discoveries also provide clues to the gene-environment interactions at this locus. For example, the shared genetic basis of inflammatory disease risk and increased *ETS2* expression during early exposure to bacteria highlights a situation in which *ETS2*-driven macrophage responses may be beneficial. This also suggests that *ETS2* may function in different phases of inflammation, consistent with *ETS2*-regulated genes being induced during serious bacterial infections. Swift bacterial killing upon first encounter is critical to avoid disseminated infection, and could provide a balancing selective pressure to maintain the chr21q22 ancestral allele in human populations. This may explain why this disease risk allele remains so common (frequency ~75% in Europeans, >90% in Africans) despite being an exceptionally old variant (>500,000 years) that was even polymorphic in archaic humans (**Extended Data Fig.9**).”

Fig 3e.: This is a very exciting use of existing data from CD patients. Is there any additional dataset that could be used for computational validation? In addition, to exclude that this signature is not just the baseline activation signature of macrophages the authors need to also show that the signature is not enriched in every inflammatory condition. For example, what about monocytes/macrophages from acute infections, tumor-associated macrophages. This comparison would strengthen the overall results.

Thank you for these helpful suggestions. To validate the result shown in Fig.3e, we identified another dataset in which single cell RNA sequencing was performed on intestinal myeloid cells from Crohn's disease (Chapuy et al. "Two distinct colonic CD14+ subsets characterized by single-cell RNA profiling in Crohn's disease". *Mucosal Immunology* 2019, PMID: 30670762). In that study, the authors identified two CD14+ myeloid subpopulations that were massively expanded in Crohn's and proposed to be pathogenic – one that had features of inflammatory monocytes (producing IL-23 and IL-1 β) and another had features of activated macrophages (producing TNF α). The marker genes for these disease-specific populations were combined and used as a gene set for fGSEA. This confirmed that ETS2 overexpression recapitulates the pathogenic phenotype of monocytes/macrophages in Crohn's disease:

Reviewer figure (not included in manuscript)

We then used a similar approach to address the reviewer's second point; whether this simply represents baseline activation and so might be enriched in most disease states. To do this, we identified publicly reported macrophage signatures from a range of different diseases. Importantly, these represented true disease signatures derived from human macrophages from patients with the respective conditions (rather than responses to *in vitro* stimulation). The signatures included:

Disease	Tissue source	PMID	Source (in paper)	Method	Category
Crohn's disease	Intestine	30670762	Cluster E and F markers	scRNAseq	Inflammatory disease (chr21q22-linked)
Takayasu arteritis	Blood	34671607	DEGs (up) in CD14+ monocytes (vs controls)	scRNAseq	Inflammatory disease (chr21q22-linked)
Ankylosing spondylitis	Blood	37388915	DEGs (up) in CD14+ monocytes (Table S3)	RNAseq (bulk)	Inflammatory disease (chr21q22-linked)
Ulcerative colitis	Intestine	31348891	DEGs (up) in m.monocyte subgroups (vs controls; Table S4)	scRNAseq	Inflammatory disease (chr21q22-linked)
Influenza A	Upper airway	34618691	DEGs (up) in macrophages (vs controls, Supp. File 4)	scRNAseq	Viral infection
COVID-19	Upper airway	34618691	DEGs (up) in macrophages (vs controls, Supp. File 4)	scRNAseq	Viral infection
Tuberculosis	Broncho-alveolar lavage	37470432	DEGs (up) in macrophages (active TB vs controls, Table S2)	scRNAseq	Bacterial infection

Bacterial sepsis	Blood	32066974	DEGs (up) in MS1 vs all monocytes (Table S2)	scRNAseq	Bacterial infection
Tumour-associated (multiple cancers)	Tumour (lung, colon, liver, breast, stomach, and pancreas)	34331874	DEGs (up) in IL4I1+ TAMs (cluster #6, Table S3B)	scRNAseq	Cancer
Tumour-associated (endometrial)	Tumour	30930117	DEGs (up) in endometrial TAMs vs endometrial tissue resident macs (Table S2)	scRNAseq	Cancer
Atherosclerosis	Atherosclerotic plaques	36190844	Foamy macrophage marker genes (Table S3)	scRNAseq	Atherosclerosis

DEGs (up), differentially-expressed genes (upregulated)
TAMs, tumour-associated macrophages
BAL, bronchoalveolar lavage

These signatures were used as gene sets for fGSEA to assess whether *ETS2* overexpression induced similar enrichment across all diseases. The results are shown in a new figure panel (Fig.3f). They demonstrate that *ETS2* overexpression does not simply cause baseline activation, but rather induces genes that are characteristic of chronic inflammatory disease and, to a lesser extent, responses to bacterial infection. In contrast, no significant enrichment was observed for tumour-associated macrophage signatures or macrophages during certain viral infections (e.g. influenza A).

The enrichment for signatures of bacterial infection is particularly interesting, since it mirrors the new genetic colocalisation results between the chr21q22 disease haplotype and *ETS2* eQTLs in response to bacteria / bacterial components (shown above and in Extended Data Fig.1). This provides clues to the evolutionary advantage that may have driven the persistence of the disease-associated variant in humans. This is now discussed further in the Discussion (in line with your previous request) and would fit with observations relating to the frequency of the variant in different populations, especially in regions where severe bacterial infection is a major contributor to infant mortality. Of note, the new Crohn's disease data in Fig.3f is the same as in the Reviewer Figure above – hence the Reviewer Figure is not included in the manuscript:

Legend: **f**. Enrichment of macrophage signatures, derived from patients with the indicated diseases (colour coded by category), in *ETS2*-overexpressing macrophages (relative to control). Numbers represent NES, dashed line denotes FDR 0.05. Crohn's disease signature is from a different study to that shown in **e**.

Line 223: The reviewer agrees that focusing on IBD first was the right thing to do. Nevertheless, since the findings could be very far-reaching, this reviewer strongly suggests to provide information for one additional autoimmune disease implicated at this particular locus. Either, the effect is specific to IBD or, it is an even broader concept for this particular transcription factor.

Thank you for this suggestion. We have now revised our manuscript to include several additional lines of evidence relating to other chr21q22-associated diseases.

First, to determine whether the independent disease associations at chr21q22 were all attributable to a common molecular effect, we performed a series of genetic colocalisation analyses and confirmed that there was a shared genetic basis – mediated by a single causal variant – for every chr21q22-associated disease (Fig.1 and Extended Data Fig.1, above).

Second, to determine whether the genetics of other chr21q22 diseases were also enriched for ETS2 pathway genes, we performed additional SNPsea enrichment analyses in Primary Sclerosing Cholangitis, Ankylosing Spondylitis and Takayasu Arteritis – similar to the analysis we performed using IBD-associated SNPs. We also included schizophrenia as a negative control since this had a similar number of SNPs to IBD (n=278) but is not thought to be auto-inflammatory. For this analysis, we collated lists of risk SNPs for each disease from previous GWAS studies. Genes tagged by these disease-associated genetic variants were then tested for enrichment in overlapping lists of ETS2 target genes – including those upregulated by *ETS2* overexpression, downregulated by *ETS2* disruption, or downregulated following chr21q22 deletion. Unlike schizophrenia, every chr21q22-associated disease showed significant enrichment in at least one of the ETS2-regulated gene lists. While the degree of significance was less striking than for IBD (likely due to the smaller SNP lists) this result shows that the genetics of every chr21q22-associated disease is enriched for genes regulated by ETS2. These new data are presented in Extended Data Fig.5:

Third, we have shown that overexpressing ETS2 in resting macrophages induces a transcriptional state that resembles the phenotype of monocytes/macrophages from several chr21q22-associated diseases, including blood monocytes from Ankylosing Spondylitis or Takayasu Arteritis and intestinal macrophages from ulcerative colitis and Crohn's disease (Figure 3f, above).

Fourth, to better understand how ETS2 contributes to macrophage heterogeneity within diseased tissues, we performed spatial transcriptomics in fixed liver tissue from PSC and healthy controls (in addition to analysing scRNA-seq data from Crohn's disease and controls). This showed that PSC is characterised by a large increase in the number of inflammatory macrophages, and that these are in close proximity to the bile duct cholangiocytes (the target of pathology in PSC). Moreover, in PSC, expression of ETS2-regulated genes was highest in macrophages closest to the cholangiocytes, consistent with a direct role in disease pathogenesis. These data are shown in Fig.5c-f and Extended Data Fig.7:

c. Spatial transcriptomics of PSC and healthy liver ($n=4$). Images show representative fields of view with cell segmentation and semi-supervised clustering results (InsituType). Legend indicates InsituType cell-types. Hep., hepatocyte; LSECs, liver sinusoidal endothelial cells. **d.** Average number of macrophages within a defined radius of a cholangiocyte. **e.** Distance from cholangiocytes to nearest macrophage. Data shown as Tukey box-and-whisker plot. Mann-Whitney test, two-tailed. Data in **d** and **e** represent 10,532 PSC and 13,322 control cholangiocytes. **f.** Scaled expression of ETS2-regulated genes in 21,067 PSC macrophages (excluding genes used to defined macrophage subsets) at defined distances from cholangiocytes. Data represent mean and 95%CI.

Extended Data Fig.7

Collectively, these data show that the effect described was not specific to IBD, but likely to be relevant across chr21q22-associated diseases.

Line 250ff: The sentence about the ENCODE project is more to be seen as an information for the reviewing process. It can be deleted. Not necessary for this project.

This sentence has now been removed.

Line 253: This reviewer thinks to remember that the referenced manuscript did not have 64 conditions, but less. Please check again and correct if necessary.

Thank you for highlighting this. To clarify, this seminal study (Xue *et al.* Immunity 2014) described activating monocytes with 28 different stimulation conditions, which is the number the reviewer is probably remembering. However, when we downloaded the microarray data, we realised that many of the 28 stimuli had been applied for different periods of time. For example, data were available for 8 different durations of exposure to IFN γ after initial treatment with GMCSF (ranging from 30 minutes to 72 hours). Since *ETS2* expression varied with duration of exposure as well as with stimulus – and because co-expression analysis works best when more conditions are available – we used all of the individual monocyte/macrophage conditions (i.e. all unique combinations of stimulus + duration of exposure) in our analysis. We initially reported this as 64 conditions, but on re-examining the data realised this was actually 67. We have now corrected this in our methods and revised the main text to better reflect how the authors of the referenced manuscript described their work:

“We therefore first used a “guilt-by-association” approach to identify genes that were co-expressed with *ETS2* across 67 human monocyte/macrophage activation conditions (comprising 28 different stimuli and several durations of exposure)¹⁷.

“Human monocyte-derived macrophage gene expression data files (n=314) relating to 28 different stimuli with multiple durations of exposure (collectively comprising 67 different activation conditions) were downloaded from Gene Expression Omnibus (GSE47189) and quantile normalised.”

Line 275: It must state: "To further elucidate the mechanisms", because as presented, it seems that the metabolic and transcriptional regulation by *ETS2* are independent and therefore, the term "further" would be correct.

We have added the word “further” as requested:

“To further elucidate the mechanisms responsible for such diverse inflammatory effects, we sought to directly identify *ETS2* target genes.”

Line 276ff and the whole last result section: The strong statement about no anti-ETS antibody working for ChIP-seq and then coming back that one worked for CUT&RUN is too dramatic. Make it more simple and more neutral scientific language. In fact, the discussion about no availability of any anti-ETS antibody for ChIP-seq could be avoided as follows: “As no anti-ETS antibody for ChIP-seq approaches could be identified, we applied two approaches to define *ETS*-driven transcriptional and functional programs on a global scale: 1. Co-expression, 2. CUT&RUN, not requiring fixation. “ This reviewer would think that this would have less "drama".

Thank you for this suggestion, which relates more to writing style than content. We did not intend for this section to be overly dramatic, but think that it is important that scientific writing has an engaging narrative as well as being accurate and factual. The narrative of how we initially investigated *ETS2*'s metabolic effects – based on co-expression analysis – before returning to re-evaluate methods for directly identifying binding sites (because metabolic effects did not account for the observed phenotype) accurately reflects the progression of this phase of the work. Indeed, had treatment with roxadustat corrected all of the transcriptional effects of deleting *ETS2*, we would not have persevered with genome-wide CUT&RUN. We have now made several edits to this section which we hope are

acceptable, including removing the sentences regarding ENCODE being unable to immunoprecipitate ETS2 and our own failed attempts to perform ChIP-seq, but have kept the structure of the section since this accurately reflects the sequence of experiments.

“We next sought to understand how *ETS2* controlled such diverse macrophage effector functions. Studying *ETS2* biology is challenging because no ChIP-seq-grade antibodies exist, precluding direct identification of its transcriptional targets. We therefore first used a “guilt-by-association” approach to identify genes that were co-expressed with *ETS2* across 67 human monocyte/macrophage activation conditions (comprising 28 different stimuli and several durations of exposure)¹⁷.”

“To further elucidate the mechanisms responsible for such diverse inflammatory effects, we sought to directly identify *ETS2* target genes. Since ChIP-seq involves steps that can alter protein epitopes and prevent antibody binding (e.g. fixation) we investigated whether any anti-*ETS2* antibodies might work for Cleavage-Under-Targets-and-Release-Using-Nuclease (CUT&RUN), which does not require these steps.”

Line 289ff: In the text, the authors state percentages, in the figure absolute genes counted. Link figure and text better. As provided, the reader needs to start to think, where the 48% and the 50% are derived from. Make it easier for the reader to link figure and text.

We have now revised the text to indicate the absolute gene count as well as the percentages, and have edited the figure to provide overlap percentages in the horizontal bar plot:

“Overall, 48.3% (754/1560) of genes dysregulated following *ETS2* disruption, and 50.3% (1078/2153) of genes dysregulated following *ETS2* overexpression, contained an *ETS2* binding peak within their core promoter or putative cis-regulatory elements (**Fig.4h**)”

Legend: **h**. UpSet plot of intersections between *ETS2* gene lists, including genes with *ETS2* peaks in their core promoters or cis-regulatory elements and significantly up- (Up) or down-regulated (Dn) genes following *ETS2* editing (KO) or overexpression (OE). Vertical bars denote shared genes between lists, indicated by connected dots in lower panel. Horizontal bars denote percentage of gene list within intersections.

Line 293ff: What exactly is meant with the statement: "... rather than being solely attributable to differences in inflammation". Not clear to this reviewer.

We agree that this was not very clear. We have now re-written this sentence:

“Notably, *ETS2* targets included *HIF1A*, *PFKFB3*, and other glycolytic genes (e.g. *GPI*, *HK2*, and *HK3*), consistent with the observed metabolic changes being directly induced as part of this complex pro-inflammatory programme.”

Line 296: Avoid words such as "Interestingly", "Intriguingly", "Clearly". All interpretations.

We have minimised use of such words in the manuscript at the reviewer’s request. However, on occasion it can be useful to highlight particular findings to the reader – both for emphasis and to improve readability. On this occasion, we think that *ETS2* binding to its own enhancer – which the

reviewer describes as “of importance and interest” in their next comment – is sufficiently noteworthy to warrant highlighting. We have therefore not changed the word “Intriguingly”, but are happy to defer to editor if they also feel this should be removed.

Lines 296: The findings concerning binding of ETS2 to its own enhancer is of importance and interest, but the suggested feed-forward loop is speculative. Either erase such statements, at least move them to the discussion, or show experimentally that this is the case. The authors do have all tools at hand, to provide experimental evidence for such a feed-forward loop.

Thank you for this suggestion. We agree that our wording may have been too speculative and have now performed two complementary experiments to test the positive feedback loop that we proposed. First, we performed H3K27ac ChIP-seq in ETS2-edited or unedited inflammatory (TPP) macrophages. After generating BAM files, we extracted the reads that mapped to the enhancer region (chr21:40,465,000-40,470,000, hg19) and used edgeR with TMM normalisation to assess the effect of *ETS2* disruption on the strength of the disease-associated enhancer (including donor as a covariate). This identified a small but significant reduction in enhancer activity following ETS2 disruption (log2FC -0.15, P = 0.014). Reciprocally, we performed H3K27ac ChIP-seq in resting (M0) macrophages transfected with either ETS2 mRNA or control mRNA (both 500ng). These data were analysed in the same way and showed a trend to increased enhancer strength following ETS2 overexpression (log2FC 0.78, P = 0.088). The results are therefore consistent with ETS2 contributing to the activity of its own enhancer, but we cannot exclude a contribution from the associated changes in inflammation. We have therefore softened the language used in the main text and have added the supporting data to Extended Data Fig.6g-i. Code to reproduce these analyses is also provided:

“...we also detected ETS2 binding at its own enhancer at chr21q22 (**Fig.4i**). This is in keeping with reports that PU.1 and ETS2 can interact synergistically⁴¹, and suggests that ETS2 might contribute to the activity of its own enhancer. Indeed, modulating ETS2 expression altered enhancer activity in a manner potentially consistent with positive autoregulation (**Extended Data Fig.6**).”

g. Schematic of experiment to assess how ETS2 disruption affects the activity of the chr21q22 ETS2 enhancer in inflammatory (TPP) macrophages. **h.** Schematic of experiment to assess how ETS2 overexpression affects the activity of the chr21q22 ETS2 enhancer in resting (M0) macrophages. **i.** Normalised H3K27ac ChIP-seq read counts (edgeR fitted values) from chr21:40,465,000-40,470,000 in experiments depicted in **g** (left) and **h** (right) (edgeR P values, n=3 for each).

Line 300: The authors link all their findings to chronic inflammation. Some of the experimental conditions are most likely acute inflammatory conditions (short term activation of monocyte-derived macrophages via signals related to chronic inflammation). Can the authors make an attempt to address the role of ETS2 in acute inflammation or alternatively, make a statement about limitations of the study, suggesting that the role of ETS2 throughout the process of induction and maintenance of inflammation (early phases, late phases, chronic phases) requires additional work. E.g. what are the triggers that are necessary for a patient to acquire IBD beyond having a genetically determined elevation of ETS2, which by itself probably does not yet trigger IBD.

Thank you for this comment. There are several complex points here, some of which we can address and some that are beyond the scope of the paper. In our revised manuscript we show that *ETS2* overexpression induces a transcriptional state that resembles monocytes/macrophages from multiple chronic inflammatory diseases (Fig.3f, reproduced above). Moreover, using scRNA-seq data, we show that diseased tissue (Crohn's disease intestine) contains expanded populations of inflammatory monocytes and macrophages that have high expression of *ETS2* and *ETS2*-regulated genes (shown below). Similar results were obtained in PSC, another chronic inflammatory disease linked to chr21q22, using spatial transcriptomics (data shown above). Together, this confirms that *ETS2* plays an important role in chronic inflammation:

Figure 5

Extended Data Figure 7

In addition, we have now shown that the genetic basis of inflammatory disease at chr21q22 is the same as the genetic basis for increased *ETS2* expression by macrophages during acute exposure to bacteria. Equivalent colocalisation was not observed after longer periods of bacterial exposure, or following acute exposure to other pathogens e.g. viruses. This is consistent with an additional role for *ETS2* in at least some forms of acute inflammation – the importance of which is underscored by the evolutionary conservation of the pleiotropic risk allele at chr21q22. Moreover, a potential role for

ETS2 in acute inflammatory responses was also suggested by *ETS2* overexpression recapitulating the phenotype of monocytes/macrophages from acute bacterial infections. We have now included discussion of these results and their potential implications (some of the relevant text excerpts are provided above in response to earlier comments):

“For example, the shared genetic basis of inflammatory disease risk and increased *ETS2* expression during early exposure to bacteria highlights a situation in which *ETS2*-driven macrophage responses may be beneficial. This also suggests that *ETS2* may function in different phases of inflammation, consistent with *ETS2*-regulated genes being induced during serious bacterial infections.”

Of note, the final sentence in this reviewer comment relates to the environmental factors that trigger IBD in a genetically susceptible individual. While our colocalisation analyses provide clues to the gene-environment interactions that are likely to be involved at this locus – which we have included in the Discussion (excerpts above) – a broader discussion regarding other possible environmental triggers of IBD, which may affect other genetic pathways, is less relevant for this paper.

Fig 5f: This seems almost too good to be true. A better way for the reader to see, whether there was an almost 100% overlay would be to show FC/FC or ratio/ratio plots comparing the DE genes after chr21q22 deletion with the MEK inhibition. Also here, one can provide percentages of similar regulation, opposite regulation and strength differences of regulation. Please provide such information.

Thank you for this suggestion. We have now replaced the heatmap with a FC/FC plot (Fig.5i) – reproduced below. This depicts all of the genes that were differentially expressed following deletion of the disease-associated enhancer on chr21q22, with their fold-change plotted against the equivalent fold-change following MEK inhibition. Those with FDR $P < 0.1$ in the MEK inhibition experiment are coloured: red for upregulated genes and blue for downregulated genes. The proportion of genes in each quadrant is shown on the plot. Overall, 96.6% of downregulated genes and 88.2% of upregulated genes showed similar regulation (Pearson $r = 0.81$). These experiments were not performed using cells from the same donors, which complicates comparisons of strength differences, but the effect sizes were generally larger following MEK inhibition, especially for downregulated genes: median downregulated \log_2FC : -2.09 (MEKi), -0.52 (chr21q22 deletion); median upregulated \log_2FC : 0.65 (MEKi), 0.54 (chr21q22 deletion). We have not added these numbers to the text as the \log_2FC differences are depicted in the plot. Of note, Extended Data Figure 8c (below) also shows the GSEA enrichment plots for this comparison:

Legend: i. FC/FC plot of differentially-expressed genes following chr21q22 enhancer deletion, plotted against their fold-change following MEK inhibition. Percentages indicate proportion of upregulated (red) and downregulated genes (blue). Coloured points (blue or red) were differentially expressed following MEK inhibition (FDR <0.1).

Extended Data Fig.8

a-c. Gene set enrichment analysis (fGSEA) in MEK1/2 inhibitor-treated TPP macrophages showing enrichment of gene sets upregulated (upper panel) or downregulated (lower panel) following *ETS2* or chr21q22 editing (MEK1/2 inhibited using PD-0325901, 0.5 μ M). Gene sets obtained from differential gene expression analysis (limma using voom transformation) following *ETS2* disruption with gRNA1 (**a**), gRNA2 (**b**), or following chr21q22 deletion (**c**).

Lines 343ff: This reviewer is not arguing that the MEK inhibition experiments are not intriguing and promising, but since upstream regulators will have additional gene programs that are also changed, it would be necessary to show the differences between upstream blockade (MEK inhibitor) and *ETS2* blockade (as presented here by KO approaches) to see, whether there might be already off targets to be recognized. For precision medicine approaches, this would be similarly important and needs to be addressed, at least by providing a computational analysis predicting for MEK-related, *ETS2* independent effects on macrophage function. The authors actually discuss the limitations of MEK inhibitors in the discussion, but it could be a very strong argument for the approach that even such potentially toxic effects are to be discovered by the chosen functional genomics approach.

We have now performed this analysis and provide the results in Extended Data Fig.8. To do this, we considered all of the genes that were differentially expressed following MEK inhibition (upregulated and downregulated) and identified those with no evidence of being *ETS2* targets in our data (incorporating differential expression from the knockout or overexpression experiments and promoter / regulatory element binding from the *ETS2* CUT&RUN analysis). Of 1366 downregulated genes and 916 upregulated genes following MEK inhibition, we identified 271 and 210 genes respectively that were neither bound by *ETS2*, nor showed matching differential expression following *ETS2* modulation (i.e. down with MEK inhibitor and *ETS2* knockout or down with MEK inhibitor and up with *ETS2* overexpression). Pathway analysis on these *ETS2* independent genes did not provide clear insights into the biological pathways involved:

e. Proportion and pathway analysis of MEK inhibitor-induced differentially expressed genes that have no evidence for being *ETS2* targets in macrophages (incorporating differential expression from knockout or overexpression experiments and promoter / regulatory element binding from *ETS2* CUT&RUN).

However, while this analysis does not highlight any specific toxicity concerns (particularly since unwanted metabolic effects are not a reported side-effect of systemic MEKi usage) it does not allow us to draw firm conclusions regarding the safety and/or side-effects of MEK inhibition. This is because the known toxic effects of MEK inhibitors do not occur in macrophages, but in unrelated tissues (such as retinal epithelium and cardiomyocytes where MEK has other physiological roles). To reliably predict unwanted effects of MEK inhibition – as may be required for precision medicine – a whole body tissue analysis would be required, which is outside the scope of this work.

Line 366f: This is another generalizing statement that is neither correct nor necessary. Even if this might be correct for the current SNP, it is not correct overall. For example, change in gene expression itself might be of great value for patient stratification irrespective of linking the patterns to disease mechanisms. Avoid such general opinion statements. The manuscript is better without.

This statement has now been removed as part of a more extensive re-write of the discussion.

Lines 375ff: The discussion about the allele frequency and its origins is interesting, but it does not help to understand, why autoimmune diseases show such a strong increase during the last decades. The authors should replace this nice intellectually stimulating discussion about origin with more solid discussion about the potential reasons that - in addition to the existing genetics - are most likely the major environmental drivers. This would be more important for developing tangible strategies for individuals at higher risk.

Thank you for this comment. The increase in autoimmune diseases during the last decades is clearly driven by environmental changes (e.g. diet, pollution, westernisation, infection, antibiotics etc) – particularly as genetics did not change within this timeframe. However, the exact triggers remain largely unknown and are likely to differ between diseases (e.g. EBV infection in multiple sclerosis, gut dysbiosis in IBD). We have now highlighted the importance of environmental triggers in our discussion, and have discussed the gene-environment interactions that we think are most likely to be relevant at the chr21q22 locus. These interactions directly relate to the origins of the risk allele and may explain why it has not been negatively selected during human evolution (hence we have shortened this section rather than remove it altogether). However, a broader discussion regarding the possible environmental drivers of autoimmune diseases, and the potential public health implications that could be considered to mitigate personal and population risk, is outside the scope of this manuscript but has been reviewed extensively elsewhere (e.g. PMIDs 24602341, 25196523, 25961452, 37283765).

“This is particularly important since rates of autoimmunity continue to rise, yet how environmental factors trigger disease in genetically-susceptible individuals remains largely unknown. Here, by investigating a pleiotropic disease locus, we uncover a central regulator of human macrophage effector functions and a key pathogenic pathway that is potentially druggable. These discoveries also provide clues to the gene-environment interactions at this locus. For example, the shared genetic basis of inflammatory disease risk and increased *ETS2* expression during early exposure to bacteria highlights a situation in which *ETS2*-driven macrophage responses may be beneficial. This also suggests that *ETS2* may function in different phases of inflammation, consistent with *ETS2*-regulated genes being induced during serious bacterial infections. Swift bacterial killing upon first encounter is critical to avoid disseminated infection, and could provide a balancing selective pressure to maintain the chr21q22 ancestral allele in human populations. This may explain why this disease risk allele remains so common (frequency ~75% in Europeans, >90% in Africans) despite being an exceptionally old variant (>500,000 years) that was even polymorphic in archaic humans (**Extended Data Fig.9**).”

Line 389f: This is a very strong statement. If the authors want to claim that this locus confers susceptibility to multiple inflammatory diseases, then more data and results (as already mentioned

above) about other autoimmune diseases throughout the manuscript is required. Otherwise focus on IBD here as well.

This is an important point, which we agree could have been better articulated in the manuscript. To clarify, there is already robust evidence that the chr21q22 locus confers susceptibility to multiple inflammatory diseases based on independent genome-wide significant associations being detected at the rs2836882-tagged haplotype. Our revisions to Fig.1 now enable us to show that the genetic basis for all of these distinct associations is identical. This is an important result since it directly implicates a single molecular effect – which we show relates to enhanced PU.1 binding, increased chromatin accessibility (a new result we added in response to another reviewer comment) and increased activity of a long-range ETS2 enhancer – as being responsible for susceptibility to all of the chr21q22-associated inflammatory diseases.

In addition to the new genetic colocalisation analyses, we have also added additional data throughout the manuscript that illustrates the relevance of an ETS2-regulated inflammatory pathway to other chr21q22-associated diseases. These data have been discussed in detail in response to earlier comments, in particular our “Line 223” response (and so are not reproduced here again) but include:

- New SNPsea analysis confirming enrichment of genetic variants from all of the other chr21q22-associated diseases in ETS2-regulated genes.
- New pathway analysis using macrophage gene signatures from 11 different diseases, confirming that *ETS2* overexpression induces genes that are characteristic of chr21q22-associated inflammatory diseases.
- New scRNA-seq and spatial transcriptomic analyses showing that there are expanded populations of inflammatory monocytes/macrophages within the affected tissues of chr21q22-associated diseases and that these cells express higher levels of ETS2 and ETS2-regulated genes.

Line 390ff: Yet another very strong statement, but the authors should recognize that attempts to explain SNPs by other means (e.g. generating animal models with the same genetic variant) have been successfully performed before. The news here is that it is now presented solely in human studies, but the authors should either here or at the beginning of the introduction give credit to those who have done similar work in animals. Actually, this would also be an option to reduce the subliminal criticism of GWAs at some places in the manuscript. While the reviewer agrees that GWAS information urgently needs to be linked to work as presented here, a scientific original paper does not have to criticize (even if only subliminal) the GWAS approach and field. Not necessary (albeit correct).

We did not intend to criticise the GWAS approach and field. Rather, we wanted to highlight that the success of GWAS – in discovering hundreds of risk loci – now provides a unique opportunity to better understand disease mechanisms. The point that the biological effects of most of disease-associated variants is unknown is widely accepted and not a criticism of GWAS *per se*. We agree that animal models have been successfully used to study a small number of conserved variants, and have now cited examples of such studies in the introduction:

“Nevertheless, to fully realise the potential of genetics, knowledge of where risk variants lie must first be translated into an understanding of how they contribute to disease¹⁰. Animal models can help facilitate this, especially for coding variants in conserved genes^{12,13}.”

Line 537: 100 Genomes phase 3 needs reference.

Line 541: eQTL Catalog: needs reference

Thank you for highlighting these omissions. We have now added both of these references:

“...with reference minor allele and LD information calculated from 503 European samples from 1000 Genomes phase 3 (ref 54)... SuSiE fine-mapping results were obtained for *ETS2* (identifier ENSG00000157557 or ILMN_1720158) in monocyte/macrophage datasets from the eQTL Catalogue⁵⁵.”

Line 546: "... previously". Code needs to be provided.

A link to the code used for this analysis was provided in the Code Availability section of the methods. We have now clarified this at the indicated place in the manuscript:

“Raw H3K27ac ChIP-seq data from primary human immune cells were downloaded from Gene Expression Omnibus (GEO series GSE18927 and GSE96014) and processed as described previously⁶² (code provided in Code Availability section).”

Line 547: Downloading data is nice, but describing the computational approaches performed with the data is better. Authors need to be more precise. Overall: Authors are requested to provide ALL CODE for ALL analyses in one repository space (authors can choose), so that FAIR principles are respected (particularly R = reproducibility!). The links provided do not provide this information yet and it is therefore not possible for the reviewer to judge, whether reproducibility is following FAIR principles.

We have double-checked the manuscript and added to the large amount of code that we had already deposited in Github. We have also added a comment to each code chunk to indicate which analyses were performed using that code. Of note, the data referred to in this comment (processed promoter-capture Hi-C data) did not require any code / computational analysis since the results are publicly available. We simply looked up the relevant region in the monocyte data results table. We have now clarified this in the text:

“Processed promoter-capture Hi-C data⁶³ from 17 primary immune cell-types were downloaded from OSF (<https://osf.io/u8t zp>) and cell-type CHiCAGO scores for chr21q22-interacting regions were extracted.”

Line 569: the word "genes" is missing.

Thanks for spotting this error. This has now been corrected:

“...most closely resembled CD14+ monocytes/macrophages from active IBD – using disease-associated lists of differentially-expressed genes⁵⁷.”

Lines 634: Accessibility path according to FAIR principles to these data is also not provided. This reviewer does not ask for open data, only for the accessibility path. (see also comment below).

Thank you for this comment – we are committed to supporting FAIR principles for data access, but the data referred to in line 634 are simple Taqman genotyping results, rather than the sort of sequencing data that should be deposited in an accessible repository. For transparency and accessibility, however, we have now provided source data in spreadsheet form for the data underlying every figure panel in the manuscript.

Line 870: This is not a title for a methods subheading

Apologies – we have now re-named all of the methods subheadings that were too short:

“Roxadustat in TPP macrophages”

“MEK inhibition in TPP macrophages”

“Colonic biopsy explant culture”

Line 928: IMPORTANT! According to European GDPR regulations, human data cannot be uploaded to GEO anymore without breaching European law. The authors should use EGA instead and provide a clear accessibility path to the data.

Thank you for highlighting this point, which we agree is important. Of note, the situation is slightly different in the UK since we now have different GDPR regulations to the rest of Europe (UK GDPR and Data Protection Act, 2018). Our institute’s lawyers have informed us that there is no consensus position in UK law on this point, but given the lack of clarity we have followed the reviewer’s suggestion and deposited our human genetic data in EGA (note the MPRA data will remain in GEO as this contains barcode sequencing only with no human component so is ineligible for EGA). Almost all of the other datasets were deposited in EGA in August/September (accession codes below) but they do not yet appear on the EGA website. We have followed this up with the EGA and have been informed that this is due to delays at their end, but is in progress. The additional sequencing data that we have generated as part of this revision (ATAC-seq and H3K27ac ChIP-seq in ETS2-edited and unedited cells, and H3K27ac ChIP-seq in ETS2-overexpressing cells) were uploaded recently and are yet to be available within our EGA account for compilation into a formal submission. This will be done as soon as it is available. Again, we have contacted EGA to try to expedite this. The accessibility paths are provided in the Data Accessibility section:

“The datasets produced in this study are accessible via the following repositories: MPRA (GEO: GSE229472), RNA-seq of *ETS2* or chr21q22-edited TPP macrophages (EGA: EGAD00001011338), RNA-seq of *ETS2* overexpression (EGA: EGAD00001011341), RNA-seq of MEK inhibitor-treated TPP macrophages (EGA: EGAD00001011337), H3K27ac ChIP-seq in TPP macrophages (EGA: EGAD0000101351), ETS2 CUT&RUN (EGA: EGAD0000101349), biopsy RNA-seq data (EGA: EGAD00001011333). MetaboLights: Metabolomics (MTBLS7665).”

Referee #2 (Remarks to the Author):

There is a lack of effective treatments for autoimmune/inflammatory diseases. Genetics has been shown to drastically improve the successful identification of novel drugs. However, the use of the vast amount of genetic data generated for the past 20 years in drug discovery programmes has been limited, due to a lack of understanding on the mechanisms by which genetic variants predispose to disease and the genes they affect. This currently remains a major challenge. Stankey et al address this critical challenge with an array of functional genomics, molecular biology, and immunology assays to study a risk locus on 21q22 that has been shown to be associated with several autoimmune diseases including IBD and AS. The authors robustly linked the risk locus to a causal gene, ETS2 (although this had already been suggested by previous CHi-C and eQTL studies), and a single causal variant, describe the mechanism by which the causal variant affect expression of the causal gene, describe the role that this gene has in macrophage inflammatory responses, and propose drugs that could target these mechanisms. The data is convincing, and the study elegantly takes a GWAS locus from just genetic association to potential translation into patient benefit.

Thank you for these positive comments and for the helpful suggestions that you make in this review. We have provided point-by-point responses below with relevant text excerpts for ease of reviewing.

One criticism may be that the role of the ETS2 gene, and the pathways it regulates, could have been explored further in disease. All experiments were performed in monocytes isolated from blood from healthy volunteers; could the authors explain if they would expect to see the same results in monocytes isolated from the affected tissues in IBD (intestine), AS (joints) and PSC (liver) patients? The only attempt to look at these mechanisms directly in disease is an interrogation of differential expression of genes in the ETS2 pathway from public datasets, but more details are needed to evaluate how robust this data is; how many patients were included in the transcriptomic analyses used?

Thank you for this suggestion. We have now revised our manuscript to include a broader exploration of the role of ETS2 in disease. Of note, it is not possible to obtain fresh biopsies from the affected tissues of most chr21q22-associated diseases – either because these are not biopsied in clinical practice (e.g. spinal joints and ligaments in ankylosing spondylitis or major arteries in Takayasu's arteritis) or because taking multiple biopsies would carry a significant risk to the patient and so specific ethical approval would be needed (and difficult to obtain given the risks involved – e.g. liver biopsies from PSC). For this reason, we have used other complementary approaches to address this important question:

First, we identified publicly-reported macrophage signatures from a range of different diseases, including multiple chr21q22-associated diseases (Crohn's disease, ulcerative colitis, Takayasu's arteritis, ankylosing spondylitis) as well as other diseases in which macrophages become activated via different stimuli e.g. viral infections, bacterial infections and cancer. Importantly, these represented true disease signatures derived from human macrophages from patients with the respective conditions (rather than *in vitro* stimulations). The signatures included:

Disease	Tissue source	PMID	Source of signature	Method	Category
Crohn's disease	Intestine	30670762	Cluster E and F markers	scRNAseq	Inflammatory disease (chr21q22-linked)
Takayasu arteritis	Blood	34671607	DEGs (up) in CD14+ monocytes (vs controls)	scRNAseq	Inflammatory disease (chr21q22-linked)
Ankylosing spondylitis	Blood	37388915	DEGs (up) in CD14+ monocytes (Table S3)	RNAseq (bulk)	Inflammatory disease (chr21q22-linked)
Ulcerative colitis	Intestine	31348891	DEGs (up) in m.monocyte subgroups (vs controls; Table S4)	scRNAseq	Inflammatory disease (chr21q22-linked)

Influenza A	Upper airway	34618691	DEGs (up) in macrophages (vs controls, Supp. File 4)	scRNAseq	Viral infection
COVID-19	Upper airway	34618691	DEGs (up) in macrophages (vs controls, Supp. File 4)	scRNAseq	Viral infection
Tuberculosis	BAL	37470432	DEGs (up) in macrophages (active TB vs controls, Table S2)	scRNAseq	Bacterial infection
Bacterial sepsis	Blood	32066974	DEGs (up) in MS1 vs all monocytes (Table S2)	scRNAseq	Bacterial infection
Tumour-associated (multiple cancers)	Tumour (lung, colon, liver, breast, stomach, and pancreas)	34331874	DEGs (up) in IL411+ TAMs (cluster #6, Table S3B)	scRNAseq	Cancer
Tumour-associated (endometrial)	Tumour	30930117	DEGs (up) in endometrial TAMs vs endometrial tissue resident macs (Table S2)	scRNAseq	Cancer
Atherosclerosis	Atherosclerotic plaques	36190844	Foamy macrophage marker genes (Table S3)	scRNAseq	Atherosclerosis

DEGs (up), differentially-expressed genes (upregulated); TAMs, tumour-associated macrophages; BAL, bronchoalveolar lavage

These signatures were then used as gene sets for fGSEA to assess whether *ETS2* overexpression induced similar enrichment across all disease states. Notably, the Crohn's signature was from a different study to the one used for Fig.3e, and thus provides validation for the enrichment observed. This analysis demonstrated that *ETS2* overexpression does not simply cause a generalised activation state, but rather induces genes that are characteristic of chronic inflammatory disease and, to a lesser extent, responses to bacterial infection. In contrast, no significant enrichment was observed for tumour-associated macrophage signatures or macrophages during certain viral infections (e.g. influenza). The results are shown in a new panel in Fig.3:

Legend: **f**. Enrichment of macrophage signatures, derived from patients with the indicated diseases (colour coded by category), in *ETS2*-overexpressing macrophages (relative to control). Numbers represent NES, dashed line denotes FDR 0.05. Crohn's disease signature is from a different study to that shown in **e**.

Second, to more broadly assess the effects of *ETS2* on macrophage heterogeneity in disease, we obtained intestinal immune cell scRNA-seq data from Crohn's disease and healthy controls and extracted the myeloid cells for analysis. After pre-processing (filtering, batch correction, and normalisation) we performed clustering using Seurat4 and detected 7 clusters, many of which were readily identifiable as specific myeloid subsets using established markers and/or prior literature. The expression of *ETS2* and *ETS2*-regulated pathways were strongly differentially expressed between these cell clusters. For example, higher expression was noted in inflammatory macrophages (cluster 1, denoted by expression of *CD209*, *CD163L1*, *CCL4*, *CD68*, *IL1B*, and *FCGR3A*) and inflammatory monocytes (cluster 2, denoted by expression of *S100A8/A9*, *TREM1*, *CD14*, *IL1RN*, and *MMP9*) while expression in tissue resident macrophages (cluster 0, denoted by expression of *C1QA*, *C1QB*, *FTL* and *CD63*) and conventional dendritic cells (cluster 5, denoted by expression of *CLEC9A*, *CADM1*, and *XCR1*) was substantially lower.

Notably, most of the top 50 marker genes in the cluster with the highest *ETS2* expression (cluster 2, *ETS2* fold-change vs other cells = 1.8, FDR $P = 2.42e-173$) were experimentally-confirmed *ETS2* targets (i.e. differentially expressed following *ETS2* knockout or overexpression). These included many inflammatory genes linked to disease pathogenesis, including *S100A8*, *S100A9*, *MMP9*, *TREM1*, *TLR2* and *ETS2* itself. Moreover, these inflammatory monocytes (as well as the *ETS2*-expressing inflammatory macrophages, cluster 1) were markedly increased in disease, consistent with previous reports. These data are now included in Figure 5a,b and Extended Data Fig.7:

Figure 5:

Extended Data Figure 7

Legend: **a.** ETS2 expression in scRNA-seq clusters of myeloid cells from Crohn's disease and healthy controls (upper panel). Relative contributions of single cells from Crohn's disease or healthy controls to individual clusters (same UMAP dimensions as for combined analysis).

Third, because equivalent scRNA-seq datasets were not available for other chr21q22-associated diseases, we set up a new collaboration to obtain fixed liver tissue from PSC and healthy controls. We then performed a spatial transcriptomics experiment (CosMx, Nanostring) to assess the effects of ETS2 on macrophage heterogeneity in a different chr21q22-associated disease. The results are shown in new data panels in Fig.5 (below) and mirror the results from the scRNA-seq analysis in Crohn's disease, with increased numbers of inflammatory macrophages being found in PSC. Moreover, this analysis showed that these cells were closely related to the bile duct cholangiocytes (the target of pathology in PSC) and that expression of ETS2-regulated genes was higher the closer the macrophages were to the cholangiocytes – consistent with a role in disease pathogenesis:

c. Spatial transcriptomics of PSC and healthy liver (n=4). Images show representative fields of view with cell segmentation and semi-supervised clustering results (InsituType). Legend indicates InsituType cell-types. Hep., hepatocyte; LSECs, liver sinusoidal endothelial cells. d. Average number of macrophages within a defined radius of a cholangiocyte. e. Distance from cholangiocytes to nearest macrophage. Data shown as Tukey box-and-whisker plot. Mann-Whitney test, two-tailed. Data in d and e represent 10,532 PSC and 13,322 control cholangiocytes. f. Scaled expression of *ETS2*-regulated genes in 21,067 PSC macrophages (excluding genes used to defined macrophage subsets) at defined distances from cholangiocytes. Data represent mean and 95%CI.

Fourth, we have now conducted a recall-by-genotype study in IBD patients (approved and facilitated by the NIHR IBD BioResource). Blood was taken from 22 IBD patients who were homozygous for either the risk allele at rs2836882 or the non-risk allele. Patients in each group were matched for age, sex, medication and disease activity. Due to the limited volume of blood that we were allowed to collect, we could not perform a detailed study of cellular phenotypes, but were able to measure a panel of *ETS2*-regulated genes in resting (M0) macrophages. The results are shown in Fig.2h and demonstrate that several *ETS2*-regulated genes are more highly expressed in risk allele carriers – consistent with the predicted effects of altered *ETS2* expression:

Legend: “h. Quantitative PCR of selected *ETS2*-target genes in resting (M0) macrophages from minor and major allele homozygote IBD patients (n=22, expression normalised to *PPIA* and scaled to minimum 0, maximum 1). Box represents median (IQR), whiskers represent minima and maxima.”

“Indeed, genotype-specific differences in the expression of multiple *ETS2*-target genes were noted in resting (M0) macrophages from IBD patients who were homozygous for either the rs2836882 risk or non-risk allele (matched for age, sex, medication, and disease activity) (Fig.2h).”

Finally, in response to the reviewer’s last question in this comment, we have added the number of patients used to generate each disease-specific ranked gene list in the figure legend. Of note, this figure panel has now been moved to Extended Data Fig.7 due to space constraints in our revised figure.

“b. Gene set enrichment analysis (fGSEA) of genes downregulated following chr21q22 enhancer deletion or *ETS2* disruption (gRNA1 or gRNA2) within intestinal macrophages from patients with active IBD (compared to control intestinal macrophages, n=20; left), AS synovium (compared to control synovium, n=15; centre), and PSC liver biopsies (compared to control liver biopsies, n=17; right).”

The authors found that *ETS2* genes were enriched for IBD risk loci. This is very interesting, and this reviewer wonders if this data could have been taken further, by performing a “*ETS2* genetic risk score” or looking back at patients carrying genetic variants in this pathway to identify clinically meaningful monocyte-driven disease subgroups (would drugs targeting this pathway be more efficient in these patients?).

Thank you for this suggestion. We have now performed this analysis using data from the UK IBD BioResource, which comprises nearly 20,000 IBD patients. To do this, we calculated a weighted Polygenic Risk Score (PRS) for 22 IBD-associated SNPs that have been reported to directly implicate ETS2-regulated genes. Considering every patient for whom genotype and phenotype data were available, we analysed 8,898 UC patients and 9,351 Crohn's disease patients - making this one of the largest IBD PRS analyses ever performed. We investigated for associations of the PRS with a series of disease phenotypes to determine whether increased genetic burden in ETS2-regulated IBD-associated genes might have any clinical consequence. The results are presented in a new Extended Data Figure (5). In summary, we observed significant associations with several clinical features that have been linked to a more severe disease phenotype, including earlier age at diagnosis, increased need for surgery, and stricturing or fistulating complications in Crohn's disease. No associations were found for response to anti-TNF α , perianal involvement in Crohn's disease and disease extent in UC.

It was not possible to assess whether drugs targeting this pathway would be more effective in patients selected by genotype (since drugs targeting this pathway have not yet been used for IBD). However, our finding that the transcriptional footprint of ETS2 is consistently enriched in monocytes/macrophages from unselected IBD patients (**Fig.3e-f, Fig.5a-b** – also reproduced below) implies that genetic stratification may not be required. Indeed, ~95% of IBD patients will carry one or two risk alleles at the chr21q22 locus alone (based on risk allele frequency ~75%) – aside from any alleles they may also carry in other genes in this pathway. Methods describing the PRS analysis have been added to the manuscript and the data and code availability sections have been updated. A sentence has been added to the main manuscript describing the results:

“A polygenic risk score comprising these variants was found to associate with several features of more severe IBD across ~18,000 patients, including earlier age at diagnosis, increased need for surgery, and stricturing or fistulating complications in Crohn's disease (**Extended Data Fig.5**).”

Extended Data Figure 5. Polygenic Risk Score of 22 ETS2-regulated IBD-associated genes

a. Summary of IBD BioResource cohorts used for PRS analysis. **b.** Association between PRS and age at diagnosis. **c.** Association between PRS and extent of ulcerative colitis (E1, proctitis; E2, left-sided; E3, extensive colitis). **d.** Association between PRS and Crohn's disease location (L1, ileal; L2, colonic; L3, ileocolonic). L2 is associated with a milder disease phenotype. **e.** Association between PRS and perianal involvement in Crohn's disease. **f.** Association between PRS and Crohn's disease behaviour (B1, inflammatory; B2, stricturing; B3, fistulating). B2 and B3 represent more aggressive, complicated forms of Crohn's disease. **g.** Association between PRS and response to anti-TNF α in Crohn's disease and ulcerative colitis (PR, primary responder; PNR, primary non-responder). **h.** Association between PRS and need for surgery in Crohn's disease and ulcerative colitis. ... Analysis in **b** performed using linear regression. Analyses in **c-h** performed using logistic regression (with diagnosis as covariate in **g** and **h**). Higher PRS was associated with: earlier age at diagnosis, ileal or ileocolonic forms of Crohn's disease, B2/B3 Crohn's disease behaviour, and increased need for surgery in IBD. SNPs included in PRS are listed in Extended Data Table 1.

Methods:

"Polygenic Risk Score.

Plink1.9 (<https://www.cog-genomics.org/plink/1.9/>) was used to calculate a polygenic risk score (PRS) for patients in the IBD BioResource using 22 ETS2-regulated IBD-associated SNPs (beta coefficients from ref.3). Linear regression was used to compare PRSs with age at diagnosis, and logistic regression to estimate the effect of PRSs on IBD sub-phenotypes including anti-TNF primary non-response (PNR), CD behaviour (B1 vs B2/B3), perianal disease and surgery. For variables with more than 2 levels (e.g. CD location or UC location), ANOVA was used to investigate the relationship with PRS. For analyses of age at diagnosis, anti-TNF α response and surgery, IBD diagnosis was included as a covariate."

Referee #3 (Remarks to the Author):

This paper establishes a mechanism by which a common haplotype linked with a spectrum of autoimmune diseases can lead to functional disease phenotypes. The authors identified a SNP located in a novel enhancer of *ETS2* that increases both PU.1 binding and downstream *ETS2* transcription. They show that *ETS2* is both necessary and sufficient for inflammatory responses in human macrophages and that *ETS2* signaling is central to inflammatory bowel disease (IBD) pathogenesis. They identified a drug that targets upstream regulators of *ETS2* and can abrogate *ETS2* driven inflammation phenotypes in primary patient samples.

Thank you for your careful review of our manuscript, and your helpful suggestions. We have provided point-by-point responses below with relevant text excerpts for ease of reviewing.

The findings are potentially important but also raise several questions. Specifically, the authors show there is a correlation between the risk haplotype and higher levels of *ETS2* expression. However, the authors also note that the risk haplotype being studied is incredibly common (in ~75% of Europeans and ~90% of Africans). It remains unclear how the identified SNP leads to disease outcomes in some but not others.

This is an important question, which we are grateful for the opportunity to address.

To clarify, the contribution of GWAS variants to complex diseases is polygenic, such that none are either necessary or sufficient to cause disease. Common risk variants are not necessary because other genetic variants (or environmental factors) can have the same functional effect on a key pathway (as illustrated by the multiple Crohn's disease variants that impair autophagy). Similarly, they are not sufficient because complex disease development, by definition, requires hits in multiple pathways – together with an environmental trigger – for disease to occur. This means that if an individual does not encounter a relevant environmental trigger, nor carries genetic hits in other pathways, they may not develop disease despite carrying risk alleles at rs2836882 (and vice versa). Crucially, this does not mean that pathways identified via GWAS are unimportant; there are several examples of GWAS hits in genes that regulate fundamental disease processes and are key therapeutic targets e.g. *HMGCR*, which is associated with cardiovascular disease and encodes HMG Co-A reductase – the target of statins. Moreover, when monogenic forms of disease do occur – for example due to missense variants in key amino acids – the genes affected are often also GWAS hits (e.g. *IL10*, *TNFAIP3*, *LACC1* in IBD, although not *ETS2* since loss of this gene causes embryonic lethality). The main reason why common non-coding variants are usually insufficient to cause disease is that the degree of pathway dysregulation they confer is modest. For example, each copy of the risk allele at rs2836882 increases *ETS2* expression by between 20-40% depending on the cellular activation state (example from resting monocytes now shown in Extended Data Fig.1, and reproduced below). The utility of these variants is therefore not as diagnostic markers but as a starting point to discover the key pathways they affect, which in turn directly contribute to disease pathogenesis.

We have now re-written our Discussion to emphasise that complex disease development requires both genetic susceptibility and an environmental trigger, and to discuss the environmental triggers that are likely to be involved at this locus – based on genetic colocalisation of the inflammatory disease signal with increased *ETS2* expression by macrophages following acute exposure to bacteria (data added in response to a question from another reviewer, also reproduced in the same figure below).

“Arguably the greatest challenge in modern genetics is to translate the success of GWAS into a better understanding of human disease. This is particularly important since rates of autoimmunity continue to rise, yet how environmental factors trigger disease in genetically-susceptible individuals remains largely unknown.”

“These discoveries also provide clues to the gene-environment interactions at this locus. For example, the shared genetic basis of inflammatory disease risk and increased *ETS2*

expression during early exposure to bacteria highlights a situation in which ETS2-driven macrophage responses may be beneficial. This also suggests that ETS2 may function in different phases of inflammation, consistent with ETS2-regulated genes being induced during serious bacterial infections. Swift bacterial killing upon first encounter is critical to avoid disseminated infection, and could provide a balancing selective pressure to maintain the chr21q22 ancestral allele in human populations. This may explain why this disease risk allele remains so common (frequency ~75% in Europeans, >90% in Africans) despite being an exceptionally old variant (>500,000 years) that was even polymorphic in archaic humans (Extended Data Fig.9).”

Extended Data Figure 1. Colocalisation between genetic associations at chr21q22.

a. Example comparison of genetic associations at chr21q22: IBD and *ETS2* eQTL in unstimulated monocytes. Plot adapted from locuscomparer. **b.** Boxplot depicting *ETS2* expression stratified by rs2836882 genotype in unstimulated monocytes⁵⁶. **c.** Radar plot of representative colocalization results for the indicated genetic associations compared to IBD. Posterior probability of independent causal variants, PP.H3, dark blue; posterior probability of shared causal variant, PP.H4, light blue. PP.H4 > 0.5 was used to call colocalisation (denoted by dashed line). Labels are coloured according to class of data (indicated in the key). Asterisks denote colocalisation. Data sources are: IBD³, PSC⁵, AS⁴, Takayasu Arteritis⁶, BLUEPRINT⁵⁸, Fairfax⁵⁶, Quach⁵⁷, Nedelec⁵⁹, Alasoo⁶⁰.

Comments:

The authors should examine whether *ETS2* expression is higher in individuals carrying the risk allele or is elevated specifically in patients with IBD symptoms. A critical question is whether *ETS2* expression be used as indicator of whether or not someone with a risk haplotype exhibits IBD symptoms?

Thank you for this question. In our revised manuscript, we show that *ETS2* expression is higher in individuals carrying the risk allele (see Extended Data Fig.1b above) and is also increased in IBD. For example, using scRNA-seq data of intestinal myeloid cells, we identify a large expansion of inflammatory monocytes and macrophages in IBD, which strongly express *ETS2* and *ETS2*-regulated genes. These cells make up ~1% of myeloid cells in health but over 50% in Crohn's disease (the data and related text are shown after your scRNA-seq question below).

Moreover, we now show that resting (M0) macrophages from IBD patients who are homozygous for the rs2836882 risk allele have higher baseline expression of multiple *ETS2* target genes than macrophages from non-risk allele carrier patients:

Legend: “h. Quantitative PCR of selected *ETS2*-target genes in resting (M0) macrophages from minor and major allele homozygote IBD patients (n=22, expression normalised to *PPIA* and scaled to minimum 0, maximum 1). Box represents median (IQR), whiskers represent minima and maxima.”

“Indeed, genotype-specific differences in the expression of multiple *ETS2*-target genes were noted in resting (M0) macrophages from IBD patients who were homozygous for either the rs2836882 risk or non-risk allele (matched for age, sex, medication, and disease activity) (Fig.2h).”

However, in response to questions from other reviewers, we also now show – using publicly-reported macrophage signatures from different diseases – that *ETS2* overexpression not only induces genes characteristic of chr21q22-associated inflammatory diseases, but also genes upregulated during bacterial infection (Fig.3f – shown below). This mirrors the genetic colocalisation results (described above) and may provide clues to the evolutionary selective pressure that has driven persistence of the risk allele. From a diagnostic perspective, however, this means that *ETS2* expression would not be a useful diagnostic marker – especially since a common differential diagnosis for IBD is infection.

Legend: **f**. Enrichment of macrophage signatures, derived from patients with the indicated diseases (colour coded by category), in ETS2-overexpressing macrophages (relative to control). Numbers represent NES, dashed line denotes FDR 0.05. Crohn's disease signature is from a different study to that shown in **e**.

Of note, while the molecular effects of individual non-coding genetic variants are not considered useful as diagnostic tools, there is increasing interest in combining common risk variants into Polygenic Risk Scores (PRS) to assess whether the overall burden of genetic risk has any clinical consequence. In response to a question from another reviewer, we calculated a weighted PRS for the 22 IBD-associated SNPs that directly implicate ETS2-regulated genes in 8,898 UC patients and 9,351 Crohn's disease patients. We then investigated for associations of this PRS with a series of disease phenotypes. The results are presented in a new Extended Data Figure (5) and demonstrate significant associations between the genetic burden in ETS2 pathway genes and several clinical features that are linked to a more severe disease phenotype. These include earlier age at diagnosis, increased need for surgery, and stricturing or fistulating complications in Crohn's disease. No associations were found for response to anti-TNF α , perianal involvement in Crohn's disease and disease extent in UC.

Extended Data Figure 5. Polygenic Risk Score of 22 ETS2-regulated IBD-associated genes

a. Summary of IBD BioResource cohorts used for PRS analysis. **b.** Association between PRS and age at diagnosis. **c.** Association between PRS and extent of ulcerative colitis (E1, proctitis; E2, left-sided; E3, extensive colitis). **d.** Association between PRS and Crohn's disease location (L1, ileal; L2, colonic; L3, ileocolonic). L2 is associated with a milder disease phenotype. **e.** Association between PRS and perianal involvement in Crohn's disease. **f.** Association between PRS and Crohn's disease behaviour (B1, inflammatory; B2, stricturing; B3, fistulating). B2 and B3 represent more aggressive, complicated forms of Crohn's disease. **g.** Association between PRS and response to anti-TNF α in Crohn's disease and ulcerative colitis. **h.** Association between PRS and need for surgery in Crohn's disease and ulcerative colitis. ... Analysis in **b** performed using linear regression. Analyses in **c-h** performed using logistic regression (with diagnosis as covariate in **g** and **h**). Higher PRS was associated with: earlier age at diagnosis, ileal or ileocolonic forms of Crohn's disease, B2/B3 Crohn's disease behaviour, and increased need for surgery in IBD. SNPs included in PRS are listed in Extended Data Table 1.

Line 105 PU.1 can bind to heterochromatin... Not sure that this indeed has been proven.

Thank you for this insightful comment, and apologies for the error. The reviewer is correct that PU.1 is best considered a non-classical pioneer factor (per Minderjahn *et al.* Nat Comms 2020) because it does not bind heterochromatin *in vitro* despite being shown to have pioneering qualities *in vivo*. For example, PU.1 expression can extensively remodel chromatin (Heinz *et al.* Mol Cell 2010, Minderjahn *et al.* Nat Comms 2020), maintain nucleosome depletion at macrophage-specific enhancers (Barozzi *et al.* Mol Cell 2014) and re-program non-myeloid cells to express a myeloid gene expression program (Nerlov and Graf Genes Dev 1998; Feng *et al.* PNAS 2008). We have therefore corrected this sentence to better reflect the current state of knowledge regarding PU.1.

“PU.1 is an important non-classical pioneer factor in myeloid cells²⁰ that can bind DNA, initiate chromatin remodelling – thus enabling other transcription factors to bind – and activate transcription²¹.”

Figure 1g. Could the authors please show cis elements associated with genome-wide PU.1 occupancy? This is to validate the quality of the ChIP-seq data.

Thank you for this suggestion. Of note, the PU.1 ChIP-seq data we used were from a published, peer-reviewed manuscript and so we expect that these may have already been assessed for quality (Schmidt *et al.* 2016 “The transcriptional regulator network of human inflammatory macrophages is defined by open chromatin” Cell Res). We have now provided a reviewer figure (below) that shows the cis-elements that are associated with genome-wide PU.1 occupancy. A legend is provided below the figure to explain what each panel shows:

Reviewer figure: Cis-elements associated with genome-wide PU.1 occupancy

a. Enrichment heatmaps of PU.1 ChIP-seq peaks ($q < 0.01$) in regions identified by ATAC-seq (accessible chromatin), H3K4me3 ChIP-seq (active promoters), and H3K27ac ChIP-seq (active regulatory elements) in TPP macrophages (4-kb peak-centred regions). **b.** Enrichment of an PU.1 binding motif in PU.1 ChIP-seq peaks (TFmotifView; hypergeometric P value). **c.** Genome-wide distribution of PU.1 ChIP peaks within gene-based genomic features. **d.** Summarised distribution of PU.1 ChIP-seq peaks around gene bodies (genome-wide). Shaded region represents 95% confidence interval. **e.** Genome-wide distribution of PU.1 ChIP-seq peaks relative to transcription start site of genes. Data in **c-e** generated using *ChIPseeker* package in R.

Could the authors please include ATAC-Seq data to confirm allelic differences in chromatin accessibility?

Thank you for this suggestion. We have addressed this in two complementary ways. First, we have obtained access to monocyte ATAC-seq data from 30 healthy volunteers and 16 patients with ankylosing spondylitis from a recent study (Brown *et al.* Cell Genomics 2023, PMID: 37388915). Within this cohort, we identified 16 heterozygous individuals and extracted allelic ATAC-seq reads at rs2836882. The results are shown in Fig.1k and demonstrate a consistent allele-specific difference in chromatin accessibility. In addition, we performed our own ATAC-seq experiment (in response to this reviewer's later request for an ATAC-seq study in *ETS2*-edited and unedited TPP macrophages). Two of the three donors in that experiment were heterozygous at rs2836882, and the pile-ups of the allelic reads from these deeply sequenced samples demonstrate the same allele-specific difference in chromatin accessibility. These results are shown in Extended Data Fig.4g. Description of these data have been added to the manuscript.

Extended Data Fig.4

“Moreover, ATAC-seq in monocytes and macrophages from rs2836882 heterozygotes revealed allelic differences in chromatin accessibility that would be consistent with differential binding of a pioneer factor (**Fig.1k, Extended Data Fig.4**).”

“Collectively, these data reveal a genetic mechanism whereby the putative causal variant at chr21q22 – identified via its functional consequences in primary macrophages – promotes binding of a pioneer transcription factor, enhances chromatin accessibility, and increases the activity of a long-range *ETS2* enhancer.”

Figure 1i. Could the authors please show the patterns for all four donors?

We have now provided the H3K27ac pile-ups for all four donors in Extended Data Fig.4h (reproduced below), and have performed a differential ChIP-seq analysis to formally assess the statistical significance of altered enhancer activity in risk allele homozygotes. We did this using MEDIPS, the software tool that performed best for differential analysis of sharp peak data (such as H3K27ac ChIP-seq) according to a recent benchmarking study (Eder and Grebien. *Genome Biology* 2022). For this analysis, we compared the quantile-normalised H3K27ac signals in 5-kb bins across the extended chr21q22 region (chr21:40,150,000-40,750,000) between risk and non-risk allele homozygotes. The results have been added to Fig.1k and show that the 5-kb region containing rs2836882 is the only bin within the extended locus that is significantly different between the donors (2.14-fold increase in risk allele homozygotes, $P_{adj} = 0.005$). We have added description of this analysis and the results to the manuscript, and have deposited code to reproduce the analysis in the “ChIP-seq” folder in our Github repository.

Legend: **I.** H3K27ac ChIP-seq data from risk (top) or non-risk (bottom) allele homozygotes at rs2836882. Data shown from two of four donors. Differential ChIP-seq analysis performed using MEDIPs.

Extended Data Fig.4

Legend: **h.** H3K27ac ChIP-seq data from risk (red) or non-risk (blue) allele homozygotes at rs2836882 (n=4).

“While several nearby enhancer peaks were similar between these donors, the enhancer activity overlying rs2836882 was significantly stronger in major (risk) allele homozygotes (**Fig.11, Extended Data Fig.4**), contributing to a ~2.5-fold increase in enhancer activity across the extended chr21q22 locus (**Extended Data Fig.4**).”

Figure 2. It would be nice to perform a scRNA-seq analysis to examine whether subsets of populations were more affected upon removal of ETS2 expression. Skewing of the populations remains a bit of a worry in bulk RNA-Seq analysis. A more detailed analysis seems necessary. Same for the overexpression data.

We agree that the manuscript would be strengthened by a deeper understanding of the specific myeloid cell subsets in which ETS2 is principally active (and which would be most affected by its deletion / overexpression). However, we were not sure that the proposed experiment would best address this question. The reasons for this are as follows:

- (1) Activating blood-derived monocytes with a cocktail of cytokines for 6 days is likely to homogenise any physiological heterogeneity that may have existed prior to stimulation. We have previously shown that this certainly occurs with CD4 T cells (Bourges *et al.* EMBO Mol Med 2020).
- (2) This experiment would only be able to assess blood-derived cells and so effects in other myeloid subsets, which are also present in diseased tissue (e.g. tissue resident macrophages, dendritic cells etc) would be missed.
- (3) The degree of *ETS2* overexpression we induce (per Fig.3b) would saturate the small fraction of scRNAseq reads from transfected cells and make any physiological heterogeneity very difficult to detect in overexpression experiments.

For these reasons, we sought to address this important question using a complementary approach that would facilitate a broader assessment of the effects of *ETS2* on macrophage heterogeneity in diseased tissue.

To do this, we first obtained intestinal immune cell scRNAseq data from IBD patients and healthy controls and extracted the myeloid cells for analysis. After pre-processing (filtering, batch correction, and normalisation) we performed clustering using Seurat4 and detected 7 clusters, many of which were readily identifiable as specific myeloid subsets using established markers and/or prior literature. The expression of *ETS2* and *ETS2*-regulated pathways differed significantly between these cell clusters. For example, higher expression was noted in inflammatory macrophages (cluster 1, denoted by expression of CD209, CD163L1, CCL4, CD68, IL1B, and FCGR3A) and inflammatory monocytes (cluster 2, denoted by expression of S100A8/A9, TREM1, CD14, IL1RN, and MMP9) while expression in tissue resident macrophages (cluster 0, denoted by expression of C1QA, C1QB, FTL and CD63) and conventional dendritic cells (cluster 5, denoted by expression of CLEC9A, CADM1, and XCR1) was substantially lower (Figure 5a,b and Extended Data Fig.7 – reproduced below).

Notably, most of the top 50 marker genes in the cluster with the highest *ETS2* expression (cluster 2, *ETS2* fold-change vs other cells = 1.8, $P_{adj} = 2.42e-173$) were experimentally-confirmed *ETS2* targets – as determined by differential expression following *ETS2* knockout or overexpression. These included many inflammatory genes linked to disease pathogenesis, including S100A8, S100A9, MMP19, TREM1, TLR2 and *ETS2* itself. Moreover, these inflammatory monocytes (as well as the *ETS2*-expressing inflammatory macrophages, cluster 1) were markedly increased in disease, consistent with previous reports:

Figure 5

Legend: **a.** Myeloid cell clusters in scRNA-seq from Crohn's disease and healthy controls (upper). Scaled expression of *ETS2*-regulated genes (genes downregulated in *ETS2*-edited cells; middle). Relative contributions of Crohn's disease and control cells to each cluster (lower). **b.** Scaled expression of selected genes from known myeloid cell subsets and *ETS2*-regulated genes.

Extended Data Figure 7

Legend: **a**. *ETS2* expression in scRNA-seq clusters of myeloid cells from Crohn's disease and healthy controls (upper panel). Relative contributions of single cells from Crohn's disease or healthy controls to individual clusters (same UMAP dimensions as for combined analysis).

“To better understand how *ETS2* affects macrophage heterogeneity in diseased tissue, and determine whether this might present a therapeutic target, we first examined intestinal scRNA-seq data from Crohn's disease and healthy controls⁴². After extracting myeloid cell data for analysis, we detected 7 cell clusters (**Fig.5a**) – many of which were identifiable as specific myeloid subsets using established markers and/or prior literature (**Fig.5b**). Several clusters were expanded in disease, including inflammatory macrophages (cluster 1, expressing *CD209*, *CCL4*, *IL1B*, and *FCGR3A*) and inflammatory monocytes (cluster 2, expressing *S100A8/A9*, *TREM1*, *CD14*, and *MMP9*). These cells – which are known to increase in Crohn's disease⁴³ – showed higher expression of *ETS2* and *ETS2*-regulated genes compared to other clusters, including tissue-resident macrophages (cluster 0, expressing *C1QA*, *C1QB*, *FTL*, and *CD63*) and conventional dendritic cells (cluster 5, expressing *CLEC9A*, *CADM1*, and *XCR1*) (**Fig.5a, Extended Data Fig.7**).”

Since equivalent scRNA-seq datasets were not available for other chr21q22-associated diseases, we set up a new collaboration to obtain fixed liver tissue from PSC and healthy controls. We then performed a spatial transcriptomics experiment (CosMx, Nanostring) to assess the effects of *ETS2* on macrophage heterogeneity in a different chr21q22-associated disease. The results are shown in new data panels in Fig.5 (below) and mirror the results from the scRNA-seq analysis in Crohn's disease, with increased numbers of inflammatory macrophages being found in PSC. Moreover, this analysis showed that these cells were closely related to the bile duct cholangiocytes (the target of pathology in PSC) and that expression of *ETS2*-regulated genes was higher the closer the macrophages were to the cholangiocytes – consistent with a role in disease pathogenesis:

“Similarly, spatial transcriptomics on fixed liver tissue from PSC and unaffected controls revealed increased numbers of inflammatory macrophages in PSC, which were found in close proximity to cholangiocytes – the principal target of pathology (**Fig.5c-e**). Moreover, *ETS2*-regulated genes were most highly expressed in the macrophages nearest to PSC cholangiocytes – consistent with a role in disease pathogenesis (**Fig.5f**). Indeed, using bulk RNA-seq data we found that the transcriptional footprint of *ETS2* was detectable in the affected tissues of multiple chr21q22-associated diseases (**Extended Data Fig.7**).”

c. Spatial transcriptomics of PSC and healthy liver ($n=4$). Images show representative fields of view with cell segmentation and semi-supervised clustering results (InsituType). Legend indicates InsituType cell-types. Hep., hepatocyte; LSECs, liver sinusoidal endothelial cells. **d.** Average number of macrophages within a defined radius of a cholangiocyte. **e.** Distance from cholangiocytes to nearest macrophage. Data shown as Tukey box-and-whisker plot. Mann-Whitney test, two-tailed. Data in **d** and **e** represent 10,532 PSC and 13,322 control cholangiocytes. **f.** Scaled expression of ETS2-regulated genes in 21,067 PSC macrophages (excluding genes used to defined macrophage subsets) at defined distances from cholangiocytes. Data represent mean and 95%CI.

Collectively, these data provide detailed insights into the specific myeloid cell subsets that are most affected by ETS2 and demonstrate how these populations change in the context of disease tissue.

Could the authors please perform ATAC-Seq analysis on wild-type cells and cells depleted for ETS expression?

Thank you for this suggestion. We have now performed this experiment and also performed H3K27ac ChIP-seq in ETS2-edited and unedited cells. The results are shown in Extended Data Fig.2. They reveal that the transcriptional changes induced by ETS2 disruption arise without major changes to genome-wide chromatin accessibility. This appears to be consistent with a recent report that transcriptional changes induced by single-factor perturbations often occur without major changes in chromatin accessibility (Kiani et al. Mol Syst Biol. 2022 PMID 36069349). In contrast, we observe a general reduction in enhancer activity following ETS2 disruption, consistent with the observed transcriptional changes. We have added a sentence to describe these results in the manuscript:

n. Chromatin accessibility in ETS2-edited versus unedited inflammatory macrophages ($n=3$). **o.** Enhancer activity (H3K27ac) in ETS2-edited versus unedited inflammatory macrophages ($n=3$). Red points denote $P_{\text{adj}} < 0.1$, grey points NS.

“Notably, however, the transcriptional changes that occurred following *ETS2* disruption were not accompanied by significant changes in genome-wide chromatin accessibility, although enhancer activity was generally reduced (**Extended Data Fig.2**).”

The motif analysis associated with the CUT&RUN experiments should reveal p-values and other elements associated with ETS occupancy.

Thank you for this suggestion. In our original analysis, we had only investigated whether an ETS2 motif was enriched within the ETS2 CUT&RUN peaks. We have now re-performed this analysis and have included additional motifs for all of the JASPAR 2020 database transcription factors that are expressed in TPP macrophages (CPM > 0.5). We present the results in Extended Data Fig.6 (reproduced below). Unsurprisingly, we detect enrichment for several ETS family members (given the motif similarities across the family) but we also find enrichment for many transcription factors that are known to interact with ETS2, including FOS, JUN, NFKB and PU.1. The results therefore provide additional support for being sites of ETS2 binding and reveal other elements that are associated with ETS2 occupancy:

“These peaks were mostly located in active regulatory regions (90% in promoters or active enhancers, **Fig.4d,e**) and were highly enriched for both a canonical ETS2 motif (4.02-fold vs global controls, **Fig.4f**) and for motifs of transcription factors known to interact with ETS2, including FOS, JUN, and NF- κ B⁴⁰ (**Extended Data Fig.6**).”

f. TFmotifView enrichment results for motifs of transcription factors expressed in TPP macrophages (CPM > 0.5) within ETS2 CUT&RUN peaks. Results shown for all significantly enriched transcription factors (Bonferroni P value < 0.05) with motifs in more than 10% peaks.

Tornado plots should be presented for CUT&RUN data in parallel with enhancer and promoter marks.

Thank you for this suggestion, which we have now added to Fig.4d. The results demonstrate enrichment of ETS2 binding in: accessible chromatin (based on ATAC-seq-defined regions), promoters (based on H3K4me3 marks) and active regulatory elements (based on H3K27ac marks). All of these datasets are from inflammatory TPP macrophages, and the results corroborate those shown in Fig.4e (which divided the ETS2-bound CUT&RUN regions according to genomic features that delineate enhancers, promoters, and non-regulatory regions).

Of note, the “peak-valley-peak” appearance of the enrichment in H3K4me3 and H3K27ac marks is consistent with ETS2 binding to accessible nucleosome-free regions that are flanked by modified histones on either side (described in “Peak-valley-peak pattern of histone modifications delineates active regulatory elements and their directionality”. Pundhir et al. *Nucleic Acids Research* 2016. doi: 10.1093/nar/gkw250):

“One antibody identified multiple, significantly-enriched genomic regions (peaks) of which 6,560 were reproducibly detected across two biological replicates (Irreproducible Discovery Rate cut-off 0.01) with acceptable quality metrics³⁹ (**Fig.4d**). These peaks were mostly located in active regulatory regions (90% in promoters or active enhancers, **Fig.4d,e**)”

Legend: **d**. Enrichment heatmaps of *ETS2* CUT&RUN peaks (IDR cut-off 0.01, n=2) in regions identified by ATAC-seq (accessible chromatin), H3K4me3 ChIP-seq (active promoters), and H3K27ac ChIP-seq (active regulatory elements) in TPP macrophages (4-kb peak-centred regions). **e**. Features of *ETS2* binding sites (based on gene coordinates and H3K27ac ChIP-seq in TPP macrophages).

Referee #4 (Remarks to the Author):

A disease associated gene desert orchestrates macrophage inflammatory responses via ETS2

Stankey et al.

Stankey et al highlight an example of how a common genetic association can improve the understanding and reveal new therapeutic treatments for inflammatory / autoimmune disease. The paper focuses on the chr21q22 locus and the major (risk) allele haplotype which is associated with 5 different inflammatory diseases. They identify a monocyte/macrophage specific enhancer which contains the SNP rs2836882. This enhancer elevates ETS2 expression and the SNP colocalizes with a PU.1 binding site.

Loss of function for ETS2 results in a decrease in phagocytosis and a decrease in oxidative bursts. RNA-seq on ETS2 edited vs unedited inflammatory macrophages from multiple donors revealed a reduction in cytokines, chemokines, secreted effector molecules, cell surface receptors, pattern recognition receptors and signaling molecules with ETS2 loss of function. Gene-set enrichment analysis showed a reduction in macrophage activation, pro-inflammatory cytokine production, phagocytosis, ROS production following ETS2 disruption.

ETS2 is necessary and sufficient for inflammatory responses in human macrophages and acts as a master regulator. This is highlighted through the ETS2 over-expression experiments where it drives the pathways identified in the ETS2 loss of function RNA-seq experiment in a dose-dependent manner. Furthermore, they showed that over-expressing ETS2 in resting macrophages induced a transcriptional state that closely resembled disease macrophages in Crohn's disease. They also showed that ETS2 target genes were more strongly enriched for IBD associated loci than almost all previously implicated pathways, suggesting that ETS2 signaling in macrophages plays a fundamental role in IBD pathogenesis.

The authors performed CUT&RUN to identify ETS2 binding sites across the genome and they found that ~50% of genes dysregulated by ETS2 over expression or disruption contained an ETS2 binding site, suggesting ETS2 directs macrophage responses at a transcriptional level.

The authors next used the NIH LINCS database to identify drugs that might modulate ETS2 activity (i.e. mimic the transcriptional effect of disrupting ETS2 in inflammatory macrophages). For example, MEK inhibitors are the most common class and an upstream regulator of ETS2. The authors treated differentiated monocytes with MEK inhibitor and saw an anti-inflammatory activity that phenocopied the effect of disrupting ETS2 or deleting the chr21q22 enhancer. Treating IBD mucosal biopsies with MEK inhibitor reduced the secreted cytokine levels to those which were similar to Infliximab.

Overall, this study presents a very thorough GWAS locus dissection / variant-to-function story at the ETS2 locus. The authors have linked a GWAS risk variant associated with multiple autoimmune and inflammatory diseases to effects on TF binding, enhancer chromatin state, and ETS2 expression. Then, they link ETS2 to changes in downstream gene expression, metabolic state, and macrophage phenotypes.

The overall experimental rigor is high, including challenging genetic experiments in primary human cell and explant models (e.g., MPRA transfected into inflammatory macrophages from multiple donors, PU.1 ChIP-seq from multiple heterozygous donors, ETS2 mRNA over-expression in primary human macrophages, MEK inhibition in human gut explants).

The paper is also very clear and well written — an enjoyable read!

Thank you for your constructive assessment of our manuscript and positive comments – particularly regarding the clarity of our manuscript and our experimental rigor. We have provided point-by-point responses below with relevant text excerpts for ease of reviewing:

One consideration around the novelty of this study is that the role of ETS2 in macrophages has been extensively studied previously. Several other individual findings reported in this paper are similar to previous observations, such as the importance of ETS2 in cytokine production in macrophages, that IBD GWAS genes are enriched for ETS2 motifs, that treatment of an experimental colitis mouse model with MEK inhibitors reduces severity, etc. The authors here have contributed many new excellent experiments to put together these pieces into a complete story about how the human genetic variant affects ETS2, including new evidence in primary human cell models. The authors could perhaps cite some of this prior work more thoroughly where applicable, and provide some additional explanations where relevant to highlight the new findings about ETS2 in the Discussion.

Thank you for this suggestion. We agree that we could have better articulated what was previously understood regarding the role of ETS2 in macrophages, and the novel insights that our work provides. Crucially, while some studies have investigated ETS2 in macrophages, these have almost all focused on single target genes (e.g. mir-155, eotaxin, IL-6) and have produced contradictory results – with ETS2 being described as both necessary and redundant for macrophage development, and both pro- and anti-inflammatory. As a result, the exact role of ETS2 in human macrophages had not been definitively elucidated. Two key knowledge gaps further complicated our understanding of ETS2's role in macrophages. First, the transcriptional targets of ETS2 had never been directly confirmed – largely due to the fact that no ChIP-seq grade antibodies exist. Accordingly, ETS2 targets could only be predicted computationally based on motif enrichment. Second, none of the previous studies on ETS2 used primary human macrophages. Consequently, it was difficult to know whether the conflicting results were due to the use of cell-lines or complex mouse models. We address both of these knowledge gaps in our manuscript – delineating the genome-wide sites of ETS2 binding using CUT&RUN and performing all of our perturbation experiments in primary human macrophages. These points, combined with our systematic characterisation of ETS2 functions, provide multiple new insights into the role of ETS2 in macrophages. For example, effects on ROS production, phagocytosis and macrophage glycometabolism had never previously been described, and nor had it been realised that ETS2 would be both necessary and sufficient for driving macrophage effector responses. We believe that these findings, together with the mechanistic insights into how the disease-associated haplotype dysregulates this pathway and can be targeted pharmacologically, provide an important conceptual advance.

We have now re-written the discussion to provide additional context and highlight the new findings that we have made:

“Although ETS2 had previously been reported to have pro-inflammatory effects on individual genes^{26,27}, including cytokines²⁴, the full extent of its multi-faceted inflammatory programme – with additional effects on ROS production, phagocytosis, glycometabolism, and macrophage activation – was unknown. Moreover, without direct evidence of ETS2 targets, or knowledge of its function in primary human cells, it was difficult to reconcile reports of anti-inflammatory effects at other genes^{25,28} – especially since most previous studies were conducted in cell lines or mouse models²⁴⁻²⁸. By systematically characterising the effects of *ETS2* disruption and overexpression in primary human macrophages, we uncover an essential role for ETS2 in inflammatory effector functions, and delineate the transcriptional and metabolic mechanisms involved. This led to the realisation that ETS2 is sufficient to induce pathogenic phenotypes in disease macrophages, and that this process can be targeted pharmacologically.”

We also reference the conflicting evidence regarding the possible role(s) for ETS2 earlier in the main text:

“In contrast, the role of *ETS2* in primary human macrophages has been less clearly defined, with previous studies using either cell-lines or complex mouse models and often focusing on a limited number of downstream molecules²⁴⁻²⁸. This has led to contradictory reports, with *ETS2* being described as both necessary and redundant for macrophage development^{29,30}, and both pro- and anti-inflammatory²⁴⁻²⁸.”

In addition, we have now added citations to reference the other previous work that the reviewer mentioned:

“To better characterise the genetic risk attributable to the macrophage ETS2 pathway, we initially focused on IBD since this has far more genetic associations than any other chr21q22-associated disease. Encouragingly, a network of 33 IBD-associated genes from intestinal mucosa was previously found to be enriched for predicted ETS2 motifs³³. Examining the list of commonly downregulated genes following *ETS2* editing in macrophages ($P_{\text{adj}} < 0.05$ for both gRNAs), we identified over 20 IBD risk genes – including many that are thought to be causal at their respective loci^{3,34} (**Extended Data Table 1**).”

“Indeed, some of these drug classes, including MEK1/2 and HSP90 inhibitors, are reportedly beneficial in animal colitis models⁴⁵, although this is often a poor indicator of clinical efficacy – with several approved IBD treatments being ineffective in mice and many drugs that improve mouse models being ineffective in human IBD⁴⁶.”

I do not have any other major comments about the paper. Some minor comments + corrections below:

Corrections:

1. Page 3, line 36 “which may lie up to one million bases away”. Technically but rarely they can be located even further. I would suggest “which may lie up to millions of bases away”

Thank you for this suggestion. The need to optimise this sentence was also highlighted by one of the other reviewers. We incorporated both suggestions to re-phrase the sentence:

“Most non-coding variants are thought to affect gene regulation¹⁴, but the need to identify the causal gene(s) – which may lie millions of bases away – and the causal cell-type(s), which may only express the causal gene under specific conditions, have hindered attempts to discover disease mechanisms.”

2. Page 3, line 39 “none have led to new therapies”. This is not correct, I think. What about TYK2, IL12/IL23R? RORC and JAK2 have also motivated drug development programs, although I am not up to date on the status of these programs.

We apologise for the clunky wording in this sentence. The reviewer is correct that there are several drug development programs that target molecules whose genes lie close to GWAS hits. These genetic associations have accordingly been cited as support for the pharmacological approaches. However, the point we were trying to make was that few of these programs were initiated specifically because of disease mechanisms having been elucidated from genetics – most were based on animal studies or prior knowledge of the underlying biology. We recognise, however, that this is a nuanced point and have re-written this sentence to clarify the point we were trying to make:

“For example, although genome-wide association studies (GWAS) have identified over 240 IBD risk loci³, including several containing potential drug targets, fewer than 10 have been mechanistically resolved.”

3. For language reasons, suggest using “regulator” instead of “master regulator”

We hadn’t considered the cultural connotations of the term “master regulator” since this is widely used in biology, but – on reflection – agree with the reviewer that better terminology is needed. Since *ETS2* is both necessary and sufficient for inflammatory responses, we wanted to highlight that this gene appears to be a key regulatory factor, capable of orchestrating a complex and multi-faceted pro-inflammatory programme. The term “regulator” did not, in our view, convey this scope and

importance, so we have amended the text to use the term “central regulator” instead. We hope this is acceptable.

Abstract: “By investigating an intergenic haplotype on chr21q22, independently linked to inflammatory bowel disease (IBD), ankylosing spondylitis (AS), primary sclerosing cholangitis and Takayasu’s arteritis³⁻⁶, we discover that the causal gene, *ETS2*, is a central regulator of inflammatory responses in human macrophages and delineate the shared disease mechanism by which the risk haplotype amplifies *ETS2* expression.”

“Having found that *ETS2* was essential for monocyte-derived macrophage effector functions, we next investigated whether it might also be sufficient to drive them – as would be expected of a central regulator of inflammatory responses.”

“This shows that *ETS2* is both necessary and sufficient for inflammatory responses in human macrophages, consistent with being a central regulator of effector functions, whose dysregulation is directly linked to human disease.”

4. Extended data Figure 5. Labels for b,c and d have been swapped around (please fix)

Thank you for spotting this error. This has now been fixed. Of note, due to the number of additional analyses required – and the limited number of extended data figures – Extended Data Figure 5 has now been amalgamated into Extended Data Figure 2:

Legend: **j**. Cytokine secretion from inflammatory macrophages following deletion of the chr21q22 enhancer. Heatmap shows relative cytokine concentrations in the supernatants of chr21q22-edited TPP macrophages versus unedited (NTC) cells (Wilcoxon signed rank test, one-sided). **k**. Extracellular ROS production by unedited (NTC), chr21q22-edited, and *ETS2* g1-edited TPP macrophages, quantified using a chemiluminescence assay. Points represent relative area under curve for edited versus unedited cells (Wilcoxon signed-rank test). **l**. Representative flow cytometry histograms demonstrating phagocytosis of fluorescently-labelled zymosan particles by chr21q22-edited and unedited (NTC) TPP macrophages. **m**. Phagocytosis index for unedited and chr21q22-edited TPP macrophages, calculated as proportion of positive cells multiplied by mean fluorescence intensity of positive cells. Plot shows relative phagocytosis index for chr21q22-edited cells versus unedited cells (Wilcoxon signed-rank test). Error bars represent mean±SEM. * P < 0.05. NTC: non-targeting control.

Minor comments:

1. Fig 2: Could the authors please show some evidence for the ‘loss of function’ for ETS2 at a RNA or protein level (Sequencing data, expression data or Western blot?)

We apologise for not describing the significant CRISPR-mediated downregulation of *ETS2* expression in the main text. The effect was shown in the volcano plot in Fig.2e (below), but we agree that this could have been made clearer.

e. Differentially-expressed genes in *ETS2*-edited versus unedited inflammatory macrophages (limma with voom transformation, n=8).

Since we had not introduced the RNA-seq experiments at the point in the manuscript where we first describe the CRISPR/Cas9 editing approach, we have added a panel to Extended Data Fig.2 showing the reduction in *ETS2* mRNA expression as measured by qPCR. Additional text has been added to indicate that the CRISPR/Cas9 editing led to reduced expression:

“These gRNAs resulted in on-target editing in ~90% and ~79% of total cells respectively and effectively reduced *ETS2* expression (**Extended Data Fig.2**).”

Legend: f. *ETS2* expression (relative to NTC) following CRISPR/Cas9 editing, measured by qPCR (housekeeping gene *PPIA*; equivalent results with other housekeeping genes).

2. Fig 3f. Legend says “enrichments”, but figure shows p-values, not degree of enrichment. P-values depend on gene set size. Would be helpful for figure to show both enrichment and p-value

We apologise for the inaccurate wording. We had used the term “enrichment” in line with the description of the SNPsea algorithm ([doi.org/10/1093/bioinformatics/btu326](https://doi.org/10.1093/bioinformatics/btu326)) but completely agree that the figure shows statistical significance, not enrichment. Unfortunately, the outputs from SNPsea (a permutation p-value for each pathway, the number of null SNP sets with a greater pathway score than the provided SNP list, and the number of null SNP sets tested) do not permit calculation of enrichment – especially when none of the 5,000,000 null SNP sets have a greater pathway score

than the list of SNPs provided. This applies to both of the ETS2-regulated gene lists, as well as 2 of the previously associated pathways. We have therefore corrected the legend to accurately reflect what is shown in the figure, but not changed the figure itself.

Of note, in this analysis the gene set size effect, which the reviewer correctly highlights, is principally related to the size of the gene set tagged by the disease-associated SNPs – and thus the number of SNPs – rather than the number of genes in each of the queried pathways. This is controlled within the algorithm by using null SNP sets that are matched on the number of linked genes. The amended figure legend reads:

“f. SNPsea analysis of genes tagged by 241 IBD-associated SNPs within ETS2-regulated genes (red) and pathways previously linked to IBD pathogenesis (black). Significant pathways (Bonferroni-corrected permutation $P < 0.05$) indicated by §.”

3. Could you clarify — Do the human gut explant models include inflammatory macrophages?

Yes, the mucosal biopsies that are obtained endoscopically are known to contain inflammatory monocytes and macrophages, which increase in number during IBD (per Bernardo *et al.* Mucosal Immunology 2018: “Human intestinal pro-inflammatory CD11c^{high}CCR2⁺CX3CR1⁺ macrophages, but not their tolerogenic CD11c⁻CCR2⁻CX3CR1⁻ counterparts, are expanded in inflammatory bowel disease”). In response to a question from another reviewer – regarding macrophage subset heterogeneity – we have now included scRNAseq data that confirms this point, and additionally demonstrates that, unlike tissue resident macrophages, the inflammatory myeloid cells in IBD highly express *ETS2* and *ETS2*-regulated genes:

Figure 5

4. Regarding Figure 4i, is it possible that the rs2836882 variant falls within the ETS2 motif? The authors have said in Fig. 1 that the SNP lies outside the PU.1 motif but it would be good to have a figure showing where the SNP lies in relation to the PU.1 and ETS2 motif.

Thank you for this insightful suggestion, which we had not previously considered. We have now performed this analysis and have added a panel to Fig.4i that shows the location of the predicted

PU.1 and ETS2 binding sites relative to their ChIP-seq peaks and rs2836882 (reproduced below). This demonstrates that rs2836882 lies close to predicted binding sites for both ETS2 and PU.1 but does not fall within either motif. Methods for this analysis have also been added to the manuscript:

Legend: “i. ETS2 binding, PU.1 binding, chromatin accessibility (ATAC-seq), and enhancer activity (H3K27ac ChIP-seq) at the disease-associated chr21q22 locus. Predicted ETS2 and PU.1 binding sites shown below. Dashed line indicates rs2836882 position.”

Methods: “Predicted ETS2 and PU.1 binding sites were identified at the rs2836882 locus (chr21:40,466,150-40,467,450) using CisBP⁹⁶ (database 2.0, PWMs log odds motif model, default settings).”

5. The link between the GWAS variant and cellular phenotypes could be made stronger by for example CRISPR editing the variant or by studying some of the cellular phenotypes in macrophages with different genotypes.

Thank you for this suggestion. We have tried multiple ways to edit the risk variant in primary human macrophages, but unfortunately none have provided a sufficiently high editing efficiency to study the downstream consequences. The methods we have tried include CRISPR HDR, HITI (Waldron et al. Nature 2017), and adenine and cytidine base editing. A major issue we have encountered is that because monocytes do not divide in culture (and are mostly in G0/G1 phase) HDR is actively suppressed and therefore HDR-based editing methods do not work. This has been described for non-proliferating cells previously (Hustedt and Durocher Nature Cell Biology 2016). Moreover, in our hands, HITI is very inefficient, and while base editing can be performed in other primary immune cells (e.g. T cells) we have found that the same methods (transfection of base editor mRNA and gRNAs) do not work in primary monocytes. We suspect this is because monocytes, unlike T cells, have high RNAase expression that will particularly affect the uncapped gRNAs. Given the time constraints of the review process we have not been able to pursue this further.

In parallel, we sought and obtained permission for a recall-by-genotype study in IBD patients (approved by the NIHR IBD BioResource). Due to the limited volume of blood that we were allowed to collect, we could not perform a detailed study of cellular phenotypes, but were able to measure a panel of ETS2-regulated genes in resting (M0) macrophages from major allele homozygote and minor allele homozygote IBD patients (n=22). Patients were matched for age, sex, medication and disease activity. The results are shown in Fig.2h and demonstrate that several ETS2-regulated genes are more highly expressed at baseline in risk allele carriers – consistent with the predicted effects of altered *ETS2* expression:

Legend: **h.** Quantitative PCR of selected ETS2-target genes in resting (M0) macrophages from minor and major allele homozygote IBD patients (n=22, expression normalised to *PPIA* and scaled to minimum 0, maximum 1). Box represents median (IQR), whiskers represent minima and maxima.

“Indeed, genotype-specific differences in the expression of multiple ETS2-target genes were noted in resting macrophages from IBD patients who were homozygous for either the rs2836882 risk or non-risk allele (matched for age, sex, medication and disease activity) (**Fig.2h**).”

Of note, in our revised manuscript we also have strengthened the link between the causal variant and the observed cellular phenotypes by other means. For example, in response to a question from another reviewer, we have now shown that there is allele-specific chromatin accessibility at the site of the causal variant – both in publicly available monocyte ATAC-seq data from 16 heterozygous individuals (healthy controls and ankylosing spondylitis patients) and in deeply sequenced samples we obtained from two heterozygous individuals. Both datasets showed a consistent allele-specific difference in chromatin accessibility that would be in keeping with differential binding of a pioneer transcription factor. These results are shown in Fig.1k and Extended Data Fig.4g, and descriptions of the results have been added to the manuscript:

Legend: “**k.** Allele-specific ATAC-seq reads at rs2836882 in monocytes from 16 heterozygous individuals (including healthy controls and ankylosing spondylitis patients).”

Extended Data Fig.4

Legend: “**g**. Allele-specific ATAC-seq reads at rs2836882 in two deeply sequenced heterozygous TPP macrophage datasets (left: 154.7 million non-duplicate paired-end reads, right: 165.4 million non-duplicate paired-end reads).”

“Moreover, ATAC-seq in monocytes and macrophages from rs2836882 heterozygotes revealed allelic differences in chromatin accessibility that would be consistent with differential binding of a pioneer factor (**Fig.1k, Extended Data Fig.4**).”

“Collectively, these data reveal a genetic mechanism whereby the putative causal variant at chr21q22 – identified via its functional consequences in primary macrophages – promotes binding of a pioneer transcription factor, enhances chromatin accessibility, and increases the activity of a long-range *ETS2* enhancer.”

Reviewer Reports on the First Revision:

Referees' comments:

Referee #1 (Remarks to the Author):

The authors presented a very thoroughly performed and written revision of their original manuscript entitled "A disease-associated gene desert orchestrates macrophage inflammatory responses via ETS2"

On 55 pages they present answers to all questions by all reviewers, including new experiments, new analyses and where necessary changes in figures and text. The chain of arguments for their provided solutions to the requests by the reviewers have been very thoughtful and easy to understand. In rare occasions the authors suggested alternative solutions to the requests, which were all meaningful and justified.

This reviewer congratulates the authors to this important work and has no further requests, suggestions or comments.

Referee #2 (Remarks to the Author):

The authors have satisfactorily addressed my comments and I have no further concerns.

Referee #3 (Remarks to the Author):

The authors have addressed my comments. The manuscript has improved and will be of interest to a general audience.

Referee #4 (Remarks to the Author):

The authors have addressed my comments, thanks. I remain enthusiastic about the paper, and the revised manuscript has been improved significantly from the first submission. The addition of the recall-by-genotype study showing that homozygous risk individuals have higher expression of ETS2 targets in macrophages, and the allele-specific ATAC and ChIP analyses, are good additions that strengthen the link between this haplotype and downstream molecular and cellular phenotypes.

I have a few other small suggestions that would be helpful to address:

- In the Abstract, "directly in humans" should be "directly in primary human cells". The current wording sounds like functional genomics (e.g. CRISPR) was done directly in vivo in humans

- re: colocalization analysis on page 5: “the same causal variant(s)” would be more accurate than “a single causal variant” is responsible. Technically the colocalization analysis cannot distinguish if the association results from a single variant or two functional closely linked variants that both affect both traits
- Page 6: “dynamics of RNA transcription” would be better as “dynamics of mRNA expression”, since the PrimeFlow probes are presumably measuring mRNAs and not nascent transcripts
- Showing the sequence (e.g. actual nucleotides, with position of PU.1 motif marked, and location of the variant) around rs2836882 would be very helpful in Fig 1, for understanding how the variant might be affecting PU.1 binding. Are there any other motifs predicted by HOMER to be disrupted?
- Overall Fig 1 might benefit from aggregation of all of the genomic tracks into a single figure (e.g. showing fine-mapping posterior probabilities, MPRA data, ChIP-seq, data, allele-specific data. It’s a little disorienting to have to examine slightly different boundaries and regions in each of the subpanels
- Fig 1e legend: Currently written as “Data representative of one of four donors.”, which sounds like the data is only representative of one of the four. Reword, if it is consistent between the four donors. E.g. “Data corresponds to one representative donor, out of four.”
- Fig 2g — This would be better as a scatterplot of changes in expression in condition 1 vs changes in expression in condition 2. The slope of the relationship would be interesting to show, with regards to whether the enhancer KO leading to a weaker effect than the gene KO
- The Extended Data Figures appear to be often cited as entire figures (e.g. Page 10: “Extended Data Fig.2”) without specifying which panel out of many. It would be helpful to cite specific panels throughout.
- To interpret Fig. 3g, the numbers of genes in each set should be reported to interpret this figure, because it is easier to get high p-values for enrichment with larger gene sets. If enrichment cannot be directly reported from SNPsea, as the authors state in the rebuttal, then using another method such as MAGMA, which should report both an effect size beta and p-value, would be a good alternative

Author Rebuttals to First Revision:

Referees' comments:

Referee #1 (Remarks to the Author):

The authors presented a very thoroughly performed and written revision of their original manuscript entitled "A disease-associated gene desert orchestrates macrophage inflammatory responses via ETS2"

On 55 pages they present answers to all questions by all reviewers, including new experiments, new analyses and where necessary changes in figures and text. The chain of arguments for their provided solutions to the requests by the reviewers have been very thoughtful and easy to understand. In rare occasions the authors suggested alternative solutions to the requests, which were all meaningful and justified.

This reviewer congratulates the authors to this important work and has no further requests, suggestions or comments.

We are very grateful to this referee for their detailed review of our manuscript, and for these very positive comments.

Referee #2 (Remarks to the Author):

The authors have satisfactorily addressed my comments and I have no further concerns.

We are grateful to this referee for their positive assessment of our revised manuscript.

Referee #3 (Remarks to the Author):

The authors have addressed my comments. The manuscript has improved and will be of interest to a general audience.

We are grateful to this referee for their positive assessment of our revised manuscript.

Referee #4 (Remarks to the Author):

The authors have addressed my comments, thanks. I remain enthusiastic about the paper, and the revised manuscript has been improved significantly from the first submission. The addition of the recall-by-genotype study showing that homozygous risk individuals have higher expression of ETS2 targets in macrophages, and the allele-specific ATAC and ChIP analyses, are good additions that strengthen the link between this haplotype and downstream molecular and cellular phenotypes.

I have a few other small suggestions that would be helpful to address:

- In the Abstract, “directly in humans” should be “directly in primary human cells”. The current wording sounds like functional genomics (e.g. CRISPR) was done directly in vivo in humans

Thank you for this suggestion. This change has now been made:

“Together, this illustrates the power of functional genomics, applied directly in primary human cells, to discover immune-mediated disease mechanisms and potential therapeutic opportunities.”

- re: colocalization analysis on page 5: “the same causal variant(s)” would be more accurate than “a single causal variant” is responsible. Technically the colocalization analysis cannot distinguish if the association results from a single variant or two functional closely linked variants that both affect both traits

Thank you for highlighting this point. This change has now been made:

“To test for a shared disease mechanism, we performed colocalisation analyses and confirmed that the genetic basis for every disease was the same, meaning that a common causal variant(s) – and a shared molecular effect – was responsible.”

- Page 6: “dynamics of RNA transcription” would be better as “dynamics of mRNA expression”, since the PrimeFlow probes are presumably measuring mRNAs and not nascent transcripts

Thank you for this suggestion. This change has now been made:

“Because flow cytometry antibodies were not available for the candidate genes, we used PrimeFlow to measure the dynamics of mRNA expression...”

- Showing the sequence (e.g. actual nucleotides, with position of PU.1 motif marked, and location of the variant) around rs2836882 would be very helpful in Fig 1, for understanding

how the variant might be affecting PU.1 binding. Are there any other motifs predicted by HOMER to be disrupted?

Thank you for this suggestion. There are several predicted PU.1 motifs at chr21q22 (these were shown relative to rs2836882 in Fig.4i). Due to the size of the region they span (~1-kb) it was not possible to show individual nucleotides for all of these, but we have annotated the sequence that contains nearest predicted PU.1 motif as the referee suggests. This has been added to Fig.1i:

We also ran HOMER using the findMotifs.pl function to compare motif enrichment in minor vs major allele-containing fasta files (200-nt centred on rs2836882). However, no transcription factor motifs were predicted to be disrupted (including for PU.1). This was expected as rs2836882 does not alter a canonical PU.1 motif, but nor does it reveal any other transcription factors that are likely to be directly affected (within the known limitations of predicting transcription factor binding from motifs).

- Overall Fig 1 might benefit from aggregation of all of the genomic tracks into a single figure (e.g. showing fine-mapping posterior probabilities, MPRA data, ChIP-seq, data, allele-specific data. It's a little disorienting to have to examine slightly different boundaries and regions in each of the subpanels

Thank you for this suggestion. In our original submission, we had tried to aggregate genomic tracks, where possible, but Referee #1 objected to data being introduced in a different order to the way it was introduced in the text. We therefore made multiple revisions to this figure to accommodate their request – for example breaking Fig.1a into 3 separate panels (now Fig.1a-c). Unfortunately, it would not be possible to re-aggregate the tracks into a single figure without introducing some data in a different order to how it is introduced in the text as several of the figure panels (1d, 1e, 1f, 1g, 1j) do not include genomic tracks and would have to be presented before or after the genomic track aggregate.

We agree, however, that it was disorientating to have slightly different boundaries in different panels and so have now re-drawn and re-labelled all of the boundaries so that they are easier to cross-reference. For example, the same boundaries are now used in 1b, 1i and 1l, and the axis points shown in 1h and 1k overlap. We have also annotated the MPRA enhancer region in Fig.1k to allow direct cross-referencing. We hope this is less disorientating:

- Fig 1e legend: Currently written as “Data representative of one of four donors.”, which sounds like the data is only representative of one of the four. Reword, if it is consistent between the four donors. E.g. “Data corresponds to one representative donor, out of four.”

Thank you for this suggestion. We have now re-worded this to indicate that the data are representative of all 4 donors, and have used similar descriptions in other figure legends where relevant:

“Data from one representative donor, out of four.”

- Fig 2g — This would be better as a scatterplot of changes in expression in condition 1 vs changes in expression in condition 2. The slope of the relationship would be interesting to show, with regards to whether the enhancer KO leading to a weaker effect than the gene KO

Thank you for this suggestion. We have now changed the GSEA plots that were shown in Fig.2g for a FC/FC scatterplot that shows the relative effects between the chr21q22 enhancer KO and ETS2 KO. As the reviewer hypothesised, the slope of the linear regression line indicates that the enhancer KO leads to generally weaker effects than the gene KO. So as not to remove the peer-reviewed GSEA results altogether, we have moved these to a new Extended Data Fig.5:

g

g. FC/FC plot of genes differentially-expressed by chr21q22 enhancer deletion, plotted against their fold-change after ETS2 editing. Percentages denote upregulated (red) and downregulated genes (blue). Coloured points (blue or red) were differentially expressed following ETS2 editing (FDR ≤ 0.1 , two-sided).

- The Extended Data Figures appear to be often cited as entire figures (e.g. Page 10: “Extended Data Fig.2”) without specifying which panel out of many. It would be helpful to cite specific panels throughout.

Thank you for this suggestion. There does not appear to be an agreed format for citing the Extended Data Figures and note that several recent Nature papers have either cited entire figures (e.g. PMID 38355791) or a mixture of individual panels and whole figures (e.g. PMID 38093011). On some occasions in our manuscript, we did intend to cite the entire Extended Data Figure (e.g. Extended Data Fig.3, Extended Data Fig.5, Extended Data Fig.10) and so have not changed these citations. For instances where we were referring to specific panel(s) we have now amended these citations to indicate the relevant parts of the figure. For example:

“Indeed, increased ETS2 expression in monocytes/macrophages – either at rest or upon early exposure to bacteria – had the same genetic basis as inflammatory disease risk (Extended Data Fig.1c).”

“Two of these polymorphisms were transcriptionally inert, but the third (rs2836882) had the strongest expression-modulating effect of any candidate SNP, with the risk allele (G) increasing transcription – consistent with the ETS2 eQTL (Fig.1h, Extended Data Fig.1b).”

“No differences in cell viability or macrophage marker expression were observed, suggesting that ETS2 was not required for macrophage survival or differentiation (Extended Data Fig.2g,h).”

“To do this, we optimised a method for controlled overexpression of genes in primary macrophages via transfection of in vitro transcribed mRNA that was modified to minimise immunogenicity (Fig.3a, Extended Data Fig.3f, Methods).”

- To interpret Fig. 3g, the numbers of genes in each set should be reported to interpret this figure, because it is easier to get high p-values for enrichment with larger gene sets. If enrichment cannot be directly reported from SNPsea, as the authors state in the rebuttal, then using another method such as MAGMA, which should report both an effect size beta and p-value, would be a good alternative

We respectfully disagree with the referee on this one point, as we think it probably reflects a misunderstanding of how SNPsea works and do not think the suggested analysis would enhance the manuscript or change the main conclusions.

The general point that the referee makes (that it is easier to get higher p-values for enrichment with larger gene sets) is correct when a gene set of variable size is assessed for its enrichment within a larger gene list, as occurs for example in GSEA (i.e. more genes means more chance of finding enrichment). However, in SNPsea the gene set being queried is not the list of genes within the pathway (e.g. ETS2-regulated genes) but rather the set of genes tagged by the disease-associated SNPs. If we apply the referee’s point to the method used by SNPsea, the appropriate analogy would be that the size of the SNP list could affect the p-value, and this is certainly true (and why we included Schizophrenia as a negative control as this has 287 SNPs but shows no enrichment in ETS2 pathways). To control for the size of the SNP list, SNPsea calculates an enrichment p-value using randomly selected null SNP lists that are of exactly the same size and tag exactly the same number of genes as the disease-associated SNPs being tested. As such, the comparison is whether genes tagged by IBD SNPs are more enriched within a particular pathway than multiple equally-sized lists of random genes. Therefore, while you might expect more overlap if the pathway list was larger, this would equally apply to every null gene list and so should not have a major effect on the p-value. To illustrate this point, we randomly downsampled one of the ETS2-regulated gene lists and re-performed SNPsea using smaller numbers of genes (with 3 random downsamples for each number of ETS2 target genes shown). The result for IBD SNPs is

shown below and indicates that the size of the pathway list does not appreciably affect the *P*-value (the dashed line is the Bonferroni significance threshold used in the manuscript):

In response to the referee's comment, we also reached out to the developer of SNPsea (Kamil Slowikowski) to see whether there might be a way to extract the raw enrichment scores that the algorithm uses. He has kindly helped us do this, as these values are not written out in the standard output. Using these values, it is now possible to calculate a standardised enrichment score by comparing the IBD SNP enrichment score with the mean and standard error of the null scores. We have plotted this enrichment score against the $-\log_{10}(P)$ in a new panel in Extended Data Fig.6, which confirms that ETS2-regulated genes are more enriched in IBD genetics than many known disease pathways:

i. Plot of enrichment statistic (standardised effect size) against statistical significance from SNPsea analysis of genes tagged by 241 IBD SNPs within ETS2-regulated genes (red) and known IBD pathways (black)

I

Importantly, the key “take home” message from our SNPsea analysis was that the ETS2 pathway is strongly implicated by IBD genetics, which we think is unequivocally shown by the analysis presented (with 0 out of 5,000,000 nulls being more enriched) and by identifying over 20 individual IBD risk genes that are regulated by ETS2. This is now further supported by the enrichment statistics we have added. While we recognise that other methods exist, including MAGMA, LDSC, and GWASjet, these all have pros/cons and none has been proven to be significantly better than any others. Since our SNPsea result is not ambiguous – and we can now provide enrichment scores for the tested pathways – we do not think there is a scientific need for additional analysis, and hope that this addresses the Referee’s comment.